# PROVABLE BENEFIT OF ADAPTIVITY IN ADAM

## ABSTRACT

The Adaptive Moment Estimation (Adam) algorithm is widely adopted in practical applications due to its fast convergence. However, its theoretical analysis is still far from satisfactory. Existing convergence analyses for Adam rely on the bounded smoothness assumption, referred to as the *L-smooth condition*. Unfortunately, this assumption does not hold for many deep learning tasks. Moreover, we believe that this assumption obscures the true benefit of Adam, as the algorithm can adapt its update magnitude according to local smoothness. This important feature of Adam becomes irrelevant when assuming globally bounded smoothness. This paper studies the convergence of randomly reshuffled Adam (RR Adam), which is the major version of Adam adopted in deep learning. We present the first convergence analysis of RR Adam without the bounded smoothness assumption. We demonstrate that RR Adam can maintain its convergence properties when smoothness is linearly bounded by the gradient norm, referred to as the $(L_0, L_1)$-*smooth condition*. Further, under the same setting, we refine the existing lower bound of SGD and show that SGD can be arbitrarily slower than Adam. To our knowledge, this is the first time that Adam and SGD are rigorously compared in the same setting where the advantage of Adam can be revealed. Our theoretical results shed new light on the advantage of Adam over SGD.

## 1 INTRODUCTION

Machine learning tasks are often formulated as solving the following finite-sum problem.

$$\min_{\boldsymbol{w} \in \mathbb{R}^d} f(\boldsymbol{w}) = \sum_{i=0}^{n-1} f_i(\boldsymbol{w}), \tag{1}$$

where $n$ denotes the number of samples or mini-batches, and $\boldsymbol{w}$ denotes the trainable parameters. Recently, it is noted that adaptive gradient methods including Adaptive Moment estimation (Adam) (Kingma and Ba, 2014) are widely used to train modern deep neural networks including GANs (Brock et al., 2018), BERTs (Kenton and Toutanova, 2019), GPTs (Brown et al., 2020) and ViTs (Dosovitskiy et al., 2020). It is often observed that Adam converges considerably faster than vanilla Stochastic Gradient Descent (SGD) for the training of Transformers, as seen in Figure 1(a). Similar phenomena are also reported in BERT training (Zhang et al., 2019b).

Despite its practical success, the theoretical analysis of Adam is less than satisfactory. Existing analyses rely on bounded smoothness assumption, i.e., the Lipschitz coefficient of gradients (or the spectrum norm of the Hessian) is globally upper bounded by constant $L$, referred to as *L-smooth condition*. However, recent studies show that the $L$-smooth condition does *not* hold in practical deep learning tasks such as LSTM (Zhang et al., 2019a) and Transformers (Crawshaw et al., 2022).

Moreover, such an assumption hides the benefit of Adam. Intuitively, Adam can overcome the issue of unbounded smoothness using adaptive learning rate. First, Adam uses the reciprocal of the square root of the exponential moving averages of past squared gradients as an effective learning rate (see Algorithm 1 for the update rule). Thus, the effective learning rate would be adapted to the local gradient norm. Second, there is a strong correlation between the Lipschitz coefficient and the gradient norm of deep neural networks (Zhang et al., 2019a; Cohen et al., 2021; Crawshaw et al., 2022). As a result, Adam can adapt the update magnitude to the local Lipschitz coefficient and is empirically observed to converge fast (Figure 1(a) and (Zhang et al., 2019a)). Unfortunately, such benefit is hidden because existing theories of Adam are built upon $L$-smooth condition.

To reveal the theoretical benefit of Adam, we analyze its convergence under a relaxed smoothness condition called $(L_0, L_1)$-smooth condition (Zhang et al., 2019a):

$$\|\nabla^2 f(\boldsymbol{w})\| \leq L_0 + L_1 \|\nabla f(\boldsymbol{w})\|. \tag{2}$$

When $L_1 = 0$, Eq. (2) degenerates into classical $L$-smooth condition. The $(L_0, L_1)$-smooth condition allows the spectral norm of the Hessian (Lipschitz coefficient of gradients) to linearly grow with the gradient norm of $\boldsymbol{w}$, so it is a relaxed version of $L$-smooth condition. The $(L_0, L_1)$-smooth condition is empirically observed to hold in LSTM (Zhang et al., 2019a; 2020) and Transformers (Figure 1(b) and (Crawshaw et al., 2022)).

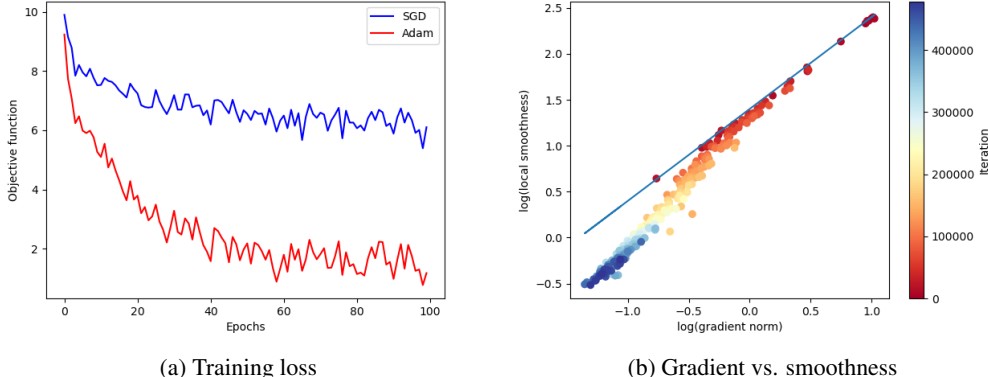

(a) Training loss  (b) Gradient vs. smoothness

Figure 1: Experiments on the WMT 2014 dataset trained with the transformer. **(a):** The training loss of SGD and Adam. **(b):** The gradient norm vs. the local smoothness on the training trajectory. The blue line in (b) stands for $\log(\text{local smoothness}) = \log(\text{gradient norm}) + 1.4$. It can be observed that $(e^{1.4}, 0)$-smooth condition holds in this task. Similar results can be seen in Zhang et al. (2019a).

Under the $(L_0, L_1)$-smooth condition, we successfully separate the convergence rate of Adam from that of SGD. Our contributions are summarized as follows.

- We establish the first convergence result of Adam without "$L$-smoothness". We prove that Adam converges under the $(L_0, L_1)$-smooth condition.

- Our convergence result enjoys several good properties. First,there is no need for the bounded gradient assumption (i.e. $\|\nabla f(\boldsymbol{w})\| \leq C$). Eliminating this assumption is essential since the $(L_0, L_1)$-smooth condition would otherwise degenerate to the $L$-smooth condition. Second, our result does not rely on other assumptions such as a bounded adaptor or a large regularizer for numerical stability. Lastly, the convergence holds for every possible trajectory, which is not only technically demanding but also much stronger than "convergence in expectation".

- We present an improved lower bound for (S)GD under the $(L_0, L_1)$-smooth condition. In this lower bound, there is a factor related to the gradient norm of the initial point, which does not exist in the upper bound of Adam. This indicates that (S)GD may converge arbitrarily slow under the $(L_0, L_1)$-smooth condition, showing the advantage of Adam over (S)GD. To our knowledge, this is the first time that Adam and SGD are rigorously compared in the same setting where the advantage of Adam can be revealed. We believe these results shed new light on understanding the benefit of Adam.

## 2 RELATED WORKS

**Convergence analysis for Adam.** Adam is firstly proposed in Kingma and Ba (2015) with a convergence proof. However, the proof is pointed out to have flaws by Reddi et al. (2018) and Reddi et al. (2018) further provide simple counterexamples with which Adam diverges. This discovery caused the convergence analysis of Adam to stagnate for a while and motivated a series of works developing variants of Adam without divergent issues (see discussion in Appendix **??**). On the

other hand, vanilla Adam works well in practice and divergence is not empirically observed. This phenomenon motivates researchers to rethink the counterexamples. The counterexamples states "for every $\beta_1 < \sqrt{\beta_2}$, there exists a problem that Adam diverges". That is to say, the divergence statement requires picking $(\beta_1, \beta_2)$ before fixing the problem, while in practice, the algorithmic parameters are often picked according to the problem. Based on this observation, a recent work (Zhang et al., 2022) proves that Adam can converge with $(\beta_1, \beta_2)$ picked after the problem is given.

We categorize the existing results of Adam into two classes based on the sampling strategy: **with-replacement sampling (a.k.a., i.i.d. sampling, abbreviated as "WR") and without-replacement sampling (a.k.a., random reshuffling, abbreviated as "RR").** We believe both sampling strategies are worth studying: WR is more favored among the theory community due to its simple form, whereas RR is widely used among practitioners because it is easy to implement. Further, RR guarantees to pass each data at least once and brings good performance (Bottou, 2009; 2012).

The first line of work analyzes WR Adam. For instance, Zaheer et al. (2018b) shows that WR RMSProp (a simplified version of Adam with $\beta_1 = 0$) converges to the neighborhood of the stationary points. De et al. (2018) prove the convergence of WR RMSProp by assuming the signs of the gradients to remain the same along the trajectory. However, this condition is not guaranteed to hold in practice. Défossez et al. (2020) prove the convergence of WR Adam with $\beta_1 < \beta_2$. However, their convergence bound is inversely proportional to $\xi$, which is the hyperparameter for numerical stability. Consequently, their bound becomes vacuous as $\xi$ approaches zero. This result does not match practical observations because small values of $\xi$, like $10^{-8}$, often yield satisfactory performance. Moreover, employing large values of $\xi$ obscures the effect of $\sqrt{v_k}$, and thus the proof is largely reduced to the proof of SGD. Huang et al. (2021); Guo et al. (2021) provide simple convergence proof for WR Adam with $\beta_1$ close to 1. However, their results require the $\sqrt{v_k}$ to be bounded in a certain interval $[C_l, C_u]$. This condition changes Adam into AdaBound (Luo et al., 2019). In summary, all the above works require certain strong conditions such as bounded $\sqrt{v_k}$ or large $\xi$. Further, they all require bounded gradient ($\|\nabla f(x)\| \leq C$) and bounded smoothness ($L$-smooth) condition.

The second line of works focus on RR Adam. Shi et al. (2021) prove the trajectory-wise convergence of RR RMSProp and Zhang et al. (2022) prove the in-expectation convergence of RR Adam. However, these works both require $L$-smooth condition. Our analysis follows this line of works and provides the first convergence result of RR Adam under relaxed smoothness condition.

**Relaxed smoothness assumption.** There are several attempts on relaxing $L$-smooth condition. Zhang et al. (2019a) proposes $(L_0, L_1)$-smooth condition to theoretically explain the acceleration effect of clipped SGD over SGD. Similar results are also extended to clipped SGD with momentum (Zhang et al., 2020), distributionally-robust optimization (Jin et al., 2021), differentially-private SGD (Yang et al., 2022) and generalized SignSGD (Crawshaw et al., 2022). However, they did not theoretically analyze Adam in this setting. Considering the great empirical impact of Adam, we believe it is important to study Adam in its original form.

One concurrent work (Li et al., 2023) studies the convergence of WR Adam under $(L_0, L_1)$-smooth condition by cleverly constructing certain stopping time. They also propose a variance-reduced variant with better convergence rate. However, their bound on Adam has polynomial dependence over $1/\xi$ (the hyperparameter for numerical stability). Similarly to (De et al., 2018), this result does not match practice observations, since Adam performs well even when $\xi$ is as small as $10^{-8}$. Further, they do not compare the rate of Adam with SGD, probably because there is no directly-applicable lower bound of SGD that can support the rigorous comparsion.

## 3 PRELIMINARIES

This section introduces notations, definitions, and assumptions that are used throughout this work.

**Notations.** We list the notations that are used in the formal definition of the randomly-shuffled Adam and its convergence analysis.

- (Vector) We define $a \odot b$ as the Hadamard product (i.e., component-wise product) between two vectors $a$ and $b$ with the same dimension. We also define $\langle a, b \rangle$ as the $\ell^2$ inner product between $a$ and $b$. We define $\mathbb{1}_d$ as an all-one vector with dimension $d$.

- (Array) We define $[m_1, m_2] \triangleq \{m_1, \cdots, m_2\}, \forall m_1, m_2 \in \mathbb{N}, m_1 \leq m_2$. Specifically, we use $[m] \triangleq \{1, \cdots, m\}$.

- (Asymptotic notation) We define $A_1(x) = \mathcal{O}_{x \to a}(A_2(x))$ if $|\frac{A_1(x)}{A_2(x)}|$ is bounded when $x \to a$. We define $A_2(x) = \Omega_{x \to a}(A_1(x))$ when $A_1(x) = \mathcal{O}_{x \to a}(A_2(x))$. We use $\tilde{\mathcal{O}}$ to denote $\mathcal{O}$ with logarithmic factors hidden, i.e., $A_1(x) = \tilde{\mathcal{O}}_{x \to a}(A_2(x))$ if $A_1(x) = \mathcal{O}_{x \to a}(A_2(x) \log |A_2(x)|)$. When the context is clear, we hide "$x \to a$" and only use $\mathcal{O}, \Omega, \tilde{\mathcal{O}}$.

**Pseudocode.** To facilitate the analysis, we provide the pseudocode of Adam in Algorithm 1.

---

**Algorithm 1** Randomly reshuffled Adam (RR-Adam)

---

**Input:** Objective function $f(\boldsymbol{w}) := \sum_{i=0}^{n-1} f_i(\boldsymbol{w})$, learning rate series $\{\eta_k\}_{k=1}^T$ and hyperparameters $(\beta_1, \beta_2) \in [0, 1)^2$. Initialize the parameter $\boldsymbol{w}_{1,0} \in \mathbb{R}^d$, the conditioner $\boldsymbol{\nu}_{1,-1} \in \mathbb{R}^{d, \geq 0}$, and the momentum $\boldsymbol{m}_{1,-1} \in \mathbb{R}^d$.

**for** $k = 1$ **to** $T$ **do**
    Randomly shuffle $[0, n-1]$ to get $\{\tau_{k,j}\}_{j=0}^{n-1}$
    **for** $i = 0$ **to** $n - 1$ **do**
        Calculate $g_{k,i} = \nabla f_{\tau_{k,i}}(\boldsymbol{w}_{\tau_{k,i}})$
        Update $\boldsymbol{\nu}_{k,i} = \beta_2 \boldsymbol{\nu}_{k,i-1} + (1 - \beta_2) g_{k,i}^{\odot 2}$,
        Update $\boldsymbol{m}_{k,i} = \beta_1 \boldsymbol{m}_{k,i-1} + (1 - \beta_1) g_{k,i}$
        Update $\boldsymbol{w}_{k,i+1} = \boldsymbol{w}_{k,i} - \eta_k \frac{1}{\sqrt{\boldsymbol{\nu}_{k,i}} + \xi} \odot \boldsymbol{m}_{k,i}$
    **end for**
    Update $\boldsymbol{\nu}_{k+1,-1} = \boldsymbol{\nu}_{k,n-1}, \boldsymbol{m}_{k+1,-1} = \boldsymbol{m}_{k,n-1}, \boldsymbol{w}_{k+1,0} = \boldsymbol{w}_{k,n}$
**end for**

---

$\boldsymbol{m}_{k,i}$ and $\boldsymbol{\nu}_{k,i}$ are weighted averages with hyperparamter $\beta_1 \in [0, 1)$ and $\beta_2 \in [0, 1)$, respectively. $\xi$ is adopted for numerical stability and it is often chosen to be $10^{-8}$ in practice. In our theory, we allow $\xi$ to be an arbitrary non-negative constant including 0.

Algorithm 1 follows a without-replacement sampling strategy (also known as shuffling), which is the default strategy used in CV, NLP, GANs, etc. However, it is not necessarily easy to analyze shuffling strategy, because the stochastic gradients sampled by random-shuffling lack statistical unbiasedness, i.e. $\mathbb{E}[\nabla f_{k,i}(x_{k,i})|x_{k,i}] \neq \nabla f(x_{k,i})$. This bias requires a much different analysis from its with-replacement counterpart. Even for SGD, the analysis for shuffling is often known to be "more challenging" (Tran et al., 2021; Mishchenko et al., 2020). However, we choose to study this version as it is closer to the practice.

We make two mild assumptions on the objective function (Eq. (1)).

**Assumption 3.1** ($(L_0, L_1)$-smooth condition). $f_i(\boldsymbol{w})$ satisfies $(L_0, L_1)$-smooth condition, i.e., there exist positive constants $(L_0, L_1)$, such that, $\forall \boldsymbol{w}_1, \boldsymbol{w}_2 \in \mathbb{R}^d$ satisfying $\|\boldsymbol{w}_1 - \boldsymbol{w}_2\| \leq \frac{1}{L_1}$,

$$\|\nabla f_i(\boldsymbol{w}_1) - \nabla f_i(\boldsymbol{w}_2)\| \leq (L_0 + L_1 \|\nabla f_i(\boldsymbol{w}_1)\|) \|\boldsymbol{w}_1 - \boldsymbol{w}_2\|. \tag{3}$$

Eq. (3) is firstly introduced by Zhang et al. (2020). When $f(\boldsymbol{w})$ is twice differentiable, Eq. (3) is equivalent to Eq. (2) (Zhang et al., 2020). We will use the version of Eq. (3) since it does not require $f(\boldsymbol{w})$ to be twice differentiable.

**Assumption 3.2** (Affine Noise Variance). $\forall \boldsymbol{w} \in \mathbb{R}^d$, the gradients of $\{f_i(\boldsymbol{w})\}_{i=0}^{n-1}$ has the following connection with the gradient of $f(\boldsymbol{w})$:

$$\sum_{i=0}^{n-1} \|\nabla f_i(\boldsymbol{w})\|^2 \leq D_1 \|\nabla f(\boldsymbol{w})\|^2 + D_0.$$

Assumption 3.2 generalizes the "bounded variance" assumption (which requires $D_1 = 1/n$) (Ghadimi et al., 2016; Zaheer et al., 2018a; Huang et al., 2021) and the "strongly growth condition" (which requires $D_0 = 0$) (Schmidt and Roux, 2013; Vaswani et al., 2019). Assumption 3.2 allows flexible choices of $D_0$ & $D_1$ and thus it is among the weakest assumption of this kind.

# 4    ADAM CONVERGES UNDER THE $(L_0, L_1)$-SMOOTH CONDITION

**Theorem 4.1.** *Consider RR-Adam defined as Algorithm 1 with diminishing learning rate $\eta_k = \frac{\eta_1}{\sqrt{k}}$. Let Assumptions 3.1 and 3.2 hold. Suppose the hyperparamters satisfy: $0 \le \beta_1^2 < \beta_2 < 1$ and $\beta_2$ is larger than a threshold $\gamma(D_1)$. Then, we have*

$$\min_{k \in [1,T]} \left\{ \frac{\|\nabla f(\boldsymbol{w}_{k,0})\|}{\sqrt{D_1}}, \frac{\|\nabla f(\boldsymbol{w}_{k,0})\|^2}{\sqrt{D_0}} \right\} \le \tilde{\mathcal{O}} \left( \frac{f(\boldsymbol{w}_{1,0}) - \min_{\boldsymbol{w}} f(\boldsymbol{w})}{\sqrt{T}} \right) + \mathcal{O}((1-\beta_2)^2 \sqrt{D_0}). \quad (4)$$

For simplicity, we defer the concrete form of $\gamma$ to Appendix D.2. We provide some remarks on the results as follows.

**On the presentation of Theorem 4.1.** As we state in Section 2, our analysis follows the line of (Shi et al., 2021; Zhang et al., 2022), and we present our theorem in a similar way: we use $T$ as the number of epochs instead of the number of iterations (in which case the total iteration would be $nT$). We use asymptotic notations since the concrete coefficiencies are rather complex (we defer a detailed statement of the theorem to Appendix D.2). Here we highlight that the hidden coefficiency of $\frac{f(\boldsymbol{w}_{1,0}) - \min_{\boldsymbol{w}} f(\boldsymbol{w})}{\sqrt{T}}$ is $\Theta(\frac{1}{(1-\beta_2)^2})$. Therefore, if $\beta_2 = 1$, the right-hand-side of Eq. (4) vanishes, and thus Theorem 4.1 can not be applied to SGDM to derive the convergence rate.

**On the range of hyperparameters.** Theorem 4.1 indicates that Adam can work when $\beta_2$ is close enough to 1. This matches the practical choice of $\beta_2$ (e.g., 0.999 in default setting, 0.95 in the GPT-3 training (Brown et al., 2020)). Note that our result does not contradict the counterexamples of Adam's non-convergence (Reddi et al., 2018; Zhang et al., 2022), as these divergence results require $\beta_2$ to be small and thus not close to 1. Rather, these counterexamples suggest that large $\beta_2$ is necessary for convergence. As for $\beta_1$, Theorem 4.1 needs $\beta_1^2 < \beta_2$. When $\beta_2$ is large, Theorem 4.1 allows a wide range of candidates of $\beta_1$ (e.g., 0.9 in default setting and 0.5 in GAN (Radford et al., 2015)).

**On the advantages besides $(L_0, L_1)$-smooth condition.** We emphasize some more advantages of Theorem 4.1. First, Theorem 4.1 does not require that the gradient norm is bounded. This is vital since otherwise $(L_0, L_1)$-smooth condition will degenerate to $L$-smooth condition. Further, we do not assume the adaptive learning rate $\eta_k/\sqrt{\boldsymbol{\nu}_k}$ to be upper bounded. We do not assume $\xi$ to be large, which agrees with the deep learning libraries because small $\xi$ such as $10^{-8}$ usually works well. Our theorem allows any non-negative $\xi$ including 0. Finally, Theorem 4.1 holds for every possible trajectory, which is much stronger than the common results of "convergence in expectation" and is technically challenging.

**On the neighborhood of stationary points.** When $D_0 \ne 0$, Theorem 4.1 only ensures that Adam converges to a neighborhood of stationary points $\{\boldsymbol{w} : \min \{\frac{\|\nabla f(\boldsymbol{w}))\|}{\sqrt{D_1}}, \frac{\|\nabla f(\boldsymbol{w})\|^2}{\sqrt{D_0}}\} \le \mathcal{O}((1-\beta_2)\sqrt{D_0})\}$. Since SGD converges to the stationary points with diminishing learning rate, one may wonder if Theorem 4.1 can be improved to obtain the same conclusion as SGD. Unfortunately, there is a counterexample in the existing literature ( function (9) in Zhang et al. (2022)) showing that **Adam does *not* converge to stationary points** even if all the conditions in Theorem 4.1 are satisfied. Specifically, Zhang et al. (2022) consider the following function:

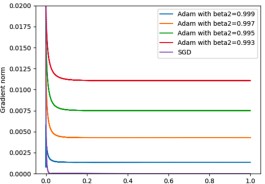

Figure 2: Reconduct of experimental results from (Zhang et al., 2022).

$$f(x) = \sum_{j=0}^{9} f_j(x) = \frac{1}{10}x^2 - 1, \text{ where } f_j(x) = \begin{cases} (x-1)^2 & \text{if } j = 0 \\ -0.1 \left(x - \frac{10}{9}\right)^2 & \text{if } 1 \le j \le 9 \end{cases}.$$

One can easily verify such an example satisfies Assumptions 3.2 and 3.1 with $D_0 > 0$. As shown in Figure 2, when running Adam (with $\beta_1 = 0.9, \eta_k = 0.1/\sqrt{k}, a = 3, x_0 = -2$), it does not converge to exact stationary points. Instead, it converges to a neighborhood of stationary points with size inversely proportional to $\beta_2$. Therefore, the non-vanishing term in Theorem 4.1 is *not* due to the limitation of the proof. Rather, it is an intrinsic property of Adam.

Why cannot Adam converge to exact stationary points when $D_0 > 0$? Intuitively, this is because even with diminishing $\eta_k$, the effective learning rate $\frac{\eta_k}{\xi \mathbb{1}_d + \sqrt{\boldsymbol{\nu}_{k,i}}}$ may not diminish due to the potentially

decreasing $\sqrt{\boldsymbol{\nu}_{k,i}}$. The good news is that $\mathcal{O}((1 - \beta_2)\sqrt{D_0})$ approaches 0 as $\beta_2$ gets close to 1. This means that the neighborhood shrinks as $\beta_2 \to 1$ (this is also observed in Figure 2). As discussed above, the practical use of $\beta_2$ is close to 1, and thus $O((1 - \beta_2)\sqrt{D_0})$ is tolerable.

## 5 PROOF SKETCH OF THEOREM 4.1

In this section, we briefly explain our proof idea for Theorem 4.1, which can be divided into two stages. In Stage I, we will prove Theorem 4.1 for Adam with $\beta_1 = 0$ to show the challenge brought by $(L_0, L_1)$-smooth condition and how we tackle it. In Stage II, we then show the additional difficulty when adding the momentum and our corresponding intuition to solve it.

**Stage I: Convergence of Adam with $\beta_1 = 0$.** By the descent lemma,

$$f(\boldsymbol{w}_{k+1,0}) - f(\boldsymbol{w}_{k,0}) \leq \underbrace{\langle \boldsymbol{w}_{k+1,0} - \boldsymbol{w}_{k,0}, \nabla f(\boldsymbol{w}_{k,0}) \rangle}_{\text{First Order}} + \underbrace{\frac{L_{loc}}{2} \|\boldsymbol{w}_{k+1,0} - \boldsymbol{w}_{k,0}\|^2}_{\text{Second Order}}, \quad (5)$$

where $L_{loc}$ is the local smoothness. We bound the first-order and the second-order term respectively. The upper bound on second-order term is relatively simple. Due to the limited space, we only show the idea of bounding first-order term here.

The ever-changing adaptive learning rate poses a challenge on deriving the bound. It is even noted that with small $\beta_2$, the first order term can be positive (Reddi et al., 2018). However, we notice that if $\boldsymbol{\nu}_{k,i}$ is stationary, i.e., RMSProp degenerates to SGD with preconditioning, the first order term equals to $-\eta_k \langle \sum_i \frac{1}{\xi \mathbb{1}_d + \sqrt{\boldsymbol{\nu}_{k,0}}} \odot \nabla f_{\tau_{k,i}}(\boldsymbol{w}_{k,i}), \nabla f(\boldsymbol{w}_{k,0}) \rangle \approx -\eta_k \langle \sum_i \frac{1}{\xi \mathbb{1}_d + \sqrt{\boldsymbol{\nu}_{k,0}}} \odot \nabla f_{\tau_{k,i}}(\boldsymbol{w}_{k,0}), \nabla f(\boldsymbol{w}_{k,0}) \rangle$, which is indeed negative. While that "$\boldsymbol{\nu}_{k,i}$ is stationary" is too good to be true, we prove that $\boldsymbol{\nu}_{k,i}$ changes little when $\beta_2$ is close to 1, assuming that the gradient is large. Below we denote $\boldsymbol{\nu}_{l,k,i}$ as the $l$-th component of $\boldsymbol{\nu}_{k,i}$.

**Lemma 5.1** (Informal). *For any $l \in [d]$ and $i \in [0, n-1]$, if $\max_{p \in [0, n-1]} |\partial_l f_p(\boldsymbol{w}_{k,0})| = \Omega(\sum_{r=1}^{k-1} \beta_2^{\frac{(k-1-r)}{2}} \eta_r \|\nabla f(\boldsymbol{w}_{r,0})\| + \eta_k)$, then $|\boldsymbol{\nu}_{l,k,i} - \boldsymbol{\nu}_{l,k,0}| = \mathcal{O}((1 - \beta_2)\boldsymbol{\nu}_{l,k,0})$.*

The idea of Lemma 5.1 is simple: since $\boldsymbol{\nu}_{k,i} = \beta_2 \boldsymbol{\nu}_{k,i-1} + (1 - \beta_2) \nabla f_{\tau_{k,i}}(\boldsymbol{w}_{\tau_{k,i}})^{\odot 2}$, the change of $\boldsymbol{\nu}_{k,i}$ w.r.t. $i$ should be small when $\beta_2$ is large. However, we need to check that the relative size of $\nabla f_{\tau_{k,i}}(\boldsymbol{w}_{\tau_{k,i}})^{\odot 2}$ w.r.t. $\boldsymbol{\nu}_{k,i-1}$ is uniformly bounded across varying $\beta_2$, otherwise the term $(1 - \beta_2)\nabla f_{\tau_{k,i}}(\boldsymbol{w}_{\tau_{k,i}})^{\odot 2}$ may not go to zero when $\beta_2 \to 1$. We resolve this challenge by expanding $\boldsymbol{\nu}_{k,i}$ in terms of squared gradients and bounding the gap between each of the terms and $\nabla f_{\tau_{k,i}}(\boldsymbol{w}_{\tau_{k,i}})^{\odot 2}$ by echoing $(L_0, L_1)$-smooth condition. We defer a detailed proof to Corollary D.9 for details.

As a conclusion, if we denote those dimensions with large gradients (i.e., satisfying the requirement of Lemma 5.1) as $\mathbb{L}_{large}^k$ and the rest as $\mathbb{L}_{small}^k$, Lemma 5.1 indicates that the $\mathbb{L}_{large}^k$ part (i.e., $\sum_{l \in \mathbb{L}_{large}^k} (\boldsymbol{w}_{l,k+1,0} - \boldsymbol{w}_{l,k,0})\partial_l f(\boldsymbol{w}_{k,0}))$ in the first order term can be bounded as

$$-\eta_k \sum_{l \in \mathbb{L}_{large}^k} \frac{\partial_l f(\boldsymbol{w}_{k,0})}{\sqrt{\boldsymbol{\nu}_{l,k,i}} + \xi} \sum_i \partial_l f_{\tau_{k,i}}(\boldsymbol{w}_{k,i})$$

$$\approx -\eta_k \sum_{l \in \mathbb{L}_{large}^k} \left( \frac{\partial_l f(\boldsymbol{w}_{k,0})^2}{\sqrt{\boldsymbol{\nu}_{l,k,0}} + \xi} + \mathcal{O}\left( (1 - \beta_2) \frac{\partial_l |f(\boldsymbol{w}_{k,0})| \sum_i |\partial_l f_{\tau_{k,i}}(\boldsymbol{w}_{k,i})|}{\sqrt{\boldsymbol{\nu}_{l,k,0}} + \xi} \right) \right)$$

$$= -\Omega \left( \eta_k \min \left\{ \frac{\|\nabla f(\boldsymbol{w}_{k,0})\|}{\sqrt{D_1}}, \frac{\|\nabla f(\boldsymbol{w}_{k,0})\|^2}{\sqrt{D_0}} \right\} \right) + O(\eta_k (1 - \beta_2)\sqrt{D_0}).$$

The last equation uses the affine noise assumption (Assumption 3.2), and we defer a detailed proof to Appendix D.4. A remaining problem is how to deal with those components in $\mathbb{L}_{small}^k$. We treat them as error terms. Concretely, $l \in \mathbb{L}_{small}^k$ indicates that $\partial_l f(\boldsymbol{w}_{k,0}) = \mathcal{O}(\sum_{r=1}^{k-1} \beta_2^{\frac{(k-1-r)}{2}} \eta_r \|\nabla f(\boldsymbol{w}_{r,0})\| + \eta_k)$. Applying it directly into $\sum_{l \in \mathbb{L}_{small}^k} (\boldsymbol{w}_{l,k+1,0} - \boldsymbol{w}_{l,k,0})\partial_l f(\boldsymbol{w}_{k,0})$, we have

$$-\eta_k \sum_{l \in \mathbb{L}_{large}^k} \frac{\partial_l f(\boldsymbol{w}_{k,0})}{\sqrt{\boldsymbol{\nu}_{l,k,i}} + \xi} \sum_i \partial_l f_{\tau_{k,i}}(\boldsymbol{w}_{k,i}) = \mathcal{O}\left( \eta_k \left( \sum_{r=1}^{k-1} \beta_2^{\frac{(k-1-r)}{2}} \eta_r \|\nabla f(\boldsymbol{w}_{r,0})\| + \eta_k \right) \right),$$

where the equation is because $\frac{\partial_l f_{\tau_{k,i}}(\boldsymbol{w}_{k,i})}{\sqrt{\boldsymbol{\nu}_{l,k,i}}+\xi}$ is bounded (proved by Lemma D.3).

In order to upper bound the first order term, we then need to prove that $-\Omega(\eta_k \min\{\frac{\|\nabla f(\boldsymbol{w}_{k,0})\|}{\sqrt{D_1}}, \frac{\|\nabla f(\boldsymbol{w}_{k,0})\|^2}{\sqrt{D_0}}\})$ dominates $\mathcal{O}(\eta_k(\sum_{r=1}^{k-1}\beta_2^{\frac{(k-1-r)}{2}}\eta_r\|\nabla f(\boldsymbol{w}_{r,0})\| + \eta_k))$. This is not necessarily true, as the historical gradient norms in the latter term can be large.

*Remark* 5.2. We recognize this as the challenge brought by $(L_0, L_1)$-smooth condition, since the latter term degenerates to $\mathcal{O}(\eta_k^2)$ with $L$-smooth condition, which is minor ($\sum_{k=1}^T \eta_k^2$ is only in order $\log T$).

We address this challenge by noting that what we need to bound is the sum of the first order term. Fortunately, although we cannot upper bound the first order term in one single epoch, we can bound the sum of it across epochs. By a sum order change, the sum of $\mathcal{O}(\eta_k(\sum_{r=1}^{k-1}\beta_2^{\frac{(k-1-r)}{2}}\eta_r\|\nabla f(\boldsymbol{w}_{r,0})\| + \eta_k))$ over $k$ equals to $\mathcal{O}(\sum_{k=1}^T \eta_k^2\|\nabla f(\boldsymbol{w}_{k,0})\| + \ln T)$. This is smaller by the sum of $-\Omega(\eta_k \min\{\frac{\|\nabla f(\boldsymbol{w}_{k,0})\|}{\sqrt{D_1}}, \frac{\|\nabla f(\boldsymbol{w}_{k,0})\|^2}{\sqrt{D_0}}\})$ by order of $\eta_k$ except a $\ln T$ term due to the mean value inequality $\eta_k^2\|\nabla f(\boldsymbol{w}_{k,0})\| \le \mathcal{O}(\eta_k^2) + \mathcal{O}(\eta_k^2\sqrt{\frac{D_1}{D_0}}\|\nabla f(\boldsymbol{w}_{k,0})\|^2)$. We then conclude the sum of the first order term is $-\Omega(\eta_k \min\{\frac{\|\nabla f(\boldsymbol{w}_{k,0})\|}{\sqrt{D_1}}, \frac{\|\nabla f(\boldsymbol{w}_{k,0})\|^2}{\sqrt{D_0}}\}) + \mathcal{O}(\ln T)$.

**Stage II: adding the momentum.** The second order term of Adam can be bounded similarly. However, the analysis of the first order term becomes more challenging even though we still have $\boldsymbol{\nu}_{k,i} \approx \boldsymbol{\nu}_{k,0}$. Specifically, even with constant $\boldsymbol{\nu}_{k,i} = \boldsymbol{\nu}_{k,0}$, $-\eta_k\langle \sum_i \frac{\boldsymbol{m}_{k,i}}{\sqrt{\boldsymbol{\nu}_{k,i}}+\xi}, -\nabla f(\boldsymbol{w}_{k,0})\rangle > 0$ is not necessarily correct, as the momentum $\boldsymbol{m}_{k,i}$ contains a heavy historical signal, and may push the update away from the negative gradient direction.

We resolve this challenge by observing that the alignment of $\boldsymbol{w}_{k+1,0} - \boldsymbol{w}_{k,0}$ and $-\nabla f(\boldsymbol{w}_{k,0})$ is required due to that our analysis is based on the potential function $f(\boldsymbol{w}_{k,0})$. However, while this potential function is suitable for the analysis of RMSProp, it is no longer appropriate for Adam based on the above discussion. We need to construct another potential function. Our construction of the potential function is based on the following observation: we revisit the update rule in Algorithm 1 and rewrite it as $\frac{\boldsymbol{m}_{k,i}-\beta_1\boldsymbol{m}_{k,i-1}}{1-\beta_1} = \nabla f_{\tau_{k,i}}(\boldsymbol{w}_{k,i})$.

Notice that the right-hand-side of the above equation contains no historical gradients but only the gradient of the current step! By dividing $(\sqrt{\boldsymbol{\nu}_{k,i}}+\xi)/\eta_k$ above,

$$\frac{\boldsymbol{w}_{k,i+1} - \boldsymbol{w}_{k,i} - \beta_1(\boldsymbol{w}_{k,i} - \boldsymbol{w}_{k,i-1})}{1-\beta_1} \approx -\frac{\eta_k}{\sqrt{\boldsymbol{\nu}_{k,0}}+\xi\mathbb{1}_d} \odot \frac{\boldsymbol{m}_{k,i}-\beta_1\boldsymbol{m}_{k,i-1}}{1-\beta_1} = -\frac{\eta_k}{\sqrt{\boldsymbol{\nu}_{k,0}}+\xi\mathbb{1}_d}\odot\nabla f_{\tau_{k,i}}(\boldsymbol{w}_{k,i}).$$

After simple rearrangement, one can see that the sequence $\{\boldsymbol{u}_{k,i} \triangleq \frac{\boldsymbol{w}_{k,i}-\beta_1\boldsymbol{w}_{k,i-1}}{1-\beta_1}\}$ are (approximately) doing SGD within one epoch (with coordinate-wise but constant learning rate $\boldsymbol{\nu}_{k,i}$)! Further notice that the distance between $\boldsymbol{u}_{k,i} = \boldsymbol{w}_{k,i} + \beta_1\frac{\boldsymbol{w}_{k,i}-\boldsymbol{w}_{k,i-1}}{1-\beta_1}$ and $\boldsymbol{w}_{k,i}$ is in order of one step's update, and thus $\boldsymbol{u}_{k,i} \approx \boldsymbol{w}_{k,i}$. Therefore, we choose our potential function as $f(\boldsymbol{u}_{k,i})$. The Taylor's expansion of $f$ at $\boldsymbol{u}_{k,0}$ then provides a new descent lemma, i.e.,

$$f(\boldsymbol{u}_{k+1,0}) - f(\boldsymbol{u}_{k,0}) \le \underbrace{\langle \boldsymbol{u}_{k+1,0} - \boldsymbol{u}_{k,0}, \nabla f(\boldsymbol{u}_{k,0})\rangle}_{\text{First Order}} + \underbrace{\frac{L_0 + L_1\|\nabla f(\boldsymbol{w}_{k,0})\|}{2}\|\boldsymbol{w}_{k+1,0} - \boldsymbol{w}_{k,0}\|^2}_{\text{Second Order}}, \quad (6)$$

By noticing $\boldsymbol{w}_{k,i} \approx \boldsymbol{u}_{k,i} \approx \boldsymbol{u}_{k,0}$, the first order term can be further approximated by $-\langle \frac{\eta_k}{\sqrt{\boldsymbol{\nu}_{k,0}}+\xi\mathbb{1}_d} \odot \nabla f(\boldsymbol{w}_{k,0}), \nabla f(\boldsymbol{w}_{k,0})\rangle$ which is negative. The rest of the proof is the same as that of Stage I.

*Remark* 5.3. We notice that similar potential functions have already been applied in the analysis of other momentum-based optimizers, e.g., momentum (S)GD in (Ghadimi et al., 2015) and (Liu et al., 2020b) and Adam-type optimizers (except Adam) in (Chen et al., 2018b). However, extending the proof to Adam is highly-nontrivial. The key difficulty lies in showing that the first-order expansion of $f(\boldsymbol{u}_{k,0})$ is positive, which further requires that the adaptive learning rate does not change much within one epoch. This is hard for Adam as the adaptive learning rate of Adam can be non-monotonic. The lack of $L$-smooth condition makes the proof even challenging due to the unbounded error brought by gradient norms.

## 6 COMPARISON BETWEEN ADAM AND SGD

Now we compare the convergence rate of Adam with SGD. To do so, we need a lower bound of SGD in the same setting as Theorem 4.1. There are several existing lower bounds of SGD under $(L_0, L_1)$ smoothness condition (e.g., (Zhang et al., 2019a; Crawshaw et al., 2022)). However, we find these lower bounds cannot be directly applicable for comparison with Adam. This is because:

- 1) The lower bounds in (Zhang et al., 2019a; Crawshaw et al., 2022) can only be applied to SGD with constant learning rate. However, to compare with diminishing-learning-rate Adam in Theorem 4.1, we need a lower bound of SGD with diminishing learning rate.

- 2) In the lower bound of (Zhang et al., 2019a; Crawshaw et al., 2022), they pick the learning rate *before* the construction of the objective function and initialization point (we restate their lower bound in Appendix B.1 for completeness). In other words, it is possible that if we fix the objective function and tune the learning rate (which is a common practice in the training of deep neural networks), SGD can converge very fast. For rigorous comparison with Adam, we need a lower bound with reversed ordering. That is, we need the following statement: "consider a fixed objective function and initialization point, then no matter how we pick the learning rate, SGD suffers from a certain rate."

Unfortunately, there is no existing lower bound that satisfies the above two properties. In the following theorem, we provide a refined lower bound of SGD in the setup that we desired.

**Theorem 6.1.** *For any $L_0, L_1, T > 0$, there exists an objective function $f$ obeying Assumption 3.1, and an initialized parameter $\boldsymbol{w}_0$ satisfying $M = \sup\{\|\nabla f(\boldsymbol{w})\| : f(\boldsymbol{w}) \leq f(\boldsymbol{w}_0)\}$, such that $\forall \eta_1 > 0$, the iterations of SGD $\{\boldsymbol{w}_t\}_{t=0}^{\infty}$ satsifies $\min_{t \in T} \|\nabla f(\boldsymbol{w}_t)\|^2 = \Omega(M(f(\boldsymbol{w}_0) - \min_{\boldsymbol{w} \in \mathbb{R}^d} f(\boldsymbol{w}))/\sqrt{T})$.*

The proof can be in Appendix C. The proof idea is mainly motivated by Zhang et al. (2019b). We highlight some differences when we try to reach the two properties mentioned previously.

- To change constant learning rate into diminishing learning rate, we show that: when the initial learning rate $\eta_0$ is larger than a certain threshold, the decay rate of the learning rate cannot offset the curvature explosion along the iteration, causing divergence; on the other hand, when initial $\eta_0$ is small, it would lead to slow convergence. This is a new finding in $(L_0, L_1)$ setting. We prove this result by mathematical induction. This part of the discussion is not required in the lower bound of Zhang et al. (2019b) with constant learning rate.

- To reverse the ordering of "picking learning rate and functions & initialization", we simply augment the worst-case example in Zhang et al. (2019b) into 2 dimensional space. It turns out this simple trick is effective in the proof.

**Comparison between Adam and SGD.** Finally, we discuss the implication the lower bound of SGD (Theorem 6.1) and the upper bound of Adam (Theorem 4.1). In the lower bound of SGD, there is an extra constant $M$ which does not appear in the upper bound of Adam. This allows us to compare the convergence rates of these two algorithms.

We summarize our findings as follows. We emphasize that Theorem 4.1 and Theorem 6.1 share exactly the same setting: both consider function class under the same assumptions; both SGD and Adam use diminishing learning rate. Therefore, the following comparison is rigorous.

**Finding 1:** When $D_0 = 0$. There exists a set of $\boldsymbol{w}$ with infinite Lebesgue measure, such that, when starting at any $\boldsymbol{w}$ in this set, Adam converges (to stationary points) faster than SGD.

**Finding 2:** When $D_0 > 0$. There exists a set of $\boldsymbol{w}$ with infinite Lebesgue measure, such that, when starting at any $\boldsymbol{w}$ in this set, Adam converges (to the neighborhood of stationary points) faster than SGD.

Note that the above statement "algorithm 1 converges faster than algorithm 2" does not mean that algorithm 1 always converges faster than algorithm 2. For sure, rarely can anyone make such a strong statement. The above statement actually means that "the worst-case complexity of algorithm 1 is

faster than that of algorithm 2, and both complexity bounds can be **simultaneously achieved when working on the same function and starting at the same initialization**" [1]

*Proof.* We now prove **Finding 1**. First, we state an important fact from the proof of Theorem 6.1.

**Fact 1:** For the counter-example we constructed in Theorem 6.1. $M = \sup\{\|\nabla f(\boldsymbol{w})\| : f(\boldsymbol{w}) \leq f(\boldsymbol{w}_0)\}$ goes to infinity as $\|\boldsymbol{w}\|$ goes to infinity. Further, for any $C > 0$, the set $\{\boldsymbol{w} : M > C\}$ is of infinite Lebesgue measure.

Based on **Fact 1**, for the worst-case example in Theorem 6.1, there must exist a region in $\mathbb{R}^d$ where $M$ is larger than all the constant terms in the upper bound of Adam in Theorem 4.1. Further, Such region is of infinite Lebesgue measure. When running Adam and SGD simultaneously on this worst-case example starting from any $\boldsymbol{w}$ in this region, the constants in the upper bound of Adam is smaller than the constants in the lower bound of SGD. Since the upper and lower bounds share the same rate, we conclude that Adam converges faster than SGD.

The proof of **Finding 2** is the same as above. We omit it for brevity. □

Note that when $D_0 > 0$, Adam is still guaranteed to converge faster, but only to the neighborhood in lieu of the exact stationary points. We emphasize that this "neighborhood" *cannot* be eliminated since there is a counter-example showing that **Adam cannot reach 0 gradient when $D_0 > 0$** (see Figure 2). So this is an intrinsic property of Adam, rather than the limitation of the theory. Nevertheless, we believe the effect of "not converging to exact stationary points" is minor in practice. This is because: 1) As shown in Theorem 4.1 and Figure 2, the size of the "ambiguity zone" is inversely proportional to $\beta_2$. Since $\beta_2$ is often chosen to be close to 1, the ambiguity zone shrinks and becomes negligible. 2) Machine learning tasks do not pursue high-precision solutions (as much as other fields like PDE). Practitioners usually aim to efficiently find approximate solutions, rather than exact solutions that over-fit the training data.

To our knowledge, the discussion above is the first time that Adam and SGD are rigorously compared in the same setting where the advantage of Adam can be revealed. We believe these results shed new light on understanding the benefit of Adam.

Finally, we briefly explain why the upper bound of Adam is independent of $M$. Intuitively, this is because: (1) it uses different learning rates for different components of $\boldsymbol{w}$. (2) For each component of $\boldsymbol{w}$, the effective learning rate adjusts according to the gradient norm (thus according to the local smoothness). Even though the initial effective learning rate is small, it gets larger when moving in a flat landscape. Combining together, the initial learning rate of Adam can be independent of $M$, and so is its convergence rate.

## 7 CONCLUSIONS AND FUTURE DIRECTIONS

In this paper, we take the first step to theoretically understanding the adaptivity in Adam. We provide the first convergence result under $(L_0, L_1)$-smooth condition, which is realistic and close to practical settings. Then, we refine the lower bound for SGD in the same setting. By comparing the complexity bound of Adam and SGD, we find out that Adam can converge arbitrarily faster than SGD when the initial gradient norm is large.

**Future directions.** One interesting future direction is to find out the benefit of using momentum in Adam. Theorem 4.1 provides the same convergence rate for $\beta_1 = 0$ (RMSProp) and $\beta_1 > 0$ (Adam). In other words, we are still not able to separate the iteration complexity of Adam and RMSProp, and thus we can not explain the benefit of *momentum* in Adam. We believe this is a challenging future work since the effect of momentum is not clear even for SGD with momentum, let alone for Adam. A potential approach is to first theoretically prove the benefit of momentum in SGD with momentum, and then try to adapt it in the analysis of Adam. Further, it is interesting to investigate whether Adam can handle more sharp smooth conditions, e.g., smoothness is bounded by a high-order polynomial of the gradient norm.

---

[1]Here, we follow the definition of "algorithm 1 is faster than algorithm 2" in (Sun and Ye, 2021), which is a widely accepted definition in the optimization field.

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

# A    ADDITIONAL RELATED WORKS

In this section, we provide discussions on more related works.

**Variants of Adam.**    Ever since the counter-example of the convergence of Adam raised by Reddi et al. (2018), many new variants of Adam have been designed. For instance, Zou et al. (2019); Gadat and Gavra (2020); Chen et al. (2018b; 2021) replaced the constant hyperparameters by iterate-dependent ones e.g. $\beta_{1t}$ or $\beta_{2t}$. AMSGrad (Reddi et al., 2019) and AdaFom (Chen et al., 2018b) enforced $\{v_t\}$ to be non-decreasing. Similarly, AdaBound (Luo et al., 2019) imposed constraints $v_t \in [C_l, C_u]$ to prevent the learning rate from vanishing or exploding. Similarly, Zhou et al. (2018b) adopted a new estimate of $v_t$ to correct the bias. In addition, there are attempts to combine Adam with Nesterov momentum (Dozat, 2016) as well as warm-up techniques (Liu et al., 2020a). There are also some works providing theoretical analysis on the variants of Adam. For instance, Zhou et al. (2018a) studied the convergence of AdaGrad and AMSGrad. Gadat and Gavra (2020) studied the asymptotic behavior of a subclass of adaptive gradient methods from landscape point of view. Their analysis applies to RMSprop-variants with iterate-dependent $\beta_{2t}$. In summary, all these works study variants of Adam, which is different from our work since we focus on vanilla Adam.

**Generalization ability of Adaptive gradient methods.**    The generalization ability of Adam is a hot debate topic. For instance, Wang et al. (2021) study the implicit bias of adaptive optimization algorithms on homogeneous neural networks. They proved that the convergent direction of Adam and RMSProp is the same as SGD. Zhou et al. (2020); Xie et al. (2022); Zou et al. (2021) argue that Adam preferred sharp local-min while GD prefers the wide ones. As such, they argue that Adam generalizes worse than SGD. Zou et al. (2021) prove that Adam generalizes worse than SGD over a specific model. There are also several attempts to improve the generalization ability of Adam. For instance, Padam (Chen et al., 2018a) introduces a partial adaptive parameter to improve the generalization performance. AdamW (Loshchilov and Hutter, 2017) improves regularization in Adam by decoupling the weight decay from the gradient-based update.

# B    ADDITIONAL DISCUSSIONS

## B.1    RESTATEMENT OF EXISTING LOWER BOUND OF SGD

We restate the lower bound of GD in Crawshaw et al. (2022) as follows:

**Proposition B.1** (Theorem 2, Crawshaw et al. (2022))**.** *For any $L_0, L_1, \varepsilon > 0, M \geq \max(\frac{L_0}{L_1}, \varepsilon)$, with a fixed constant learning rate $\eta$ for GD, there exists an objective function $f$ obeying Assumption 3.1, and an initialized parameter $\boldsymbol{w}_0$ satisfying $M = \sup\{\|\nabla f(\boldsymbol{w})\| : f(\boldsymbol{w}) \leq f(\boldsymbol{w}_0)\}$, such that denoting $\{\boldsymbol{w}_t\}_{t=0}^{\infty}$ be the iterations of running GD over $f$ with initialization $\boldsymbol{w}_0$ and learning rate $\eta$, if $t \leq \tilde{\Theta}(\frac{M(f(\boldsymbol{w}_0) - \min_{\boldsymbol{w} \in \mathbb{R}^d} f(\boldsymbol{w}))}{\varepsilon^2})$, then $\|\nabla f(\boldsymbol{w}_t)\| > \varepsilon$.*

The above proposition is used to claim that the convergence rate of GD has a dependence over an additional constant $M$, which can be arbitrarily large and make the convergence rate of GD arbitrarily slow. However, we argue that the above proposition is not a standard lower bound and does not suffice to support such a claim. This is because: Proposition B.1 picks the learning rate *before* the construction of the objective function and initialization point. In other words, the construction is based on the knowledge of the learning rate, and it is possible that if we fix the objective function and tune the learning rate (which is a common practice in the training of deep neural networks), GD can converge very fast and be independent of the additional constant $M$. It is more desired that in the lower bound, the objective function and initialization point is constructed before picking the learning rate, which is also a standard setting in other lower bounds Arjevani et al. (2023); Carmon et al.. In the following theorem, we provide a lower bound of the convergence rate of SGD in the more desired setup.

## B.2    COMPARISONS OF OPTIMIZERS OVER THE FINE-TUNING TASK

For the case where the gradient along the trajectory is small, the $(L_0, L_1)$-smooth condition can be easily described by the $L$-smooth condition with small gap, and thus SGD works well. This

may explain the phenomenon that SGD is also adopted in some finetuning tasks, as pretraining can be viewed as selecting a good initialization and we can expect that the gradient is small along the trajectory. It is an interesting future work to formalize the above discussion.

### B.3 Advantage of Adam over the GD/SGD with gradient clipping

Zhang et al. (2019a) shows that GD/SGD with gradient clipping converges under $(L_0, L_1)$ smooth condition. A natural question is that what is the benefit of Adam over GD/SGD with gradient clipping. Honestly, we are not able to give strong theoretical evidence to this question. However, comparing our current result for Adam to the result for GD/SGD with clipping, one advantage is that Adam can handle more complex noise that satisfies affine variance noise assumption while existing analyses of GD/SGD with gradient clipping under the $(L_0, L_1)$-smooth condition all assume that the distance between stochastic gradient and true gradient are bounded with probability 1 ([Zhang et al., 2019] and [Zhang et al. 2020]), which is strictly stronger than ours. It will be interesting to either provide counterexample that SGD with clipping does not converge with affine noise assumption, or prove that SGD with clipping does converge and find other perspective to demonstrate the advantage of Adam over SGD with clipping.

### B.4 Insight for practitioners

First, Adam receives great popularity among practitioners (with more than 100k citations). It is important to theoretically understand this algorithm.

Second, our result theoretically verifies a well-known practitioners' choice: When running experiments on tasks such as Transformers and LSTM training, use Adam instead of SGD.

Third, we provide suggestions for hyperparameter tuning (based on the convergence conditions in Theorem 4.1): when running Adam, we suggest tune up $\beta_2$ and try different $\beta_1$s such that $\beta_1 < \sqrt{\beta_2}$. This suggestion would save much effort of grid-searching the $(\beta_1, \beta_2)$ combination.

## C Proof of Theorem 6.1

In this section, we prove Theorem 6.1. We consider the following function with variable $\boldsymbol{w} = (x, y) \in \mathbb{R}^2$: $f(\boldsymbol{w}) = f((x, y)) = f_1(x) + f_2(y)$, where

$$
f_1(x) = \begin{cases} \dfrac{L_0 \exp^{L_1 x - 1}}{L_1^2} & , x \in [\dfrac{1}{L_1}, \infty), \\[2mm] \dfrac{L_0 x^2}{2} + \dfrac{L_0}{2L_1^2} & , x \in [-\dfrac{1}{L_1}, \dfrac{1}{L_1}], \\[2mm] \dfrac{L_0 \exp^{-L_1 x - 1}}{L_1^2} & , x \in (-\infty, -\dfrac{1}{L_1}]. \end{cases}
\tag{7}
$$

$$
f_2(y) = \begin{cases} \varepsilon(y - 1) + \dfrac{\varepsilon}{2} & , y \in [1, \infty), \\[2mm] \dfrac{\varepsilon}{2} y^2 & , y \in [-1, 1], \\[2mm] -\varepsilon(y + 1) + \dfrac{\varepsilon}{2} & , y \in (-\infty, -1]. \end{cases}
\tag{8}
$$

The construction of both functions (7) and (8) are motivated by Zhang et al. (2019a). One improvement here is that we introduce a single function with variable $\boldsymbol{w} \in \mathbb{R}^2$ $f(\boldsymbol{w}) = f((x, y)) = f_1(x) + f_2(y)$, which helps us to derive a stronger conclusion, i.e., the constructed $f$ is independent of $\eta_1$. It is easy to see that this $f(\boldsymbol{w})$ satisfies $(L_0, L_1)$ condition with $L_0 = L_0$ and $L_1 = L_1$. We now restate Theorem **??** as follows with constants specified.

**Theorem C.1** (Theorem **??**, restated)**.** *Consider function $f(\boldsymbol{w}) = f((x, y)) = f_1(x) + f_2(y)$ with $f_1(x)$ and $f_2(y)$ defined in (7) and (8). Consider gradient descent with diminishing learning rates: $\boldsymbol{w}_{k+1} = \boldsymbol{w}_k - \eta_k \nabla f(\boldsymbol{w}_k)$, where $\eta_k = \frac{\eta_1}{\sqrt{k}}$. Then for any $\epsilon > 0$ and*

$M > \{\frac{2(e^{\frac{\log 2}{\sqrt{2}-1}-1}-\frac{1}{4})L_0}{L_1}, \varepsilon\}$, *there exists an initialization* $\boldsymbol{w}_0 = (x_0, y_0)$ *such that,* $M = \sup\{\|\nabla f(\boldsymbol{w})\| : \boldsymbol{w} \text{ such that } f(\boldsymbol{w}) \leq f(\boldsymbol{w}_0)\}$, *and for any* $\eta_1 > 0$, $\|\nabla f(\boldsymbol{w}_k)\| \geq \epsilon$ *whenever* $k < (\frac{\frac{L_1 M}{2}+\frac{L_0}{4}}{2(1+\sqrt{2})(\log(\frac{L_1 M}{2L_0}+\frac{1}{4})+1)})^2 \frac{(\frac{f_2(y_0)-\min_y f_2(y)}{\varepsilon}-\frac{3}{2})^2}{\varepsilon^2}$.

Before giving the proof of Theorem C.1, we briefly discuss the difference between ours and (Zhang et al., 2019a, Theorem 4). Generally speaking, our result is stronger than (Zhang et al., 2019a, Theorem 4). This is because we pick the function before the learning rate: we prove that there exists a function $f$ and an initialization, such that with any learning rate GD takes a long time to reach the stationary point, while (Zhang et al., 2019a, Theorem 4) picks the learning rate before the function: they prove that with any learning rate, there exists a function $f$ and an initialization, such that GD takes a long time to reach the stationary point.

We next present the proof of Theorem C.1 in **Part I** and **Part II** as follows. For simplicity, we let $\|\cdot\|$ to be the $\ell_\infty$ norm, and the proof can be easily extended to other norms given the equivalence between norms in $\mathbb{R}^2$. We pick $x_0 = \frac{\log(\frac{L_1 M}{2L_0}+\frac{1}{4})+1}{L_1}$ and $y_0 = \frac{f_1(x_0)-\frac{L_0}{2L_1^2}}{\varepsilon} - \frac{1}{2}$. We have $f_1(x_0) - \min_x f_1(x) = f_2(y_0) - \min_y f_2(y)$, and thus $f((x_0, y_0)) - \min_{x,y} f((x, y)) = 2(f_1(x_0) - \min_x f_1(x))$. As $M > \varepsilon$, $\sup\{\|\nabla f(\boldsymbol{w})\| : \boldsymbol{w} \text{ such that } f(\boldsymbol{w}) \leq f(\boldsymbol{w}_0)\}$ is achieved at $(x_0', 0)$ where $x_0'$ satisfies $f_1(x_0') - \min_x f_1(x) = 2(f_1(x_0) - \min_x f_1(x))$. By simple calculation, we have $\sup\{\|\nabla f(\boldsymbol{w})\| : \boldsymbol{w} \text{ such that } f(\boldsymbol{w}) \leq f(\boldsymbol{w}_0)\} = M$.

In the proof, we use $x_k$ to denote the value of $x$ (i.e., the first component of $\boldsymbol{w} \in \mathbb{R}$) in the $k$-th iteration of gradient descent. Similarly for $y_k$.

**Part I: Large $\eta_1$ can cause divergence.** In this part, we prove that: when using the large initial learning rate $\eta_1 \geq \frac{L_1(1+\sqrt{2})|x_0|}{L_0 \exp^{L_1|x_0|-1}}$, decay-learning-rate gradient descent will never reach stationary points.

We prove this claim by induction. When $k = 1$, we claim the following two statements are true:

(1-I): $|x_1| \geq \sqrt{2}|x_0|$.

(1-II): $\eta_2 = \frac{\eta_1}{\sqrt{2}} \geq \frac{L_1(1+\sqrt{2})|x_1|}{L_0 \exp^{L_1|x_1|-1}}$.

We first prove (1-I): without loss of generality, we assume $x_1 > 0$. By the update rule of gradient descent, we have

$$
\begin{aligned}
x_1 &= x_0 - \eta_1 \frac{\partial f(x_0)}{\partial x} \\
&\overset{(7)}{=} x_0 - \eta_1 \frac{L_0 \exp^{L_1 x_0 - 1}}{L_1} \\
&\leq x_0 - \frac{L_1(1+\sqrt{2})|x_0|}{L_0 \exp^{L_1|x_0|-1}} \frac{L_0 \exp^{L_1 x_0 - 1}}{L_1} = -\sqrt{2}x_0.
\end{aligned}
$$

So $|x_1| \geq \sqrt{2}|x_0|$ and (1-1) is proved. We now prove (1-II). Before that, we introduce the following lemma.

**Lemma C.2.** *Consider any* $x, y \in \{z : |z| \geq \frac{\log 2}{(\sqrt{2}-1)L_1}, z \in \mathbb{R}\}$. *When* $|y| > \sqrt{2}|x|$, *then we have* $\frac{|y|}{\exp^{L_1|y|}} \leq \frac{1}{\sqrt{2}} \frac{|x|}{\exp^{L_1|x|}}$.

*Proof.* Let $g(z) = \frac{z}{\exp^{L_1 z}}$. It is easy to see that $\nabla g(z) < 0$ when $z > 0$. Therefore, when $z_1 \geq \sqrt{2} z_2$, we have $g(z_1) \leq g(\sqrt{2} z_2)$. When $z_2 \geq \frac{\log 2}{(\sqrt{2}-1) L_1}$, we have

$$z_2 \geq \frac{\log 2}{(\sqrt{2}-1) L_1}$$
$$\Leftrightarrow \quad \sqrt{2} L_1 z_2 > \log 2 + L_1 z_2$$
$$\Leftrightarrow \quad \exp^{\sqrt{2} L_1 z_2} \geq 2 \exp^{L_1 z_2}$$
$$\Leftrightarrow \quad \frac{1}{\exp^{\sqrt{2} L_1 z_2}} \geq \frac{1}{2 \exp^{L_1 z_2}}$$
$$\Leftrightarrow \quad \frac{\sqrt{2} z_2}{\exp^{\sqrt{2} L_1 z_2}} \geq \frac{z_2}{\sqrt{2} \exp^{L_1 z_2}}.$$

Therefore, we have $\frac{z_1}{\exp^{z_1 L_1}} \leq \frac{\sqrt{2} z_2}{\exp^{\sqrt{2} L_1 z_2}} \geq \frac{z_2}{\sqrt{2} \exp^{L_1 z_2}}$. Proof of Lemma C.2 is completed.

$\square$

Now we prove (1-II):

$$
\eta_2 = \frac{\eta_1}{\sqrt{2}} \quad \geq \quad \frac{L_1(1+\sqrt{2})|x_0|}{L_0 \exp^{L_1|x_0|-1}} \frac{1}{\sqrt{2}}
$$
$$
\overset{\text{(1-I) and Lemma C.2}}{\geq} \quad \frac{L_1(1+\sqrt{2})|x_1|}{L_0 \exp^{L_1|x_1|-1}} \sqrt{2} \frac{1}{\sqrt{2}}
$$
$$
= \quad \frac{L_1(1+\sqrt{2})|x_1|}{L_0 \exp^{L_1|x_1|-1}}
$$

So (1-II) is proved. Now we suppose the following two claims hold for $k = 2m$ where $m \in \mathbb{N}^+$.

(2m-I): $|x_{2m+1}| \geq \sqrt{2}|x_{2m}|$.

(2m-II): $\eta_{2m+2} = \frac{\eta_1}{\sqrt{2m+2}} \geq \frac{L_1(1+\sqrt{2})|x_{2m+1}|}{L_0 \exp^{L_1|x_{2m+1}|-1}}$

Then for $k = 2m + 1$, we prove the following claims hold for $k = 2m + 1$.

((2m+1)-I): $|x_{2m+2}| \geq \sqrt{2}|x_{2m+1}|$.

((2m+1)-II): $\eta_{2m+3} = \frac{\eta_1}{\sqrt{2m+3}} \geq \frac{L_1(1+\sqrt{2})|x_{2m+2}|}{L_0 \exp^{L_1|x_{2m+2}|-1}}$.

We first prove ((2m+1)-I):

$$
x_{2m+2} \quad = \quad x_{2m+1} - \eta_{2m+2} \frac{\partial f(x_{2m+1})}{\partial x}
$$
$$
\overset{(7)}{=} \quad x_{2m+1} - \eta_{2m+2} \frac{L_0}{L_1} \exp^{L_1 x_{2m+1}-1}
$$
$$
\overset{\text{(2m-II)}}{\leq} \quad x_{2m+1} - \frac{L_1(1+\sqrt{2})|x_{2m+1}|}{L_0 \exp^{L_1|x_{2m+1}|-1}} \frac{L_0 \exp^{L_1 x_{2m+1}-1}}{L_1}
$$
$$
\leq \quad -\sqrt{2} x_{2m+1}.
$$

So $|x_{2m+2}| \geq \sqrt{2}|x_{2m+1}|$ and ((2m+1)-I) is proved. We now prove ((2m+1)-II).

$$
\begin{aligned}
\eta_{2m+3} &= \eta_{2m+2}\sqrt{\frac{2m+2}{2m+3}} \\
&\overset{\text{(2m-II)}}{\geq} \frac{L_1(1+\sqrt{2})|x_{2m+1}|}{L_0\exp^{L_1|x_{2m+1}|-1}}\sqrt{\frac{2m+2}{2m+3}} \\
&\overset{\text{(2m-I) and Lemma C.2}}{\geq} \frac{L_1(1+\sqrt{2})|x_{2m+2}|}{L_0\exp^{L_1|x_{2m+2}|-1}}\sqrt{2}\sqrt{\frac{2m+2}{2m+3}} \\
&\geq \frac{L_1(1+\sqrt{2})|x_{2m+2}|}{L_0\exp^{L_1|x_{2m+2}|-1}}.
\end{aligned}
$$

So ((2m+1)-II) is proved. We can derive a similar claim when $k$ is odd. By the principle of induction, we know that $|x_{k+1}| \geq \sqrt{2}|x_k|$ for any $k \geq 1$. Since $f_1(x)$ grows exponentially, gradient descent in **Part I** will never reach stationary points.

**Part II: Small $\eta_1$ can cause slow convergence.** In this part, we prove that: when using initialization $\boldsymbol{w} \in \Omega$, decay-learning-rate gradient descent with *small* initial learning rate $\eta_1 < \frac{L_1(1+\sqrt{2})|x_0|}{L_0\exp^{L_1|x_0|-1}} = \frac{(1+\sqrt{2})(\log(\frac{L_1 M}{2L_0}+\frac{1}{4})+1)}{\frac{L_1 M}{2}+\frac{L_0}{4}}$ will cause slow convergence. For any $k \geq 1$, we have

$$
\begin{aligned}
y_k - y_{k+1} &= \eta_k \frac{\partial f(\boldsymbol{w}_k)}{\partial y} \\
&\overset{(8)}{=} \varepsilon\frac{\eta_1}{\sqrt{k}} \\
&< \varepsilon\frac{L_1(1+\sqrt{2})|x_0|}{L_0\exp^{L_1|x_0|-1}}\frac{1}{\sqrt{k}}
\end{aligned}
$$

Therefore, we have

$$
\sum_{k=1}^{K}(y_{k+1}-y_k) = \sum_{k=1}^{K}\varepsilon\frac{\eta_1}{\sqrt{k}} < 2\sqrt{k}\varepsilon\eta_1 < 2\sqrt{k}\varepsilon\frac{L_1(1+\sqrt{2})|x_0|}{L_0\exp^{L_1|x_0|-1}}
$$

When using initialization $y_0$, it is easy to have the following conclusion: when $2\frac{(1+\sqrt{2})(\log(\frac{L_1 M}{2L_0}+\frac{1}{4})+1)}{\frac{L_1 M}{2}+\frac{L_0}{4}}\sqrt{k} = 2\varepsilon\sqrt{k}\frac{L_1(1+\sqrt{2})|x_1|}{L_0\exp^{L_1|x_1|-1}} < y_0 - 1 = \frac{(f_1(x_0)-\min_x f_1(x))}{\varepsilon} - \frac{3}{2}$, we have $\frac{\partial f(\boldsymbol{w}_k)}{\partial y} = \varepsilon$. In other words, we have: $\|\nabla f(\boldsymbol{w}_k)\| \geq \varepsilon$ for all $k < (\frac{\frac{L_1 M}{2}+\frac{L_0}{4}}{2(1+\sqrt{2})(\log(\frac{L_1 M}{2L_0}+\frac{1}{4})+1)})^2\frac{(\frac{f_2(y_0)-\min_y f_2(y)}{\varepsilon}-\frac{3}{2})^2}{\varepsilon^2}$. Recall that $f(\boldsymbol{w}_1) - \min_{\boldsymbol{w}} f(\boldsymbol{w}) = 2(f_2(y_0) - \min_x f_2(x))$, the proof is completed.

# D  PROOF OF THEOREM 4.1

This appendix provides the formal proof of Theorem 4.1, which is organized as follows. In Section D.1, we first introduce notations that are used in the proof. In Section D.2, We restate Theorem 4.1 with constants specified. In Section D.3, we then make preparations by proving auxiliary lemmas. Finally, in Section D.4, we prove Theorem 4.1.

## D.1  NOTATIONS

Here we provide a complete list of notations used in the appendix for a clear reference.

- We use $(k_1, i_1) \leq (<)(k_2, i_2)$ for $\forall k_1, k_2 \in \mathbb{N}^+$ and $i_1, i_2 \in \{0, \cdots, n-1\}$, if either $k_1 < k_2$ or $k_1 = k_2$ and $i_1 \leq (<)i_2$

- We define function $g(x) : [0,1] \to \mathbb{R}^{/-}$ as

$$g(\beta_2) \triangleq \max \left\{ \frac{1}{\sqrt{\beta_2^{n-1}}} - 1, 1 - \frac{1}{\sqrt{\beta_2^{n-1} + 8n \frac{1-\beta_2^{n-1}}{\beta_2^n}}}, 1 - \sqrt{\beta_2}, \sqrt{\frac{\beta_2}{\left(1 - (1-\beta_2)\frac{2n}{\beta_2^n}\right)} - 1} \right\}.$$

- We define constants $\{C_i\}_{i=1}^{10}$ as follows:

$$C_1 \triangleq \frac{(1-\beta_1)^2}{1-\beta_2} \frac{1}{1 - \frac{\beta_1^2}{\beta_2}} + 1,$$

$$C_2 \triangleq nC_1 + \frac{\beta_1}{1-\beta_1} C_1 \left(1 + \sqrt{2}\right),$$

$$C_3 \triangleq C_1 \left( n(L_0 + L_1\sqrt{D_0}) + 2\sqrt{2}(L_0 + L_1\sqrt{D_0}) \frac{\sqrt{1-\beta_2}}{1-\sqrt{\beta_2}} \frac{\sqrt{\beta_2}}{1-\sqrt{\beta_2}} + 8\sqrt{2n}L_0 \frac{1}{1-\beta_2^n} \right),$$

$$C_4 \triangleq 4L_1 C_1 \sqrt{D_1} \frac{\sqrt{1-\beta_2}}{1-\sqrt{\beta_2}}$$

$$C_5 \triangleq n^2(1 + n\sqrt{d}C_1\eta_1 L_1\sqrt{n}\sqrt{D_1}) \left( C_4 + \frac{dC_4\sqrt{D_1}}{1-\sqrt{\beta_2^n}} \right),$$

$$C_6 \triangleq \left( dC_3 + \frac{C_4 n\sqrt{D_1}}{1-\sqrt{\beta_2^n}} \right) \eta_1^2,$$

$$C_7 \triangleq 3n \left( C_4 + \frac{dC_4}{1-\sqrt{\beta_2^n}} \right) \left( nL_0 + L_1\sqrt{n}\sqrt{D_0} \right) n^2\sqrt{d}C_1\eta_1^3 + \left( dC_3 + \frac{C_2 C_4 n\sqrt{D_1}}{1-\sqrt{\beta_2^n}} \right) \eta_1^2,$$

$$C_8 \triangleq \sqrt{\frac{2n^2}{\beta_2^n}} L_1\sqrt{D_1}n\sqrt{n} + dg(\beta_2)\left(n-1+\frac{1+\beta_1}{1-\beta_1}\right) \frac{\sqrt{2}n}{\beta_2^{\frac{n}{2}}} L_1 C_1 \sqrt{D_1} \left(1 + \frac{1}{1-\beta_2^n}\right)(n + n^{\frac{5}{2}}\sqrt{d}C_1\eta_1 L_1\sqrt{D_1}) + 2\frac{\beta_1}{(1-\beta_1)\eta_1}\sqrt{d}C_1,$$

$$C_9 \triangleq \sqrt{\frac{2n^2}{\beta_2^n}} d(n^2 L_0 + n\sqrt{n}L_1\sqrt{D_0})C_1\eta_1^2 + g(\beta_2)\left(n-1+\frac{1+\beta_1}{1-\beta_1}\right) \frac{\sqrt{2}n}{\beta_2^{\frac{n}{2}}} \left(n + \frac{2\sqrt{2}\beta_1}{1-\beta_1}\right) C_1(L_0 + L_1\sqrt{D_0})d\sqrt{d}\eta_1^2,$$

$$C_{10} \triangleq 3dg(\beta_2)\left(n-1+\frac{1+\beta_1}{1-\beta_1}\right) \frac{\sqrt{2}n}{\beta_2^{\frac{n}{2}}} L_1 C_1\sqrt{D_1} \left(1 + \frac{1}{1-\beta_2^n}\right) n \left(nL_0 + L_1\sqrt{n}\sqrt{D_0}\right) n\sqrt{d}C_1\eta_1^3 + C_9,$$

$$C_{11} \triangleq (\frac{1}{2} + C_2)C_5 + C_8 + \frac{3L_1\sqrt{n}\sqrt{D_1}C_2^2 d}{2},$$

$$C_{12} \triangleq (\frac{1}{2} + C_2)C_6 + C_9 + \frac{nL_0 + L_1\sqrt{n}\sqrt{D_0}}{2} 3C_2^2 d\eta_1^2,$$

$$C_{13} \triangleq (\frac{1}{2} + C_2)C_7 + C_{10} + \frac{nL_0 + L_1\sqrt{n}\sqrt{D_0}}{2} 3C_2^2 d\eta_1^2.$$

$$(9)$$

### D.2 RESTATE THEOREM 4.1

Here we restate Theorem 4.1 with constants specified.

**Theorem D.1** (Theorem 4.1, restated). *Consider Adam defined as Alg. (1) with diminishing learning rate $\eta_k \equiv \frac{\eta_1}{\sqrt{k}}$. Let Assumptions 3.1 and 3.2 hold. Suppose the hyperparamters satisfy: $\gamma < \beta_2 < 1$ and $0 \leq \beta_1^2 < \beta_2$, where $\gamma$ is defined as the solution of $\sqrt{d}g(x)\frac{n}{x^{\frac{n}{2}}} = \frac{1}{2(4+\sqrt{2})\sqrt{D_1}\left(n-1+\frac{1+\beta_1}{1-\beta_1}\right)}$ with respect to $x$. Then, either*

$$\min_{k\in[1,T]} \|\nabla f(\boldsymbol{w}_{k,0})\| \leq 2\sqrt{d}(2\sqrt{2}+1)\sqrt{D_0}g(\beta_2)\left(n-1+\frac{1+\beta_1}{1-\beta_1}\right)\sqrt{\frac{2n}{\beta_2^n}},$$

*or*

$$\min_{k\in[1,T]} \left\{ \frac{\|\nabla f(\boldsymbol{w}_{k,0})\|}{\sqrt{D_1}}, \frac{\|\nabla f(\boldsymbol{w}_{k,0})\|^2}{\sqrt{D_0} + \xi} \right\} \leq (4(2\sqrt{2}+1))\frac{f(\boldsymbol{w}_{1,0}) - \min_{\boldsymbol{w}} f(\boldsymbol{w})}{\eta_1\sqrt{T}}$$

$$+ 4(2\sqrt{2}+1)\left(C_{12} + \frac{\sqrt{D_0}+\xi}{4\sqrt{D_1}}C_{11}\eta_1^2\right)\frac{\ln T}{\eta_1\sqrt{T}} + 4(2\sqrt{2}+1)\frac{C_{13} + \frac{\sqrt{D_0}+\xi}{4\sqrt{D_1}}C_{11}}{\eta_1\sqrt{T}}.$$

### D.3 AUXILIARY LEMMAS

In this section, we introduce auxiliary lemmas that will be latter used. In the remaining proof of this paper, we assume **without the loss of generality that $\eta_1$ is small enough**, such that the following requirements are fulfilled: ($C_1$ and $C_2$ are defined in Eq. (9)).

- $2C_2\sqrt{d}\eta_1 \leq \frac{1}{L_1}$. This will latter ensure that we can directly apply the definition of $(L_0, L_1)$-smooth condition (Assumption 3.1) to parameter sequence $\{\boldsymbol{w}_{k,i}\}_{k,i}$;

- $\frac{1}{4(2\sqrt{2}+1)} \geq \sqrt{D_1}C_{11}\eta_1$. This will latter ensure the second-order term is smaller than the first-order term at the end of the proof.

The proof can be easily extended to general cases by selecting large enough $K$ and using the epoch $K$ as a new start point and derive the results after epoch $K$, because the epochs before epoch $K$ can be uniformly bounded due to $\eta_k$ decaying and $K$ finite, and we then derive the desired result for all epochs.

Without the loss of generality, we also take the following initialization: $\boldsymbol{w}_{1,0} = \boldsymbol{w}_0$, $\boldsymbol{m}_{1,-1} = \nabla f_{\tau_{1,-1}}(\boldsymbol{w}_0)$ where $\tau_{1,-1}$ can be any integer in $[0, n-1]$, and $\boldsymbol{\nu}_{l,1,-1} = \max_j\{\partial_l f_j(\boldsymbol{w}_0)^2\}$ $\forall l$ where the maximum is taken component-wisely. We take the initialization to have a more concise proof, while the proof can be easily extended to all the initialization as the information of the initialization in the exponentially decayed average of Adam (both in $\boldsymbol{m}_{k,i}$ and $\boldsymbol{\nu}_{k,i}$) decays rapidly with $k$ increasing.

The following lemma shows that $f$ is also $(L_0, L_1)$-smooth under Assumptions 3.1 and 3.2 (while the $L_0$ and $L_1$ are different from those of $f_i$).

**Lemma D.2.** *With Assumptions 3.1 and 3.2, $f$ satisfies $(nL_0 + L_1\sqrt{n}\sqrt{D_0}, L_1\sqrt{n}\sqrt{D_1})$-smooth condition.*

*Proof.* $\forall \boldsymbol{w}_1, \boldsymbol{w}_2 \in \mathbb{R}^d$ satisfying $\|\boldsymbol{w}_1 - \boldsymbol{w}_2\| \leq \frac{1}{L_1}$,

$$\|\nabla f(\boldsymbol{w}_1) - \nabla f(\boldsymbol{w}_2)\| \leq \sum_{i=0}^{n-1}\|\nabla f_i(\boldsymbol{w}_1) - \nabla f_i(\boldsymbol{w}_2)\| \leq \sum_{i=0}^{n-1}(L_0 + L_1\|\nabla f_i(\boldsymbol{w}_1)\|)\|\boldsymbol{w}_1 - \boldsymbol{w}_2\|$$

$$\leq \left(nL_0 + L_1\sqrt{n}\sqrt{\sum_{i=0}^{n-1}\|\nabla f_i(\boldsymbol{w}_1)\|^2}\right)\|\boldsymbol{w}_1 - \boldsymbol{w}_2\| \leq (nL_0 + L_1\sqrt{n}\sqrt{D_0 + D_1\|\nabla f(\boldsymbol{w}_1)\|^2})\|\boldsymbol{w}_1 - \boldsymbol{w}_2\|$$

$$\leq (nL_0 + L_1\sqrt{n}(\sqrt{D_0} + \sqrt{D_1}\|\nabla f(\boldsymbol{w}_1)\|))\|\boldsymbol{w}_1 - \boldsymbol{w}_2\| \leq (nL_0 + L_1\sqrt{n}\sqrt{D_0} + L_1\sqrt{n}\sqrt{D_1}\|\nabla f(\boldsymbol{w}_1)\|)\|\boldsymbol{w}_1 - \boldsymbol{w}_2\|.$$

The proof is completed. $\square$

The following lemma bounds the update norm of Adam.

**Lemma D.3** (Bounded Update). *If $\beta_1 < \sqrt{\beta_2}$, we have $\forall k \in \mathbb{N}^+$, $i \in \{0, \cdots, n-1\}$,*

$$\frac{|\boldsymbol{m}_{l,k,i}|}{\sqrt{\boldsymbol{\nu}_{l,k,i}} + \xi} \leq C_1,$$

*where $C_1$ is defined in Eq. (9).*

*Furthermore, we have $|\boldsymbol{w}_{l,k,i+1} - \boldsymbol{w}_{l,k,i}| \leq C_1\eta_k$, and thus $\|\boldsymbol{w}_{k,i+1} - \boldsymbol{w}_{k,i}\| \leq C_1\eta_k\sqrt{d}$.*

*Proof.* By the definition of $\boldsymbol{m}_{k,i}$, we have

$$
(\boldsymbol{m}_{l,k,i})^2
$$

$$
= \left( (1-\beta_1) \sum_{j=0}^{i} \beta_1^{(k-1)n+i-((k-1)n+j)} \partial_l f_{\tau_{k,j}}(\boldsymbol{w}_{k,j}) \right.
$$

$$
\left. + (1-\beta_1) \sum_{m=1}^{k-1} \sum_{j=0}^{n-1} \beta_1^{(k-1)n+i-((m-1)n+j)} \partial_l f_{\tau_{m,j}}(\boldsymbol{w}_{m,j}) + \beta_1^{(k-1)n+i+1} \partial_l f_{\tau_{1,-1}}(\boldsymbol{w}_{1,0}) \right)^2
$$

$$
\leq \left( (1-\beta_1) \sum_{j=0}^{i} \beta_1^{(k-1)n+i-((k-1)n+j)} |\partial_l f_{\tau_{k,j}}(\boldsymbol{w}_{k,j})| \right.
$$

$$
\left. + (1-\beta_1) \sum_{m=1}^{k-1} \sum_{j=0}^{n-1} \beta_1^{(k-1)n+i-((m-1)n+j)} |\partial_l f_{\tau_{m,j}}(\boldsymbol{w}_{m,j})| + \beta_1^{(k-1)n+i+1} \max_{s\in[n]} |\partial_l f_s(\boldsymbol{w}_{1,0})| \right)^2
$$

$$
\stackrel{(\star)}{\leq} \left( (1-\beta_2) \sum_{j=0}^{i} \beta_2^{(k-1)n+i-((k-1)n+j)} |\partial_l f_{\tau_{k,j}}(\boldsymbol{w}_{k,j})|^2 \right.
$$

$$
\left. + (1-\beta_2) \sum_{m=1}^{k-1} \sum_{j=0}^{n-1} \beta_2^{(k-1)n+i-((m-1)n+j)} |\partial_l f_{\tau_{m,j}}(\boldsymbol{w}_{m,j})|^2 + \beta_2^{(k-1)n+i+1} \max_{s\in[n]} |\partial_l f_s(\boldsymbol{w}_{1,0})|^2 \right)
$$

$$
\cdot \left( \frac{(1-\beta_1)^2}{1-\beta_2} \sum_{j=0}^{(k-1)n+i} \left( \frac{\beta_1^2}{\beta_2} \right)^j + \left( \frac{\beta_1^2}{\beta_2} \right)^{(k-1)n+i+1} \right)
$$

$$
\stackrel{(*)}{=} \left( \frac{(1-\beta_1)^2}{1-\beta_2} \sum_{j=0}^{(k-1)n+i} \left( \frac{\beta_1^2}{\beta_2} \right)^j + \left( \frac{\beta_1^2}{\beta_2} \right)^{(k-1)n+i+1} \right) \boldsymbol{\nu}_{l,k,i}
$$

$$
\leq \left( \frac{(1-\beta_1)^2}{1-\beta_2} \frac{1}{1-\frac{\beta_1^2}{\beta_2}} + 1 \right) \boldsymbol{\nu}_{l,k,i} = C_1 \boldsymbol{\nu}_{l,k,i},
$$

where Eq. $(\star)$ is due to the Cauchy-Schwartz's Inequality, and Eq. $(*)$ is due to the definition of $\boldsymbol{\nu}_{l,1,-1}$. We complete the proof of the first claim. The second claim then follows directly from the update rule

$$
\boldsymbol{w}_{l,k,i+1} - \boldsymbol{w}_{l,k,i} = \eta_k \frac{\boldsymbol{m}_{l,k,i}}{\sqrt{\boldsymbol{\nu}_{l,k,i}} + \xi}.
$$

The proof is completed. $\qquad \square$

Define $\boldsymbol{u}_k \triangleq \frac{\boldsymbol{w}_{k,0} - \beta_1 \boldsymbol{w}_{k,-1}}{1-\beta_1}$ (with $\boldsymbol{w}_{1,-1} \triangleq \boldsymbol{w}_{1,0}$), and let $\boldsymbol{u}_{l,k}$ be the $i$-th component of $\boldsymbol{u}_k$, $\forall k \in \mathbb{N}^+$, $l \in [d]$. The following lemma bounds the distance between $\boldsymbol{u}_{l,k}$ and $\boldsymbol{w}_{l,k,0}$ and the distance between $\boldsymbol{u}_{l,k+1}$ and $\boldsymbol{u}_{l,k}$.

**Lemma D.4.** $\forall k \geq 1$,

$$
|\boldsymbol{u}_{l,k} - \boldsymbol{w}_{l,k,0}| \leq C_2 \eta_k, \tag{10}
$$

$$
|\boldsymbol{u}_{l,k+1} - \boldsymbol{u}_{l,k}| \leq C_2 \eta_k, \tag{11}
$$

*where $C_2$ is defined in Eq. (9).*

*Proof.* By Lemma D.3, we immediately have $\forall l \in [d]$, $|\boldsymbol{u}_{l,k} - \boldsymbol{w}_{l,k,0}|$ is bounded as

$$|\boldsymbol{u}_{l,k} - \boldsymbol{w}_{l,k,0}| = \left| \frac{\boldsymbol{w}_{l,k,0} - \beta_1 \boldsymbol{w}_{l,k,-1}}{1 - \beta_1} - \boldsymbol{w}_{l,k,0} \right|$$

$$= \frac{\beta_1}{1-\beta_1} |\boldsymbol{w}_{l,k,0} - \boldsymbol{w}_{l,k,-1}| \leq \frac{\beta_1}{1-\beta_1} C_1 \eta_1 \frac{1}{\sqrt{k-1}} \leq \frac{\sqrt{2}\beta_1}{1-\beta_1} C_1 \eta_1 \frac{1}{\sqrt{k}} \leq \frac{\sqrt{2}\beta_1}{1-\beta_1} C_1 \eta_k \leq C_2 \eta_k,$$

and

$$|\boldsymbol{u}_{l,k+1} - \boldsymbol{u}_{l,k}|$$

$$= \left| \frac{\boldsymbol{w}_{l,k+1,0} - \beta_1 \boldsymbol{w}_{l,k+1,-1}}{1 - \beta_1} - \frac{\boldsymbol{w}_{l,k,0} - \beta_1 \boldsymbol{w}_{l,k,-1}}{1 - \beta_1} \right|$$

$$= \left| (\boldsymbol{w}_{l,k+1,0} - \boldsymbol{w}_{l,k,0}) + \frac{\beta_1}{1-\beta_1} (\boldsymbol{w}_{l,k+1,0} - \boldsymbol{w}_{l,k+1,-1}) - \frac{\beta_1}{1-\beta_1} (\boldsymbol{w}_{l,k,0} - \boldsymbol{w}_{l,k,-1}) \right|$$

$$\leq \left| (\boldsymbol{w}_{l,k+1,0} - \boldsymbol{w}_{l,k,0}) + \frac{\beta_1}{1-\beta_1} (\boldsymbol{w}_{l,k+1,0} - \boldsymbol{w}_{l,k+1,-1}) - \frac{\beta_1}{1-\beta_1} (\boldsymbol{w}_{l,k,0} - \boldsymbol{w}_{l,k,-1}) \right|$$

$$\leq n C_1 \eta_1 \frac{1}{\sqrt{k}} + \frac{\beta_1}{1-\beta_1} C_1 \eta_1 \left( \frac{1}{\sqrt{k}} + \frac{\sqrt{2}}{\sqrt{k}} \right) = C_2 \eta_1 \frac{1}{\sqrt{k}} = C_2 \eta_k.$$

$\square$

In the following lemma, we bound the change of the gradient within one epoch.

**Lemma D.5.** $\forall k \in \mathbb{N}^+, i \in \{0, \cdots, n-1\}$,

$$\|\nabla f(\boldsymbol{w}_{k,i})\| \leq (1 + n\sqrt{d} C_1 \eta_1 L_1 \sqrt{n} \sqrt{D_1}) \|\nabla f(\boldsymbol{w}_{k,0})\| + \left( n L_0 + L_1 \sqrt{n} \sqrt{D_0} \right) n\sqrt{d} C_1 \eta_k,$$

*where $C_1$ is defined in Eq. (9).*

*Proof.* By Assumption 3.1 and Lemma D.2, we have

$$\|\nabla f(\boldsymbol{w}_{k,i})\| \leq \|\nabla f(\boldsymbol{w}_{k,0})\| + \left( n L_0 + L_1 \sqrt{n} \sqrt{D_0} + L_1 \sqrt{n} \sqrt{D_1} \|\nabla f(\boldsymbol{w}_{k,0})\| \right) \|\boldsymbol{w}_{k,i} - \boldsymbol{w}_{k,0}\|$$

$$\leq \|\nabla f(\boldsymbol{w}_{k,0})\| + \left( n L_0 + L_1 \sqrt{n} \sqrt{D_0} + L_1 \sqrt{n} \sqrt{D_1} \|\nabla f(\boldsymbol{w}_{k,0})\| \right) i\sqrt{d} C_1 \eta_k$$

$$\leq (1 + n\sqrt{d} C_1 \eta_1 L_1 \sqrt{n} \sqrt{D_1}) \|\nabla f(\boldsymbol{w}_{k,0})\| + \left( n L_0 + L_1 \sqrt{n} \sqrt{D_0} \right) n\sqrt{d} C_1 \eta_k.$$

The proof is completed. $\square$

We further need a descent lemma assuming $(L_0, L_1)$-smooth condition similar to the case assuming $L$ smoothness. Specifically, for a function $h$ satisfying $L$-smooth condition and two points $\boldsymbol{w}$ and $\boldsymbol{v}$, by Taylor's expansion, we have

$$h(\boldsymbol{w}) \leq h(\boldsymbol{v}) + \langle \nabla h(\boldsymbol{v}), \boldsymbol{w} - \boldsymbol{v} \rangle + \frac{L}{2} \|\boldsymbol{w} - \boldsymbol{v}\|^2.$$

This is called "Descent Lemma" by existing literature Sra (2014), as it guarantees that the loss decreases with proper parameter update. Paralleling to the above inequality, we establish the following descent lemma under the $(L_0, L_1)$-smooth condition.

**Lemma D.6.** *Assume that function $h : \mathcal{X} \to \mathbb{R}$ satisfies the $(L_0, L_1)$-smooth condition, i.e., $\forall \boldsymbol{w}, \boldsymbol{v} \in \mathcal{X}$ satisfying $\|\boldsymbol{w} - \boldsymbol{v}\| \leq \frac{1}{L_1}$,*

$$\|\nabla h(\boldsymbol{w}) - \nabla h(\boldsymbol{v})\| \leq (L_0 + L_1 \|\nabla h(\boldsymbol{v})\|) \|\boldsymbol{w} - \boldsymbol{v}\|.$$

*Then, for any three points $\boldsymbol{u}, \boldsymbol{w}, \boldsymbol{v} \in \mathcal{X}$ satisfying $\|\boldsymbol{w} - \boldsymbol{u}\| \leq \frac{1}{L_1}$ and $\|\boldsymbol{v} - \boldsymbol{u}\| \leq \frac{1}{L_1}$, we have*

$$h(\boldsymbol{w}) \leq h(\boldsymbol{v}) + \langle \nabla h(\boldsymbol{u}), \boldsymbol{w} - \boldsymbol{v} \rangle + \frac{1}{2}(L_0 + L_1 \|\nabla h(\boldsymbol{u})\|)(\|\boldsymbol{v} - \boldsymbol{u}\| + \|\boldsymbol{w} - \boldsymbol{u}\|) \|\boldsymbol{w} - \boldsymbol{v}\|.$$

*Proof.* By the Fundamental Theorem of Calculus, we have

$$
\begin{aligned}
h(\boldsymbol{w}) =& h(\boldsymbol{v}) + \int_0^1 \langle \nabla h(\boldsymbol{v} + a(\boldsymbol{w} - \boldsymbol{v})), \boldsymbol{w} - \boldsymbol{v}\rangle \mathrm{d}a \\
=& h(\boldsymbol{v}) + \langle \nabla h(\boldsymbol{u}), \boldsymbol{w} - \boldsymbol{v}\rangle + \int_0^1 \langle \nabla h(\boldsymbol{v} + a(\boldsymbol{w} - \boldsymbol{v})) - \nabla h(\boldsymbol{u}), \boldsymbol{w} - \boldsymbol{v}\rangle \mathrm{d}a \\
\leq& h(\boldsymbol{v}) + \langle \nabla h(\boldsymbol{u}), \boldsymbol{w} - \boldsymbol{v}\rangle + \int_0^1 \|\nabla h(\boldsymbol{v} + a(\boldsymbol{w} - \boldsymbol{v})) - \nabla h(\boldsymbol{u})\|\|\boldsymbol{w} - \boldsymbol{v}\| \mathrm{d}a \\
\overset{(\star)}{\leq}& h(\boldsymbol{v}) + \langle \nabla h(\boldsymbol{u}), \boldsymbol{w} - \boldsymbol{v}\rangle + \int_0^1 (L_0 + L_1\|\nabla h(\boldsymbol{u})\|)\|\boldsymbol{v} + a(\boldsymbol{w} - \boldsymbol{v}) - \boldsymbol{u}\|\|\boldsymbol{w} - \boldsymbol{v}\| \mathrm{d}a \\
\leq& h(\boldsymbol{v}) + \langle \nabla h(\boldsymbol{u}), \boldsymbol{w} - \boldsymbol{v}\rangle + \int_0^1 (L_0 + L_1\|\nabla h(\boldsymbol{u})\|)((1 - a)\|\boldsymbol{v} - \boldsymbol{u}\| + a\|\boldsymbol{w} - \boldsymbol{u}\|)\|\boldsymbol{w} - \boldsymbol{v}\| \mathrm{d}a \\
\leq& h(\boldsymbol{v}) + \langle \nabla h(\boldsymbol{u}), \boldsymbol{w} - \boldsymbol{v}\rangle + \frac{1}{2}(L_0 + L_1\|\nabla h(\boldsymbol{u})\|)(\|\boldsymbol{v} - \boldsymbol{u}\| + \|\boldsymbol{w} - \boldsymbol{u}\|)\|\boldsymbol{w} - \boldsymbol{v}\|,
\end{aligned}
$$

where Inequality $(\star)$ is due to

$$
\|\boldsymbol{v} + a(\boldsymbol{w} - \boldsymbol{v}) - \boldsymbol{u}\| = \|(1 - a)(\boldsymbol{v} - \boldsymbol{u}) + a(\boldsymbol{w} - \boldsymbol{u})\| \leq (1 - a)\|\boldsymbol{v} - \boldsymbol{u}\| + a\|\boldsymbol{w} - \boldsymbol{u}\| \leq \frac{1}{L_1}.
$$

Thus the definition of $(L_0, L_1)$-smooth condition can be applied and the proof is completed. $\quad\square$

Based on Lemma D.3, we bound the momentum using the gradient of the current step plus some error terms.

**Lemma D.7** (Estimation of the norm of the momentum). *We have for all $l \in [d], k \in \mathbb{Z}^+, i \in [n]$,*

$$
\begin{aligned}
|\boldsymbol{m}_{l,k,i}| \leq& \max_{i' \in [n]} |\partial_l f_{i'}(\boldsymbol{w}_{k,0})| + \left(n + \frac{2\sqrt{2}\beta_1}{1 - \beta_1}\right) C_1(L_0 + L_1\sqrt{D_0})\sqrt{d}\eta_k + L_1 C_1 \sqrt{D_1}\eta_k \sum_{j=0}^{i-1} \|\nabla f(\boldsymbol{w}_{k,j})\| \\
& + L_1 C_1 \sqrt{D_1} \sum_{t=1}^{k-1} \eta_{k-t} \sum_{j=0}^{n-1} \beta_1^{tn+i-j} \|\nabla f(\boldsymbol{w}_{k-t,j})\|,
\end{aligned}
$$

*where $C_1$ is defined in Eq. (9). Similarly, $l \in [d], k \in \mathbb{Z}^+/\{1\}$,*

$$
|\boldsymbol{m}_{l,k-1,n-1}| \leq \max_{i' \in [n]} |\partial_l f_{i'}(\boldsymbol{w}_{k,0})| + \sum_{t=1}^{k-1}\sum_{j=0}^{n-1} \beta_1^{tn-1-j} C_1 \eta_{k-t}\sqrt{d}L_1\sqrt{D_1}\|\nabla f(\boldsymbol{w}_{k-t,j})\| + \frac{2\sqrt{2}(L_0 + L_1\sqrt{D_0})C_1\sqrt{d}\eta_k}{1 - \beta_1}.
$$

*Proof.* To begin with, for any $t \in [k-1]$ and any $j \in [0, n-1]$, we have the following estimation for $\partial_l f_i(\boldsymbol{w}_{k-t,j})$:

$$|\partial_l f_i(\boldsymbol{w}_{k-t,j})|$$

$$\leq |\partial_l f_i(\boldsymbol{w}_{k,0})| + \sum_{p=j}^{n-1} |\partial_l f_i(\boldsymbol{w}_{k-t,p}) - \partial_l f_i(\boldsymbol{w}_{k-t,p+1})| + \sum_{r=1}^{t-1}\sum_{p=0}^{n-1} |\partial_l f_i(\boldsymbol{w}_{k-r,p}) - \partial_l f_i(\boldsymbol{w}_{k-r,p+1})|$$

$$\overset{(\star)}{\leq} |\partial_l f_i(\boldsymbol{w}_{k,0})| + \sum_{p=j}^{n-1} (L_0 + L_1\|\nabla f_i(\boldsymbol{w}_{k-t,p})\|)\|\boldsymbol{w}_{k-t,p} - \boldsymbol{w}_{k-t,p+1}\|$$

$$+ \sum_{r=1}^{t-1}\sum_{p=0}^{n-1} (L_0 + L_1\|\nabla f_i(\boldsymbol{w}_{k-r,p})\|)\|\boldsymbol{w}_{k-r,p} - \boldsymbol{w}_{k-r,p+1}\|$$

$$\leq |\partial_l f_i(\boldsymbol{w}_{k,0})| + \sum_{p=j}^{n-1} (L_0 + L_1\|\nabla f_i(\boldsymbol{w}_{k-t,p})\|)C_1\eta_{k-t}\sqrt{d} + \sum_{r=1}^{t-1}\sum_{p=0}^{n-1} (L_0 + L_1\|\nabla f_i(\boldsymbol{w}_{k-r,p})\|)C_1\eta_{k-r}\sqrt{d}$$

$$\leq |\partial_l f_i(\boldsymbol{w}_{k,0})| + \sum_{p=j}^{n-1} \left(L_0 + L_1\sqrt{\sum_{i'\in[n]}\|\nabla f_{i'}(\boldsymbol{w}_{k-t,p})\|^2}\right) C_1\eta_{k-t}\sqrt{d}$$

$$+ \sum_{r=1}^{t-1}\sum_{p=0}^{n-1} \left(L_0 + L_1\sqrt{\sum_{i'\in[n]}\|\nabla f_{i'}(\boldsymbol{w}_{k-r,p})\|^2}\right) C_1\eta_{k-r}\sqrt{d},$$

where Inequality $(\star)$ is due to $(L_0, L_1)$-smooth condition. By Assumption 3.2, the RHS of the above inequality can be bounded as

$$|\partial_l f_i(\boldsymbol{w}_{k,0})| + \sum_{p=j}^{n-1} \left(L_0 + L_1\sqrt{D_1}\|\nabla f(\boldsymbol{w}_{k-t,p})\| + L_1\sqrt{D_0}\right) C_1\eta_{k-t}\sqrt{d}$$

$$+ \sum_{r=1}^{t-1}\sum_{p=0}^{n-1} \left(L_0 + L_1\sqrt{D_1}\|\nabla f(\boldsymbol{w}_{k-r,p})\| + L_1\sqrt{D_0}\right) C_1\eta_{k-r}\sqrt{d}$$

$$\overset{(*)}{\leq} |\partial_l f_i(\boldsymbol{w}_{k,0})| + \sum_{p=j}^{n-1} L_1\sqrt{D_1}\|\nabla f(\boldsymbol{w}_{k-t,p})C_1\eta_{k-t}\sqrt{d} + \sum_{r=1}^{t-1}\sum_{p=0}^{n-1} L_1\sqrt{D_1}\|\nabla f(\boldsymbol{w}_{k-r,p})\|C_1\eta_{k-r}\sqrt{d}$$

$$+ 2(L_0 + L_1\sqrt{D_0})C_1\sqrt{d}\eta_{k-1}(tn-j)$$

$$\leq |\partial_l f_i(\boldsymbol{w}_{k,0})| + \sum_{p=j}^{n-1} L_1\sqrt{D_1}\|\nabla f(\boldsymbol{w}_{k-t,p})C_1\eta_{k-t}\sqrt{d} + \sum_{r=1}^{t-1}\sum_{p=0}^{n-1} L_1\sqrt{D_1}\|\nabla f(\boldsymbol{w}_{k-r,p})\|C_1\eta_{k-r}\sqrt{d}$$

$$+ 2\sqrt{2}(L_0 + L_1\sqrt{D_0})C_1\sqrt{d}\eta_k(tn-j).$$

where Inequality $(*)$ is due to $\forall a, b \in \mathbb{N}^+, a > b, \sum_{i=0}^{b} \frac{1}{\sqrt{a-i}} \leq 2\frac{b+1}{a}$. Similarly, we have that for any $j \in [0, n-1]$,

$$|\partial_l f_i(\boldsymbol{w}_{k,j})| \leq |\partial_l f_i(\boldsymbol{w}_{k,0})| + \sum_{p=0}^{j-1} |\partial_l f_i(\boldsymbol{w}_{k,p+1}) - \partial_l f_i(\boldsymbol{w}_{k,p})|$$

$$\leq |\partial_l f_i(\boldsymbol{w}_{k,0})| + \sum_{p=0}^{j-1} \left(L_0 + L_1\sqrt{D_1}\|\nabla f(\boldsymbol{w}_{k,p})\| + L_1\sqrt{D_0}\right) C_1\eta_k\sqrt{d}$$

$$= |\partial_l f_i(\boldsymbol{w}_{k,0})| + \sum_{p=0}^{j-1} L_1\sqrt{D_1}\|\nabla f(\boldsymbol{w}_{k,p})\|C_1\eta_k\sqrt{d} + j(L_0 + L_1\sqrt{D_0})C_1\sqrt{d}\eta_k.$$

Therefore, the norm of $\boldsymbol{m}_{l,k,i}$ can be bounded as

$$|\boldsymbol{m}_{l,k,i}|$$

$$\leq (1-\beta_1)\sum_{j=0}^{i}\beta_1^{(k-1)n+i-((k-1)n+j)}|\partial_l f_{\tau_{k,j}}(\boldsymbol{w}_{k,j})| + (1-\beta_1)\sum_{t=1}^{k-1}\sum_{j=0}^{n-1}\beta_1^{tn+i-j}|\partial_l f_{\tau_{k-t,j}}(\boldsymbol{w}_{k-t,j})|$$

$$+ \beta_1^{(k-1)n+i+1}|\partial_l f_{\tau_{1,0}}(\boldsymbol{w}_{1,0})|$$

$$\leq (1-\beta_1)\sum_{j=0}^{i}\beta_1^{(k-1)n+i-((k-1)n+j)}|\partial_l f_{\tau_{k,j}}(\boldsymbol{w}_{k,0})| + (1-\beta_1)\sum_{t=1}^{k-1}\sum_{j=0}^{n-1}\beta_1^{tn+i-j}|\partial_l f_{\tau_{k-t,j}}(\boldsymbol{w}_{k,0})|$$

$$+ \beta_1^{(k-1)n+i+1}|\partial_l f_{\tau_{1,0}}(\boldsymbol{w}_{k,0})|$$

$$+ (1-\beta_1)\sum_{j=0}^{i}\beta_1^{(k-1)n+i-((k-1)n+j)}\left(\sum_{p=0}^{j-1}C_1\eta_k\sqrt{d}L_1\sqrt{D_1}\|\nabla f(\boldsymbol{w}_{k,p})\| + (L_0+L_1\sqrt{D_0})C_1\eta_k\sqrt{d}j\right)$$

$$+ (1-\beta_1)\sum_{t=1}^{k-1}\sum_{j=0}^{n-1}\beta_1^{tn+i-j}\left(\sum_{p=j}^{n-1}C_1\eta_{k-t}\sqrt{d}L_1\sqrt{D_1}\|\nabla f(\boldsymbol{w}_{k-t,p})\|\right.$$

$$+ \sum_{r=1}^{t-1}\sum_{p=0}^{n-1}C_1\eta_{k-r}\sqrt{d}L_1\sqrt{D_1}\|\nabla f(\boldsymbol{w}_{k-r,p})\| + 2\sqrt{2}(L_0+L_1\sqrt{D_0})C_1\sqrt{d}\eta_k(tn-j)\Big)$$

$$+ \beta_1^{(k-1)n+i+1}\left(\sum_{t=1}^{k-1}\sum_{p=0}^{n-1}L_1\sqrt{D_1}\|\nabla f(\boldsymbol{w}_{k-r,p})\|C_1\eta_{k-r}\sqrt{d} + 2\sqrt{2}(L_0+L_1\sqrt{D_0})C_1\sqrt{d}\eta_k(k-1)n\right)$$

$$\overset{(\star)}{\leq} \max_{i\in[n]}|\partial_l f_i(\boldsymbol{w}_{k,0})| + \left(n + \frac{2\sqrt{2}\beta_1}{1-\beta_1}\right)\sqrt{d}C_1(L_0+L_1\sqrt{D_0})\eta_k + L_1C_1\sqrt{D_1}\eta_k\sum_{j=0}^{i-1}\|\nabla f(\boldsymbol{w}_{k,j})\|$$

$$+ L_1C_1\sqrt{D_1}\sum_{t=1}^{k-1}\eta_{k-t}\sum_{j=0}^{n-1}\beta_1^{tn+i-j}\|\nabla f(\boldsymbol{w}_{k-t,j})\|,$$

where Inequality $(\star)$ is due to an exchange in the sum order.

Following the same routine, we have

$$|\boldsymbol{m}_{l,k,-1}|$$

$$\leq (1-\beta_1)\sum_{t=1}^{k-1}\sum_{j=0}^{n-1}\beta_1^{tn-1-j}|\partial_l f_{\tau_{k-t,j}}(\boldsymbol{w}_{k-t,j})| + \beta_1^{(k-1)n}|\partial_l f_{\tau_{1,0}}(\boldsymbol{w}_{1,0})|$$

$$\leq (1-\beta_1)\sum_{t=1}^{k-1}\sum_{j=0}^{n-1}\beta_1^{tn-1-j}|\partial_l f_{\tau_{k-t,j}}(\boldsymbol{w}_{k,0})| + \beta_1^{(k-1)n}|\partial_l f_{\tau_{1,0}}(\boldsymbol{w}_{k,0})|$$

$$+ (1-\beta_1)\sum_{t=1}^{k-1}\sum_{j=0}^{n-1}\beta_1^{tn-1-j}C_1\sqrt{d}\left(\sum_{p=j}^{n-1}L_1\sqrt{D_1}\|\nabla f(\boldsymbol{w}_{k-t,p})\|\eta_{k-t} + \sum_{r=1}^{t-1}\sum_{p=0}^{n-1}L_1\sqrt{D_1}\|\nabla f(\boldsymbol{w}_{k-r,p})\|\eta_{k-r}\right.$$

$$+ 2\sqrt{2}(L_0+L_1\sqrt{D_0})C_1\sqrt{d}\eta_k(tn-j)\Big)$$

$$+ \beta_1^{(k-1)n}\left(\sum_{t=1}^{k-1}\sum_{p=0}^{n-1}L_1\sqrt{D_1}\|\nabla f(\boldsymbol{w}_{k-r,p})\|C_1\eta_{k-r}\sqrt{d} + 2\sqrt{2}(L_0+L_1\sqrt{D_0})C_1\sqrt{d}\eta_k(k-1)n\right)$$

$$\leq \max_{i\in[n]}|\partial_l f_i(\boldsymbol{w}_{k,0})| + \sum_{t=1}^{k-1}\sum_{j=0}^{n-1}\beta_1^{tn-1-j}C_1\eta_{k-t}\sqrt{d}L_1\sqrt{D_1}\|\nabla f(\boldsymbol{w}_{k-t,j})\|$$

$$+ \frac{2\sqrt{2}(L_0+L_1\sqrt{D_0})C_1\sqrt{d}\eta_k}{1-\beta_1}.$$

The proof is completed. □

Similarly, we can upper and lower bound the adaptor $\boldsymbol{\nu}_{k,0}$ by the gradient plus some error terms.

**Lemma D.8** (Estimation of the norm of the adaptor). *We have for all $l \in [d], k \in \mathbb{Z}^+$,*

$$
\begin{aligned}
|\boldsymbol{\nu}_{l,k,0}| \geq & \beta_2^n \frac{1-\beta_2}{1-\beta_2^n} \sum_{i \in [n]} \partial_l f_i(\boldsymbol{w}_{k,0})^2 - \sqrt{\sum_{i \in [n]} |\partial_l f_i(\boldsymbol{w}_{k,0})^2|} \left( 8\sqrt{2n}\eta_k C_1 L_0 \frac{1-\beta_2}{(1-\beta_2^n)^2}\beta_2^n \right. \\
& \left. + 4L_1 C_1 \frac{1-\beta_2}{1-\beta_2^n} \frac{\sqrt{1-\beta_2}}{1-\sqrt{\beta_2}} \left( \sum_{t=1}^{k-1} \beta_2^n \sqrt{\beta_2}^{(r-1)n} \eta_{k-t} \sum_{j=0}^{n-1} (\sqrt{D_1}\|\nabla f(\boldsymbol{w}_{k-t,j})\| + \sqrt{D_0}) \right) \right),
\end{aligned}
$$

*and*

$$
\begin{aligned}
|\boldsymbol{\nu}_{l,k,0}| \leq & 2\max_{i \in [n]} \partial_l f_i(\boldsymbol{w}_{k,0})^2 + 2\left( 2\sqrt{2}\eta_k C_1(L_0 + L_1\sqrt{D_0}) \frac{\sqrt{1-\beta_2}}{1-\sqrt{\beta_2}} \frac{\sqrt{\beta_2}}{1-\sqrt{\beta_2}} \right. \\
& \left. + L_1 C_1 \sqrt{D_1} \sum_{t=1}^{k-1} \eta_{k-t} \frac{\sqrt{1-\beta_2}}{1-\sqrt{\beta_2}} \sum_{j=0}^{n-1} \sqrt{\beta_2}^{(t-1)n}\|\nabla f(\boldsymbol{w}_{k-t,j})\| \right)^2,
\end{aligned}
$$

*where $C_1$ is defined in Eq. (9).*

*Proof.* By the definition of $\boldsymbol{\nu}_{l,k,0}$, we have

$$
\begin{aligned}
& \boldsymbol{\nu}_{l,k,0} \\
= & (1-\beta_2)\partial_l f_{\tau_{k,0}}(\boldsymbol{w}_{k,0})^2 + \sum_{t=1}^{k-1}\sum_{j=0}^{n-1}(1-\beta_2)\beta_2^{tn-j}\partial_l f_{\tau_{k-t,j}}(\boldsymbol{w}_{k-t,j})^2 + \beta_2^{(k-1)n+1}\max_{i \in [n]}\partial_l f_i(\boldsymbol{w}_{1,0})^2 \\
\geq & (1-\beta_2)\partial_l f_{\tau_{k,0}}(\boldsymbol{w}_{k,0})^2 + \sum_{t=1}^{k-1}\sum_{j=0}^{n-1}(1-\beta_2)\beta_2^{tn}\partial_l f_{\tau_{k-t,j}}(\boldsymbol{w}_{k-t,j})^2 + \beta_2^{(k-1)n+1}\frac{1}{n}\sum_{i=1}^{n}\partial_l f_i(\boldsymbol{w}_{1,0})^2 \\
= & (1-\beta_2)\partial_l f_{\tau_{k,0}}(\boldsymbol{w}_{k,0})^2 + \sum_{t=1}^{k-1}\sum_{j=0}^{n-1}(1-\beta_2)\beta_2^{tn}(\partial_l f_{\tau_{k-t,j}}(\boldsymbol{w}_{k,0}) + \partial_l f_{\tau_{k-t,j}}(\boldsymbol{w}_{k-t,j}) - \partial_l f_{\tau_{k-t,j}}(\boldsymbol{w}_{k,0}))^2 \\
& + \beta_2^{(k-1)n+1}\frac{1}{n}\sum_{i=1}^{n}(\partial_l f_i(\boldsymbol{w}_{k,0}) + \partial_l f_i(\boldsymbol{w}_{1,0}) - \partial_l f_i(\boldsymbol{w}_{k,0}))^2 \\
\geq & (1-\beta_2)\partial_l f_{\tau_{k,0}}(\boldsymbol{w}_{k,0})^2 + \sum_{t=1}^{k-1}\sum_{j=0}^{n-1}(1-\beta_2)\beta_2^{tn}\partial_l f_{\tau_{k-t,j}}(\boldsymbol{w}_{k,0})^2 + \beta_2^{(k-1)n+1}\frac{1}{n}\sum_{i=1}^{n}\partial_l f_i(\boldsymbol{w}_{k,0})^2 \\
& - \sum_{t=1}^{k-1}\sum_{j=0}^{n-1}(1-\beta_2)\beta_2^{tn}|\partial_l f_{\tau_{k-t,j}}(\boldsymbol{w}_{k,0})||\partial_l f_{\tau_{k-t,j}}(\boldsymbol{w}_{k,0}) - \partial_l f_{\tau_{k-t,j}}(\boldsymbol{w}_{k-t,j})| \\
& - \beta_2^{(k-1)n+1}\frac{1}{n}\sum_{i=1}^{n}|\partial_l f_i(\boldsymbol{w}_{k,0})||\partial_l f_i(\boldsymbol{w}_{k,0}) - \partial_l f_i(\boldsymbol{w}_{1,0})|
\end{aligned}
$$

Since $f_i$ is $(L_0, L_1)$-smooth, the RHS of the above inequality can be further lower bounded as follows:

$$
\left( \beta_2^n \frac{1 - \beta_2^{(k-1)n}}{1 - \beta_2^n}(1 - \beta_2) + \frac{\beta_2^{(k-1)n+1}}{n} \right) \sum_{i \in [n]} \partial_l f_i(\boldsymbol{w}_{k,0})^2
$$

$$
- \sum_{t=1}^{k-1} \sum_{j=0}^{n-1} (1 - \beta_2)\beta_2^{tn} |\partial_l f_{\tau_{k-t,j}}(\boldsymbol{w}_{k,0})| \left( \sum_{r=1}^{t} \sum_{p=0}^{n-1} L_1 \sqrt{D_1} \|\nabla f(\boldsymbol{w}_{k-r,p})\| C_1 \eta_{k-r} \sqrt{d} + 2\sqrt{2}(L_0 + L_1\sqrt{D_0})C_1\sqrt{d}\eta_k tn \right)
$$

$$
- \beta_2^{(k-1)n+1} \frac{1}{n} \sum_{i=1}^{n} |\partial_l f_i(\boldsymbol{w}_{k,0})| \left( \sum_{r=1}^{k-1} \sum_{p=0}^{n-1} L_1 \sqrt{D_1} \|\nabla f(\boldsymbol{w}_{k-r,p})\| C_1 \eta_{k-r} \sqrt{d} + 2\sqrt{2}(L_0 + L_1\sqrt{D_0})C_1\sqrt{d}\eta_k(k-1)n \right)
$$

$$
\geq \beta_2^n \frac{1 - \beta_2}{1 - \beta_2^n} \sum_{i \in [n]} \partial_l f_i(\boldsymbol{w}_{k,0})^2
$$

$$
- \sum_{t=1}^{k-1} \sum_{j=0}^{n-1} (1 - \beta_2)\beta_2^{tn} |\partial_l f_{\tau_{k-t,j}}(\boldsymbol{w}_{k,0})| \left( \sum_{r=1}^{t} \sum_{p=0}^{n-1} L_1 \sqrt{D_1} \|\nabla f(\boldsymbol{w}_{k-r,p})\| C_1 \eta_{k-r} \sqrt{d} + 2\sqrt{2}(L_0 + L_1\sqrt{D_0})C_1\sqrt{d}\eta_k tn \right)
$$

$$
- \beta_2^{(k-1)n+1} \frac{1}{n} \sum_{i=1}^{n} |\partial_l f_i(\boldsymbol{w}_{k,0})| \left( \sum_{r=1}^{k-1} \sum_{p=0}^{n-1} L_1 \sqrt{D_1} \|\nabla f(\boldsymbol{w}_{k-r,p})\| C_1 \eta_{k-r} \sqrt{d} + 2\sqrt{2}(L_0 + L_1\sqrt{D_0})C_1\sqrt{d}\eta_k(k-1)n \right),
$$

where the last inequality we use $\beta_2^n \frac{1 - \beta_2^{(k-1)n}}{1 - \beta_2^n}(1 - \beta_2) + \frac{\beta_2^{(k-1)n+1}}{n} \geq \beta_2^n \frac{1 - \beta_2}{1 - \beta_2^n}$.

$$
\geq \beta_2^n \frac{1 - \beta_2}{1 - \beta_2^n} \sum_{i \in [n]} \partial_l f_i(\boldsymbol{w}_{k,0})^2 - 8\sqrt{2}\eta_k C_1 L_0 \frac{1 - \beta_2}{(1 - \beta_2^n)^2}\beta_2^n \sum_{i \in [n]} |\partial_l f_i(\boldsymbol{w}_{k,0})|
$$

$$
- 4L_1 C_1 \frac{1 - \beta_2}{1 - \beta_2^n} \sum_{i \in [n]} |\partial_l f_i(\boldsymbol{w}_{k,0})| \left( \sum_{r=1}^{k-1} \beta_2^{rn} \eta_{k-r} \sum_{j=0}^{n-1} \|\nabla f_i(\boldsymbol{w}_{k-r,j})\| \right)
$$

$$
\geq \beta_2^n \frac{1 - \beta_2}{1 - \beta_2^n} \sum_{i \in [n]} \partial_l f_i(\boldsymbol{w}_{k,0})^2 - 8\sqrt{2}\eta_k C_1 L_0 \frac{1 - \beta_2}{(1 - \beta_2^n)^2}\beta_2^n \sum_{i \in [n]} |\partial_l f_i(\boldsymbol{w}_{k,0})|
$$

$$
- 4L_1 C_1 \frac{1 - \beta_2}{1 - \beta_2^n} \|\nabla f_i(\boldsymbol{w}_{k,0})\| \left( \sum_{r=1}^{k-1} \beta_2^{rn} \eta_{k-r} \sum_{j=0}^{n-1} (\sqrt{D_1} \|\nabla f(\boldsymbol{w}_{k-r,j})\| + \sqrt{D_0}) \right)
$$

$$
\geq \beta_2^n \frac{1 - \beta_2}{1 - \beta_2^n} \sum_{i \in [n]} \partial_l f_i(\boldsymbol{w}_{k,0})^2 - 8\sqrt{2n}\eta_k C_1 L_0 \frac{1 - \beta_2}{(1 - \beta_2^n)^2}\beta_2^n \sqrt{\sum_{i \in [n]} |\partial_l f_i(\boldsymbol{w}_{k,0})^2|}
$$

$$
- 4L_1 C_1 \frac{1 - \beta_2}{1 - \beta_2^n} \sqrt{\sum_{i \in [n]} |\partial_l f_i(\boldsymbol{w}_{k,0})^2|} \left( \sum_{r=1}^{k-1} \beta_2^{rn} \eta_{k-r} \sum_{j=0}^{n-1} (\sqrt{D_1} \|\nabla f(\boldsymbol{w}_{k-r,j})\| + \sqrt{D_0}) \right)
$$

$$
\geq \beta_2^n \frac{1 - \beta_2}{1 - \beta_2^n} \sum_{i \in [n]} \partial_l f_i(\boldsymbol{w}_{k,0})^2 - 8\sqrt{2n}\eta_k C_1 L_0 \frac{1 - \beta_2}{(1 - \beta_2^n)^2}\beta_2^n \sqrt{\sum_{i \in [n]} |\partial_l f_i(\boldsymbol{w}_{k,0})^2|}
$$

$$
- 4L_1 C_1 \frac{1 - \beta_2}{1 - \beta_2^n} \frac{\sqrt{1 - \beta_2}}{1 - \sqrt{\beta_2}} \sqrt{\sum_{i \in [n]} |\partial_l f_i(\boldsymbol{w}_{k,0})^2|} \left( \sum_{r=1}^{k-1} \beta_2^n \sqrt{\beta_2}^{(r-1)n} \eta_{k-r} \sum_{j=0}^{n-1} (\sqrt{D_1} \|\nabla f(\boldsymbol{w}_{k-r,j})\| + \sqrt{D_0}) \right).
$$

The first claim is proved.

As for the upper bound, we have

$$\boldsymbol{\nu}_{l,k,0}$$

$$=(1-\beta_2)\partial_l f_{\tau_{k,0}}(\boldsymbol{w}_{k,0})^2 + \sum_{t=1}^{k-1}\sum_{j=0}^{n-1}(1-\beta_2)\beta_2^{tn-j}\partial_l f_{\tau_{k-t,j}}(\boldsymbol{w}_{k-t,j})^2 + \beta_2^{(k-1)n+1}\max_{i\in[n]}\partial_l f_i(\boldsymbol{w}_{1,0})^2$$

$$\leq 2(1-\beta_2)\partial_l f_{\tau_{k,0}}(\boldsymbol{w}_{k,0})^2 + 2\sum_{t=1}^{k-1}\sum_{j=0}^{n-1}(1-\beta_2)\beta_2^{tn-j}\partial_l f_{\tau_{k-t,j}}(\boldsymbol{w}_{k,0})^2 + 2\beta_2^{(k-1)n+1}\max_{i\in[n]}\partial_l f_i(\boldsymbol{w}_{k,0})^2$$

$$+2\sum_{t=1}^{k-1}\sum_{j=0}^{n-1}(1-\beta_2)\beta_2^{tn-j}\left(\sum_{p=j}^{n-1}L_1\sqrt{D_1}\|\nabla f(\boldsymbol{w}_{k-t,p})C_1\eta_{k-t}\sqrt{d}\right.$$

$$\left.+\sum_{r=1}^{t-1}\sum_{p=0}^{n-1}L_1\sqrt{D_1}\|\nabla f(\boldsymbol{w}_{k-r,p})\|C_1\eta_{k-r}\sqrt{d}+2\sqrt{2}(L_0+L_1\sqrt{D_0})C_1\sqrt{d}\eta_k(tn-j)\right)^2$$

$$+2\beta_2^{(k-1)n+1}\left(\sum_{r=1}^{k-1}\sum_{p=0}^{n-1}L_1\sqrt{D_1}\|\nabla f(\boldsymbol{w}_{k-r,p})\|C_1\eta_{k-r}\sqrt{d}+2\sqrt{2}(L_0+L_1\sqrt{D_0})C_1\sqrt{d}\eta_k(k-1)n\right)^2$$

$$\leq 2\max_{i\in[n]}\partial_l f_i(\boldsymbol{w}_{k,0})^2 + 2\left(\sum_{t=1}^{k-1}\sum_{j=0}^{n-1}\sqrt{1-\beta_2}\sqrt{\beta_2}^{tn-j}\left(\sum_{p=j}^{n-1}L_1\sqrt{D_1}\|\nabla f(\boldsymbol{w}_{k-t,p})C_1\eta_{k-t}\sqrt{d}\right.\right.$$

$$\left.+\sum_{r=1}^{t-1}\sum_{p=0}^{n-1}L_1\sqrt{D_1}\|\nabla f(\boldsymbol{w}_{k-r,p})\|C_1\eta_{k-r}\sqrt{d}+2\sqrt{2}(L_0+L_1\sqrt{D_0})C_1\sqrt{d}\eta_k(tn-j)\right)$$

$$\left.+\sqrt{\beta_2}^{(k-1)n+1}\left(\sum_{r=1}^{k-1}\sum_{p=0}^{n-1}L_1\sqrt{D_1}\|\nabla f(\boldsymbol{w}_{k-r,p})\|C_1\eta_{k-r}\sqrt{d}+2\sqrt{2}(L_0+L_1\sqrt{D_0})C_1\sqrt{d}\eta_k(k-1)n\right)\right)^2$$

$$\leq 2\max_{i\in[n]}\partial_l f_i(\boldsymbol{w}_{k,0})^2 + 2\left(2\sqrt{2}\eta_k C_1(L_0+L_1\sqrt{D_0})\frac{\sqrt{1-\beta_2}}{1-\sqrt{\beta_2}}\frac{\sqrt{\beta_2}}{1-\sqrt{\beta_2}}\right.$$

$$\left.+L_1 C_1\sqrt{D_1}\sum_{t=1}^{k-1}\eta_{k-t}\frac{\sqrt{1-\beta_2}}{1-\sqrt{\beta_2}}\sum_{j=0}^{n-1}\sqrt{\beta_2}^{tn-j}\|\nabla f(\boldsymbol{w}_{k-t,j})\|\right)^2$$

$$\leq 2\max_{i\in[n]}\partial_l f_i(\boldsymbol{w}_{k,0})^2 + 2\left(2\sqrt{2}\eta_k C_1(L_0+L_1\sqrt{D_0})\frac{\sqrt{1-\beta_2}}{1-\sqrt{\beta_2}}\frac{\sqrt{\beta_2}}{1-\sqrt{\beta_2}}\right.$$

$$\left.+L_1 C_1\sqrt{D_1}\sum_{t=1}^{k-1}\eta_{k-t}\frac{\sqrt{1-\beta_2}}{1-\sqrt{\beta_2}}\sum_{j=0}^{n-1}\sqrt{\beta_2}^{(t-1)n}\|\nabla f(\boldsymbol{w}_{k-t,j})\|\right)^2.$$

The proof is completed. $\qquad\square$

We then immediately have the following corollary when $\max_{i\in[n]}|\partial_l f_i(\boldsymbol{w}_{k,0})|$ is large enough compared to the error term.

**Corollary D.9** (Lemma 5.1, formal). *If*

$$\max_{i\in[n]}|\partial_l f_i(\boldsymbol{w}_{k,0})| \geq 4L_1 C_1\frac{\sqrt{1-\beta_2}}{1-\sqrt{\beta_2}}\left(\sum_{r=1}^{k-1}\sqrt{\beta_2}^{(r-1)n}\eta_{k-r}\sum_{j=0}^{n-1}(\sqrt{D_1}\|\nabla f(\boldsymbol{w}_{k-r,j})\| + \sqrt{D_0})\right)$$

$$+2\sqrt{2}\eta_k C_1(L_0+L_1\sqrt{D_0})\frac{\sqrt{1-\beta_2}}{1-\sqrt{\beta_2}}\frac{\sqrt{\beta_2}}{1-\sqrt{\beta_2}} + 8\sqrt{2n}\eta_k C_1 L_0\frac{1}{1-\beta_2^n}$$

$$+\eta_k C_1\left(n(L_0+L_1\sqrt{D_0}) + L_1\sqrt{D_1}\left(\sum_{p=0}^{n-1}\|\nabla f(\boldsymbol{w}_{k,p})\|\right)\right), \tag{12}$$

*then*

$$\frac{\beta_2^n}{2}\frac{1}{n}\sum_{i\in[n]}\partial_l f_i(\boldsymbol{w}_{k,0})^2 \leq \boldsymbol{\nu}_{l,k,0} \leq 4\max_{i\in[n]}\partial_l f_i(\boldsymbol{w}_{k,0})^2,$$

*where $C_1$ is defined in Eq. (9). Furthermore, if Eq. (12) holds, we have $\forall i \in \{0, \cdots, n-1\}$,*

$$\beta_2^{n-1} \boldsymbol{\nu}_{l,k,0} \leq \boldsymbol{\nu}_{l,k,i} \leq \left(\beta_2^{n-1} + 8n\frac{1 - \beta_2^{n-1}}{\beta_2^n}\right)\boldsymbol{\nu}_{l,k,0},$$

*and*

$$\frac{1}{\beta_2}\left(1 - (1 - \beta_2)\frac{2n}{\beta_2^n}\right)\boldsymbol{\nu}_{l,k,0} \leq \boldsymbol{\nu}_{l,k,-1} \leq \frac{1}{\beta_2}\boldsymbol{\nu}_{l,k,0},$$

*Proof.* The first claim is derived by directly applying the range of $\max_{i\in[n]}|\partial_l f_i(\boldsymbol{w}_{k,0})|$ into Lemma D.8.

As for the second claim, we have

$$\boldsymbol{\nu}_{l,k,i} = \beta_2^i \boldsymbol{\nu}_{l,k,0} + (1 - \beta_2)(\partial_l f_{\tau_{k,i}}(\boldsymbol{w}_{k,i})^2 + \cdots + \beta_2^{i-1}\partial_l f_{\tau_{k,i}}(\boldsymbol{w}_{k,1})^2).$$

On the other hand, since $\forall j \in \{0, \cdots, n-1\}$

$$|\partial_l f_i(\boldsymbol{w}_{k,j})| \leq \max_{p\in[n]}|\partial_l f_p(\boldsymbol{w}_{k,0})| + \eta_k C_1\left(j(L_0 + L_1\sqrt{D_0}) + L_1\sqrt{D_1}\left(\sum_{p=0}^{j-1}\|\nabla f(\boldsymbol{w}_{k,p})\|\right)\right)$$

$$\leq \max_{p\in[n]}|\partial_l f_p(\boldsymbol{w}_{k,0})| + \eta_k C_1\left(n(L_0 + L_1\sqrt{D_0}) + L_1\sqrt{D_1}\left(\sum_{p=0}^{n-1}\|\nabla f(\boldsymbol{w}_{k,p})\|\right)\right),$$

we have

$$\beta_2^{n-1}\boldsymbol{\nu}_{l,k,0} \leq \boldsymbol{\nu}_{l,k,i}$$
$$\leq \beta_2^i \boldsymbol{\nu}_{l,k,0} + 2(1 - \beta_2)\max_{p\in[n]}\partial_l f_p(\boldsymbol{w}_{k,0})^2(1 + \cdots + \beta_2^{i-1})$$
$$+ 2(1 - \beta_2)(1 + \cdots + \beta_2^{i-1})\eta_k^2 C_1^2\left(n(L_0 + L_1\sqrt{D_0}) + L_1\sqrt{D_1}\left(\sum_{p=0}^{n-1}\|\nabla f(\boldsymbol{w}_{k,p})\|\right)\right)^2$$
$$= \beta_2^i \boldsymbol{\nu}_{l,k,0} + 2(1 - \beta_2^i)\max_{p\in[n]}\partial_l f_p(\boldsymbol{w}_{k,0})^2 + 2(1 - \beta_2^i)\eta_k^2 C_1^2\left(n(L_0 + L_1\sqrt{D_0}) + L_1\sqrt{D_1}\left(\sum_{p=0}^{n-1}\|\nabla f(\boldsymbol{w}_{k,p})\|\right)\right)^2.$$

Therefore, if Eq. (12) holds, we then have

$$\boldsymbol{\nu}_{l,k,i} \leq \beta_2^i \boldsymbol{\nu}_{l,k,0} + 4(1 - \beta_2^i)\max_{p\in[n]}\partial_l f_p(\boldsymbol{w}_{k,0})^2$$
$$\leq \beta_2^i \boldsymbol{\nu}_{l,k,0} + 4\frac{n}{n}(1 - \beta_2^i)\sum_{p\in[n]}\partial_l f_p(\boldsymbol{w}_{k,0})^2 \leq \left(\beta_2^i + 8n\frac{1 - \beta_2^i}{\beta_2^n}\right)\boldsymbol{\nu}_{l,k,0}$$
$$\leq \left(\beta_2^{n-1} + 8n\frac{1 - \beta_2^{n-1}}{\beta_2^n}\right)\boldsymbol{\nu}_{l,k,0}.$$

Following the same routine, we have

$$\beta_2 \boldsymbol{\nu}_{l,k,-1} \leq \boldsymbol{\nu}_{l,k,0},$$

and if Eq. (12) holds,

$$\boldsymbol{\nu}_{l,k,-1} = \frac{1}{\beta_2}\left(\boldsymbol{\nu}_{l,k,0} - (1 - \beta_2)\partial_l f_{\tau_{k,0}}(\boldsymbol{w}_{k,0})^2\right) \geq \frac{1}{\beta_2}\left(\boldsymbol{\nu}_{l,k,0} - (1 - \beta_2)\max_p \partial_l f_p(\boldsymbol{w}_{k,0})^2\right)$$
$$\geq \boldsymbol{\nu}_{l,k,0}\frac{1}{\beta_2}\left(1 - (1 - \beta_2)\frac{2n}{\beta_2^n}\right).$$

The proof of the second claim is completed. $\qquad\square$

*Remark* D.10. By the notations in Eq. (9)., Eq. (12) can be translated into

$$\max_{i \in [n]} |\partial_l f_i(\boldsymbol{w}_{k,0})| \geq C_3 \eta_k + C_4 \sum_{r=1}^{k-1} \sqrt{\beta_2}^{(r-1)n} \eta_{k-r} \sum_{j=0}^{n-1} \|\nabla f(\boldsymbol{w}_{k-r,j})\|$$
$$+ C_4 n \sum_{r=1}^{k-1} \sqrt{\beta_2}^{(r-1)n} \eta_{k-r} + \eta_k C_4 \left( \sum_{j=0}^{n-1} \|\nabla f(\boldsymbol{w}_{k,j})\| \right). \tag{13}$$

Furthermore, we define $g(\beta_2)$ as

$$g(\beta_2) \triangleq \max \left\{ \frac{1}{\sqrt{\beta_2}^{n-1}} - 1, 1 - \frac{1}{\sqrt{\beta_2^{n-1} + 8n\frac{1-\beta_2^{n-1}}{\beta_2^n}}}, 1 - \sqrt{\beta_2}, \sqrt{\frac{\beta_2}{\left(1 - (1-\beta_2)\frac{2n}{\beta_2^n}\right)} - 1} \right\},$$

and the conclusion of Corollary D.9 can be translated into that if Eq. (13) holds,

$$\left| \frac{1}{\sqrt{\boldsymbol{\nu}_{l,k,i}}} - \frac{1}{\sqrt{\boldsymbol{\nu}_{l,k,0}}} \right| \leq g(\beta_2) \frac{1}{\sqrt{\boldsymbol{\nu}_{l,k,0}}},$$

and

$$\left| \frac{1}{\sqrt{\boldsymbol{\nu}_{l,k,-1}}} - \frac{1}{\sqrt{\boldsymbol{\nu}_{l,k,0}}} \right| \leq g(\beta_2) \frac{1}{\sqrt{\boldsymbol{\nu}_{l,k,0}}}.$$

Based on whether Eq. (13) is fulfilled, we divide $[d]$ into $\mathbb{L}_{large}^k$ and $\mathbb{L}_{small}^k$ ($\forall k \geq 1$), which are respectively defined as

$$\mathbb{L}_{large}^k = \{l : l \in [d], \text{s.t. Eq. (13) holds}\},$$
$$\mathbb{L}_{small}^k = \{l : l \in [d], \text{s.t. Eq. (13) doesn't hold}\}.$$

The following lemma characterizes the property of $\mathbb{L}_{small}^k$.

**Lemma D.11.** *Define* $\boldsymbol{u}_k \triangleq \frac{\boldsymbol{w}_{k,0} - \beta_1 \boldsymbol{w}_{k,-1}}{1 - \beta_1}$ *(with* $\boldsymbol{w}_{1,-1} \triangleq \boldsymbol{w}_{1,0}$*). Then,*

$$\sum_{k=1}^{T} \left| \sum_{l \in \mathbb{L}_{small}^k} \partial_l f(\boldsymbol{w}_{k,0})(\boldsymbol{u}_{l,k+1} - \boldsymbol{u}_{l,k}) \right| \leq C_2 \left( C_5 \sum_{k=1}^{T} \eta_k^2 \|\nabla f(\boldsymbol{w}_{k,0})\| + C_6 \ln T + C_7 \right),$$

*where* $C_2$, $C_5$, $C_6$, *and* $C_7$ *are defined in Eq. (9).*

*Proof.* By directly applying the definition of $\mathbb{L}_{large}^k$ and Lemma D.4, we have

$$\frac{1}{n} \left| \sum_{l \in \mathbb{L}_{small}^k} \partial_l f(\boldsymbol{w}_{k,0})(\boldsymbol{u}_{l,k+1} - \boldsymbol{u}_{l,k}) \right|$$
$$\leq dC_2 \eta_k \left( C_3 \eta_k + C_4 \sum_{r=1}^{k-1} \sqrt{\beta_2}^{(r-1)n} \eta_{k-r} \sum_{j=0}^{n-1} \|\nabla f(\boldsymbol{w}_{k-r,j})\| + C_4 n \sum_{r=1}^{k-1} \sqrt{\beta_2}^{(r-1)n} \eta_{k-r} + \eta_k C_4 \left( \sum_{p=0}^{n-1} \|\nabla f(\boldsymbol{w}_{k,p})\| \right) \right).$$

Summing over $k$ from 1 to $t$ then leads to

$$\frac{1}{n}\sum_{k=1}^{T}\left|\sum_{l\in\mathbb{L}_{small}^{k}}\partial_{l}f(\boldsymbol{w}_{k,0})(\boldsymbol{u}_{l,k+1}-\boldsymbol{u}_{l,k})\right|$$

$$\leq\sum_{k=1}^{T}dC_{2}C_{3}\eta_{k}^{2}+dC_{2}C_{4}\sum_{k=1}^{T}\eta_{k}\sum_{r=1}^{k-1}\sqrt{\beta_{2}}^{(r-1)n}\eta_{k-r}\sum_{j=0}^{n-1}\|\nabla f(\boldsymbol{w}_{k-r,j})\|+C_{2}C_{4}n\sum_{k=1}^{T}\eta_{k}\sum_{r=1}^{k-1}\sqrt{\beta_{2}}^{(r-1)n}\eta_{k-r}$$

$$+C_{2}C_{4}\sum_{k=1}^{T}\eta_{k}^{2}\sum_{p=0}^{n-1}\|\nabla f(\boldsymbol{w}_{k,p})\|$$

$$\leq\sum_{k=1}^{T}dC_{2}C_{3}\eta_{k}^{2}+\frac{dC_{2}C_{4}}{1-\sqrt{\beta_{2}^{n}}}\sum_{k=1}^{T-1}\eta_{k}^{2}\sum_{j=0}^{n-1}\|\nabla f(\boldsymbol{w}_{k,j})\|+\frac{C_{2}C_{4}n}{1-\sqrt{\beta_{2}^{n}}}\sum_{k=1}^{T-1}\eta_{k}^{2}+C_{2}C_{4}\sum_{k=1}^{T}\eta_{k}^{2}\sum_{p=0}^{n-1}\|\nabla f(\boldsymbol{w}_{k,p})\|$$

$$\leq\left(dC_{2}C_{3}+\frac{C_{2}C_{4}n}{1-\sqrt{\beta_{2}^{n}}}\right)\eta_{1}^{2}(1+\ln T)+\left(C_{2}C_{4}+\frac{dC_{2}C_{4}}{1-\sqrt{\beta_{2}^{n}}}\right)\sum_{k=1}^{T}\eta_{k}^{2}\sum_{j=0}^{n-1}\|\nabla f(\boldsymbol{w}_{k,j})\|,$$

where in the second inequality we exchange the sum order. By Lemma D.5, the above inequality further leads to

$$\sum_{k=1}^{T}\left|\sum_{l\in\mathbb{L}_{small}^{k}}\partial_{l}f(\boldsymbol{w}_{k,0})(\boldsymbol{u}_{l,k+1}-\boldsymbol{u}_{l,k})\right|$$

$$\leq n\left(C_{2}C_{4}+\frac{dC_{2}C_{4}}{1-\sqrt{\beta_{2}^{n}}}\right)\sum_{k=1}^{T}\eta_{k}^{2}\sum_{j=0}^{n-1}\left((1+n\sqrt{d}C_{1}\eta_{1}L_{1}\sqrt{n})\|\nabla f(\boldsymbol{w}_{k,0})\|+\left(nL_{0}+L_{1}\sqrt{n}\sqrt{D_{0}}\right)n\sqrt{d}C_{1}\eta_{k}\right)$$

$$+n\left(dC_{2}C_{3}+\frac{C_{2}C_{4}n}{1-\sqrt{\beta_{2}^{n}}}\right)\eta_{1}^{2}(1+\ln T)$$

$$\leq n^{2}(1+n\sqrt{d}C_{1}\eta_{1}L_{1}\sqrt{n})\left(C_{2}C_{4}+\frac{dC_{2}C_{4}}{1-\sqrt{\beta_{2}^{n}}}\right)\sum_{k=1}^{T}\eta_{k}^{2}\|\nabla f(\boldsymbol{w}_{k,0})\|+\left(dC_{2}C_{3}+\frac{C_{2}C_{4}n}{1-\sqrt{\beta_{2}^{n}}}\right)\eta_{1}^{2}(1+\ln T)$$

$$+n\left(C_{2}C_{4}+\frac{dC_{2}C_{4}}{1-\sqrt{\beta_{2}^{n}}}\right)\left(nL_{0}+L_{1}\sqrt{n}\sqrt{D_{0}}\right)n^{2}\sqrt{d}C_{1}\sum_{k=1}^{T}\eta_{k}^{3}$$

$$\leq n^{2}(1+n\sqrt{d}C_{1}\eta_{1}L_{1}\sqrt{n}\sqrt{D_{1}})\left(C_{2}C_{4}+\frac{dC_{2}C_{4}\sqrt{D_{1}}}{1-\sqrt{\beta_{2}^{n}}}\right)\sum_{k=1}^{T}\eta_{k}^{2}\|\nabla f(\boldsymbol{w}_{k,0})\|+\left(dC_{2}C_{3}+\frac{C_{2}C_{4}n\sqrt{D_{1}}}{1-\sqrt{\beta_{2}^{n}}}\right)$$

$$\times\eta_{1}^{2}(1+\ln T)+3n\left(C_{2}C_{4}+\frac{dC_{2}C_{4}}{1-\sqrt{\beta_{2}^{n}}}\right)\left(nL_{0}+L_{1}\sqrt{n}\sqrt{D_{0}}\right)n^{2}\sqrt{d}C_{1}\eta_{1}^{3}. \tag{14}$$

By the notations in Eq. (9), the proof is completed. $\qquad\square$

The next lemma characterizes the property of $\mathbb{L}_{large}^{k}$.

**Lemma D.12.** *Define* $\boldsymbol{u}_{k}\triangleq\frac{\boldsymbol{w}_{k,0}-\beta_{1}\boldsymbol{w}_{k,-1}}{1-\beta_{1}}$ *(with* $\boldsymbol{w}_{1,-1}\triangleq\boldsymbol{w}_{1,0}$*). We have*

$$\sum_{k=1}^{T}\sum_{l\in\mathbb{L}_{large}^{k}}\partial_{l}f(\boldsymbol{w}_{k,0})(\boldsymbol{u}_{l,k+1}-\boldsymbol{u}_{l,k})$$

$$\leq-\sum_{k=1}^{T}\sum_{l\in[d]}\frac{\eta_{k}\partial_{l}f(\boldsymbol{w}_{k,0})^{2}}{2\max_{i\in[n]}|\partial_{l}f_{i}(\boldsymbol{w}_{k,0})|+\xi}$$

$$+\sum_{k=1}^{T}\sum_{l\in\mathbb{L}_{large}^{k}}\eta_{k}g(\beta_{2})\left(n-1+\frac{1+\beta_{1}}{1-\beta_{1}}\right)\frac{|\partial_{l}f(\boldsymbol{w}_{k,0})|}{\sqrt{\frac{\beta_{2}^{n}}{2n}}\max_{i\in[n]}|\partial_{l}f_{i}(\boldsymbol{w}_{k,0})|+\xi}\left(\max_{i\in[n]}|\partial_{l}f_{i}(\boldsymbol{w}_{k,0})|\right)$$

$$+\left((C_{8}+\frac{1}{2}C_{5})\sum_{k=1}^{T}\eta_{k}^{2}\|\nabla f(\boldsymbol{w}_{k,0})\|+(C_{9}+\frac{1}{2}C_{6})\ln T+(C_{10}+\frac{1}{2}C_{7})\right).$$

*Proof.* Compared to the proof of Lemma D.11, the proof of this lemma is more complicated. To begin with, we provide a decomposition of $\boldsymbol{u}_{k+1} - \boldsymbol{u}_k$. According to the definition of $\boldsymbol{u}_k$, we have

$$
\begin{aligned}
&\boldsymbol{u}_{k+1} - \boldsymbol{u}_k \\
&= \frac{(\boldsymbol{w}_{k+1,0} - \beta_1 \boldsymbol{w}_{k+1,-1}) - (\boldsymbol{w}_{k,0} - \beta_1 \boldsymbol{w}_{k,-1})}{1 - \beta_1} \\
&= \frac{(\boldsymbol{w}_{k+1,0} - \boldsymbol{w}_{k,0}) - \beta_1 (\boldsymbol{w}_{k+1,-1} - \boldsymbol{w}_{k,-1})}{1 - \beta_1} \\
&= \frac{\sum_{i=0}^{n-1} (\boldsymbol{w}_{k,i+1} - \boldsymbol{w}_{k,i}) - \beta_1 \sum_{i=0}^{n-1} (\boldsymbol{w}_{k,i} - \boldsymbol{w}_{k,i-1})}{1 - \beta_1} \\
&= \frac{(\boldsymbol{w}_{k+1,0} - \boldsymbol{w}_{k+1,-1}) + (1 - \beta_1) \sum_{i=0}^{n-2} (\boldsymbol{w}_{k,i+1} - \boldsymbol{w}_{k,i}) - \beta_1 (\boldsymbol{w}_{k,0} - \boldsymbol{w}_{k,-1})}{1 - \beta_1} \\
&\overset{(\star)}{=} -\frac{\frac{\eta_k}{\sqrt{\boldsymbol{\nu}_{k,n-1}}} \odot \boldsymbol{m}_{k,n-1} + (1 - \beta_1) \sum_{i=0}^{n-2} \frac{\eta_k}{\sqrt{\boldsymbol{\nu}_{k,i}}} \odot \boldsymbol{m}_{k,i} - \beta_1 \frac{\eta_{k-1}}{\sqrt{\boldsymbol{\nu}_{k-1,n-1}}} \odot \boldsymbol{m}_{k-1,n-1}}{1 - \beta_1} \\
&= -\frac{\eta_k}{\sqrt{\boldsymbol{\nu}_{k,0}}} \odot \frac{\boldsymbol{m}_{k,n-1} + (1 - \beta_1) \sum_{i=0}^{n-2} \boldsymbol{m}_{k,i} - \beta_1 \boldsymbol{m}_{k-1,n-1}}{1 - \beta_1} - \eta_k \left( \left( \frac{1}{\sqrt{\boldsymbol{\nu}_{k,n-1}}} - \frac{1}{\sqrt{\boldsymbol{\nu}_{k,0}}} \right) \odot \frac{\boldsymbol{m}_{k,n-1}}{1 - \beta_1} \right. \\
&\qquad + \sum_{i=0}^{n-2} \left( \frac{1}{\sqrt{\boldsymbol{\nu}_{k,i}}} - \frac{1}{\sqrt{\boldsymbol{\nu}_{k,0}}} \right) \odot \boldsymbol{m}_{k,i} - \frac{\beta_1}{1 - \beta_1} \left( \frac{1}{\sqrt{\boldsymbol{\nu}_{k-1,n-1}}} - \frac{1}{\sqrt{\boldsymbol{\nu}_{k,0}}} \right) \odot \boldsymbol{m}_{k-1,n-1} \Bigg) \\
&\qquad - \frac{\beta_1}{1 - \beta_1} (\eta_{k-1} - \eta_k) \frac{1}{\sqrt{\boldsymbol{\nu}_{k-1,n-1}}} \odot \boldsymbol{m}_{k-1,n-1}.
\end{aligned}
\tag{15}
$$

Here equation $(\star)$ is due to a direct application of the update rule of $\boldsymbol{w}_{k,i}$. We then analyze the above three terms respectively, namely, we define

$$
a_l^1 \triangleq -\frac{\eta_k}{\sqrt{\boldsymbol{\nu}_{l,k,0}}} \frac{\boldsymbol{m}_{l,k,n-1} + (1 - \beta_1) \sum_{i=0}^{n-2} \boldsymbol{m}_{l,k,i} - \beta_1 \boldsymbol{m}_{l,k-1,n-1}}{1 - \beta_1} = -\frac{\eta_k}{\sqrt{\boldsymbol{\nu}_{l,k,0}}} \sum_{i=0}^{n-1} \partial_l f_{\tau_{k,i}}(\boldsymbol{w}_{k,i}),
$$

$$
a_l^2 \triangleq -\eta_k \left( \left( \frac{1}{\sqrt{\boldsymbol{\nu}_{l,k,n-1}}} - \frac{1}{\sqrt{\boldsymbol{\nu}_{l,k,0}}} \right) \frac{\boldsymbol{m}_{l,k,n-1}}{1 - \beta_1} + \sum_{i=0}^{n-2} \left( \frac{1}{\sqrt{\boldsymbol{\nu}_{l,k,i}}} - \frac{1}{\sqrt{\boldsymbol{\nu}_{l,k,0}}} \right) \boldsymbol{m}_{l,k,i} \right.
$$

$$
\left. - \frac{\beta_1}{1 - \beta_1} \left( \frac{1}{\sqrt{\boldsymbol{\nu}_{l,k-1,n-1}}} - \frac{1}{\sqrt{\boldsymbol{\nu}_{l,k,0}}} \right) \boldsymbol{m}_{l,k-1,n-1} \right),
$$

$$
a_l^3 \triangleq -\frac{\beta_1}{1 - \beta_1} (\eta_{k-1} - \eta_k) \frac{1}{\sqrt{\boldsymbol{\nu}_{l,k-1,n-1}}} \boldsymbol{m}_{l,k-1,n-1}.
$$

One can then easily observe that by Eq. (15),

$$
\sum_{l \in \mathbb{L}_{large}^k} \partial_l f(\boldsymbol{w}_{k,0})(\boldsymbol{u}_{l,k+1} - \boldsymbol{u}_{l,k}) = \sum_{l \in \mathbb{L}_{large}^k} \partial_l f(\boldsymbol{w}_{k,0}) a_l^1 + \sum_{l \in \mathbb{L}_{large}^k} \partial_l f(\boldsymbol{w}_{k,0}) a_l^2 + \sum_{l \in \mathbb{L}_{large}^k} \partial_l f(\boldsymbol{w}_{k,0}) a_l^3.
$$

① **Tackling Term** $\sum_{l \in \mathbb{L}_{large}^k} \partial_l f(\boldsymbol{w}_{k,0}) a_l^1$**:**

We have

$$
\begin{aligned}
&\sum_{l \in \mathbb{L}_{large}^k} \partial_l f(\boldsymbol{w}_{k,0}) a_l^1 \\
&= -\sum_{l \in \mathbb{L}_{large}^k} \partial_l \frac{\eta_k}{\sqrt{\boldsymbol{\nu}_{l,k,0}}} \partial_l f(\boldsymbol{w}_{k,0}) \left( \sum_{i=0}^{n-1} \partial_l f_{\tau_{k,i}}(\boldsymbol{w}_{k,0}) \right) - \sum_{l \in \mathbb{L}_{large}^k} \frac{\eta_k}{\sqrt{\boldsymbol{\nu}_{l,k,0}}} \partial_l f(\boldsymbol{w}_{k,0}) \left( \sum_{i=0}^{n-1} (\partial_l f_{\tau_{k,i}}(\boldsymbol{w}_{k,i}) - \partial_l f_{\tau_{k,i}}(\boldsymbol{w}_{k,0})) \right) \\
&= -\sum_{l \in \mathbb{L}_{large}^k} \frac{\eta_k}{\sqrt{\boldsymbol{\nu}_{l,k,0}}} \partial_l f(\boldsymbol{w}_{k,0})^2 - \sum_{l \in \mathbb{L}_{large}^k} \frac{\eta_k}{\sqrt{\boldsymbol{\nu}_{l,k,0}}} \partial_l f(\boldsymbol{w}_{k,0}) \left( \sum_{i=0}^{n-1} (\partial_l f_{\tau_{k,i}}(\boldsymbol{w}_{k,i}) - \partial_l f_{\tau_{k,i}}(\boldsymbol{w}_{k,0})) \right) \\
&\overset{(\star)}{=} -\sum_{l \in \mathbb{L}_{large}^k} \frac{\eta_k}{\sqrt{\boldsymbol{\nu}_{l,k,0}}} \partial_l f(\boldsymbol{w}_{k,0})^2 + \mathcal{O}\left(\eta_k^2\right) + \mathcal{O}\left(\eta_k^2 \|\nabla f(\boldsymbol{w}_{k,0})\|\right),
\end{aligned}
$$

where Eq. $(\star)$ is due to

$$
\left| \sum_{l \in \mathbb{L}^k_{large}} \frac{\eta_k}{\sqrt{\boldsymbol{\nu}_{l,k,0}}} \partial_l f(\boldsymbol{w}_{k,0}) \left( \sum_{i=0}^{n-1} (\partial_l f_{\tau_{k,i}}(\boldsymbol{w}_{k,i}) - \partial_l f_{\tau_{k,i}}(\boldsymbol{w}_{k,0})) \right) \right|
$$

$$
\overset{(*)}{\leq} \eta_k \sqrt{\frac{2n^2}{\beta_2^n}} \left( \sum_{l \in \mathbb{L}^k_{large}} \sum_{i=0}^{n-1} |\partial_l f_{\tau_{k,i}}(\boldsymbol{w}_{k,i}) - \partial_l f_{\tau_{k,i}}(\boldsymbol{w}_{k,0})| \right)
$$

$$
\leq \eta_k \sqrt{\frac{2n^2}{\beta_2^n}} \left( \sqrt{d} \sum_{i=0}^{n-1} \|\nabla f_{\tau_{k,i}}(\boldsymbol{w}_{k,i}) - \nabla f_{\tau_{k,i}}(\boldsymbol{w}_{k,0})\| \right)
$$

$$
\overset{(\circ)}{\leq} \eta_k \sqrt{\frac{2n^2}{\beta_2^n}} \sqrt{d} \sum_{i=0}^{n-1} (L_0 + L_1 \|\nabla f_{\tau_{k,i}}(\boldsymbol{w}_{k,0})\|) \|\boldsymbol{w}_{k,i} - \boldsymbol{w}_{k,0}\|
$$

$$
\leq \eta_k \sqrt{\frac{2n^2}{\beta_2^n}} \sqrt{d}(nL_0 + L_1\sqrt{D_1}\sqrt{n}\|\nabla f(\boldsymbol{w}_{k,0})\| + \sqrt{n}L_1\sqrt{D_0})n\sqrt{d}C_1\eta_k
$$

$$
\overset{(\bullet)}{\leq} \sqrt{\frac{2n^2}{\beta_2^n}} d(n^2 L_0 + n\sqrt{n}L_1\sqrt{D_0})C_1\eta_k^2 + \eta_k^2 d\sqrt{\frac{2n^2}{\beta_2^n}} L_1\sqrt{D_1}n\sqrt{n}\|\nabla f(\boldsymbol{w}_{k,0})\|.
$$

Here Eq. $(*)$ is due to Corollary D.9, Eq. $(\circ)$ is due to $f_i$ is $(L_0, L_1)$-smooth, $\forall i$, and Eq. $(\bullet)$ is due to Lemma D.3.

② **Tackling Term** $\sum_{l \in \mathbb{L}^k_{large}} \partial_l f(\boldsymbol{w}_{k,0})a_l^2$**:**

We have for any $l \in \mathbb{L}_{max}$,

$$
|\partial_l f(\boldsymbol{w}_{k,0})a_l^2|
$$

$$
\leq \eta_k |\partial_l f(\boldsymbol{w}_{k,0})| \left( \left| \frac{1}{\sqrt{\boldsymbol{\nu}_{l,k,n-1}}} - \frac{1}{\sqrt{\boldsymbol{\nu}_{l,k,0}}} \right| \frac{|\boldsymbol{m}_{l,k,n-1}|}{1-\beta_1} + \sum_{i=0}^{n-2} \left| \frac{1}{\sqrt{\boldsymbol{\nu}_{l,k,i}}} - \frac{1}{\sqrt{\boldsymbol{\nu}_{l,k,0}}} \right| |\boldsymbol{m}_{l,k,i}| \right.
$$

$$
\left. - \frac{\beta_1}{1-\beta_1} \left| \frac{1}{\sqrt{\boldsymbol{\nu}_{l,k-1,n-1}}} + \frac{1}{\sqrt{\boldsymbol{\nu}_{l,k,0}}} \right| |\boldsymbol{m}_{l,k-1,n-1}| \right)
$$

$$
\overset{(\star)}{\leq} \eta_k g(\beta_2) \frac{|\partial_l f(\boldsymbol{w}_{k,0})|}{\sqrt{\boldsymbol{\nu}_{l,k,0}}} \left( \frac{|\boldsymbol{m}_{l,k,n-1}|}{1-\beta_1} + \sum_{i=0}^{n-2} |\boldsymbol{m}_{l,k,i}| + \frac{\beta_1}{1-\beta_1} |\boldsymbol{m}_{l,k-1,n-1}| \right)
$$

$$
\overset{(*)}{\leq} \eta_k g(\beta_2) \left( n - 1 + \frac{1+\beta_1}{1-\beta_1} \right) \frac{|\partial_l f(\boldsymbol{w}_{k,0})|}{\sqrt{\boldsymbol{\nu}_{l,k,0}}} \left( \max_{i \in [n]} |\partial_l f_i(\boldsymbol{w}_{k,0})| \right)
$$

$$
+ \eta_k^2 g(\beta_2) \left( n - 1 + \frac{1+\beta_1}{1-\beta_1} \right) \frac{\sqrt{2}n}{\beta_2^{\frac{n}{2}}} \left( n + \frac{2\sqrt{2}\beta_1}{1-\beta_1} \right) C_1(L_0 + L_1\sqrt{D_0})\sqrt{d}
$$

$$
+ \eta_k^2 g(\beta_2) \left( n - 1 + \frac{1+\beta_1}{1-\beta_1} \right) \frac{\sqrt{2}n}{\beta_2^{\frac{n}{2}}} L_1 C_1 \sqrt{D_1} \sum_{j=0}^{n-1} \|\nabla f(\boldsymbol{w}_{k,j})\|
$$

$$
+ \eta_k g(\beta_2) \left( n - 1 + \frac{1+\beta_1}{1-\beta_1} \right) \frac{\sqrt{2}n}{\beta_2^{\frac{n}{2}}} L_1 C_1 \sqrt{D_1} \sum_{t=1}^{k-1} \eta_{k-t} \sum_{j=0}^{n-1} \beta_1^{tn-1-j} \|\nabla f(\boldsymbol{w}_{k-t,j})\|,
$$

where Inequality $(\star)$ is due to Corollary D.9, and $g(\beta_2)$ is defined in Lemma D.10, and Inequality $(*)$ is due to Lemma D.7, by which we have $\forall i \in \{-1, \cdots, n-1\}$

$$
|\boldsymbol{m}_{l,k,i}| \leq \max_{i' \in [n]} |\partial_l f_{i'}(\boldsymbol{w}_{k,0})| + \left( n + \frac{2\sqrt{2}\beta_1}{1-\beta_1} \right) C_1(L_0 + L_1\sqrt{D_0})\sqrt{d}\eta_k + L_1 C_1 \sqrt{D_1} \eta_k \sum_{j=0}^{n-1} \|\nabla f(\boldsymbol{w}_{k,j})\|
$$

$$
+ L_1 C_1 \sqrt{D_1} \sum_{t=1}^{k-1} \eta_{k-t} \sum_{j=0}^{n-1} \beta_1^{tn-1-j} \|\nabla f(\boldsymbol{w}_{k-t,j})\|.
$$

Therefore, summing over $\mathbb{L}_{large}^k$ and $k$ leads to

$$\sum_{k=1}^{T} \left| \sum_{l \in \mathbb{L}_{large}^k} \partial_l f(\boldsymbol{w}_{k,0}) a_l^2 \right|$$

$$\leq \sum_{k=1}^{T} \sum_{l \in \mathbb{L}_{large}^k} \eta_k g(\beta_2) \left( n - 1 + \frac{1+\beta_1}{1-\beta_1} \right) \frac{|\partial_l f(\boldsymbol{w}_{k,0})|}{\sqrt{\boldsymbol{\nu}_{l,k,0}}} \left( \max_{i \in [n]} |\partial_l f_i(\boldsymbol{w}_{k,0})| \right)$$

$$+ \sum_{k=1}^{T} \eta_k^2 g(\beta_2) \left( n - 1 + \frac{1+\beta_1}{1-\beta_1} \right) \frac{\sqrt{2}n}{\beta_2^{\frac{n}{2}}} \left( n + \frac{2\sqrt{2}\beta_1}{1-\beta_1} \right) C_1 (L_0 + L_1 \sqrt{D_0}) d\sqrt{d}$$

$$+ dg(\beta_2) \left( n - 1 + \frac{1+\beta_1}{1-\beta_1} \right) \frac{\sqrt{2}n}{\beta_2^{\frac{n}{2}}} L_1 C_1 \sqrt{D_1} \sum_{k=1}^{T} \eta_k^2 \sum_{j=0}^{n-1} \|\nabla f(\boldsymbol{w}_{k,j})\|$$

$$+ dg(\beta_2) \left( n - 1 + \frac{1+\beta_1}{1-\beta_1} \right) \frac{\sqrt{2}n}{\beta_2^{\frac{n}{2}}} L_1 C_1 \sqrt{D_1} \sum_{k=1}^{T} \eta_k \sum_{t=1}^{k-1} \eta_{k-t} \sum_{j=0}^{n-1} \beta_1^{(t-1)n} \|\nabla f(\boldsymbol{w}_{k-t,j})\|$$

$$\leq \sum_{k=1}^{T} \sum_{l \in \mathbb{L}_{large}^k} \eta_k g(\beta_2) \left( n - 1 + \frac{1+\beta_1}{1-\beta_1} \right) \frac{|\partial_l f(\boldsymbol{w}_{k,0})|}{\sqrt{\boldsymbol{\nu}_{l,k,0}}} \left( \max_{i \in [n]} |\partial_l f_i(\boldsymbol{w}_{k,0})| \right)$$

$$+ g(\beta_2) \left( n - 1 + \frac{1+\beta_1}{1-\beta_1} \right) \frac{\sqrt{2}n}{\beta_2^{\frac{n}{2}}} \left( n + \frac{2\sqrt{2}\beta_1}{1-\beta_1} \right) C_1 (L_0 + L_1 \sqrt{D_0}) d\sqrt{d} \eta_1 (1 + \ln T)$$

$$+ dg(\beta_2) \left( n - 1 + \frac{1+\beta_1}{1-\beta_1} \right) \frac{\sqrt{2}n}{\beta_2^{\frac{n}{2}}} L_1 C_1 \sqrt{D_1} \left( 1 + \frac{1}{1-\beta_2^n} \right) \sum_{k=1}^{T} \eta_k^2 \sum_{j=0}^{n-1} \|\nabla f(\boldsymbol{w}_{k,j})\|$$

$$\overset{(\star)}{\leq} \sum_{k=1}^{T} \sum_{l \in \mathbb{L}_{large}^k} \eta_k g(\beta_2) \left( n - 1 + \frac{1+\beta_1}{1-\beta_1} \right) \frac{|\partial_l f(\boldsymbol{w}_{k,0})|}{\sqrt{\boldsymbol{\nu}_{l,k,0}}} \left( \max_{i \in [n]} |\partial_l f_i(\boldsymbol{w}_{k,0})| \right)$$

$$+ g(\beta_2) \left( n - 1 + \frac{1+\beta_1}{1-\beta_1} \right) \frac{\sqrt{2}n}{\beta_2^{\frac{n}{2}}} \left( n + \frac{2\sqrt{2}\beta_1}{1-\beta_1} \right) C_1 (L_0 + L_1 \sqrt{D_0}) d\sqrt{d} \eta_1 (1 + \ln T)$$

$$+ dg(\beta_2) \left( n - 1 + \frac{1+\beta_1}{1-\beta_1} \right) \frac{\sqrt{2}n}{\beta_2^{\frac{n}{2}}} L_1 C_1 \sqrt{D_1} \left( 1 + \frac{1}{1-\beta_2^n} \right)$$

$$\cdot \sum_{k=1}^{T} \eta_k^2 \sum_{j=0}^{n-1} \left( (1 + n\sqrt{d} C_1 \eta_1 L_1 \sqrt{n} \sqrt{D_1}) \|\nabla f(\boldsymbol{w}_{k,0})\| + \left( nL_0 + L_1 \sqrt{n} \sqrt{D_0} \right) n\sqrt{d} C_1 \eta_k \right)$$

$$\leq \sum_{k=1}^{T} \sum_{l \in \mathbb{L}_{large}^k} \eta_k g(\beta_2) \left( n - 1 + \frac{1+\beta_1}{1-\beta_1} \right) \frac{|\partial_l f(\boldsymbol{w}_{k,0})|}{\sqrt{\boldsymbol{\nu}_{l,k,0}}} \left( \max_{i \in [n]} |\partial_l f_i(\boldsymbol{w}_{k,0})| \right)$$

$$+ g(\beta_2) \left( n - 1 + \frac{1+\beta_1}{1-\beta_1} \right) \frac{\sqrt{2}n}{\beta_2^{\frac{n}{2}}} \left( n + \frac{2\sqrt{2}\beta_1}{1-\beta_1} \right) C_1 (L_0 + L_1 \sqrt{D_0}) d\sqrt{d} \eta_1^2 (1 + \ln T)$$

$$+ dg(\beta_2) \left( n - 1 + \frac{1+\beta_1}{1-\beta_1} \right) \frac{\sqrt{2}n}{\beta_2^{\frac{n}{2}}} L_1 C_1 \sqrt{D_1} \left( 1 + \frac{1}{1-\beta_2^n} \right) (n + n^{\frac{5}{2}} \sqrt{d} C_1 \eta_1 L_1 \sqrt{D_1}) \sum_{k=1}^{T} \eta_k^2 \|\nabla f(\boldsymbol{w}_{k,0})\|$$

$$+ 3 dg(\beta_2) \left( n - 1 + \frac{1+\beta_1}{1-\beta_1} \right) \frac{\sqrt{2}n}{\beta_2^{\frac{n}{2}}} L_1 C_1 \sqrt{D_1} \left( 1 + \frac{1}{1-\beta_2^n} \right) n \left( nL_0 + L_1 \sqrt{n} \sqrt{D_0} \right) n\sqrt{d} C_1 \eta_1^3.$$

where Inequality $(\star)$ is due to Lemma D.5.

③ **Tackling Term** $\sum_{l \in \mathbb{L}_{large}^k} \partial_l f(\boldsymbol{w}_{k,0}) a_l^3$:

For any $l \in \mathbb{L}_{large}^k$,

$$|\partial_l f(\boldsymbol{w}_{k,0})a_l^3| \leq \frac{\beta_1}{1-\beta_1}|\eta_{k-1}-\eta_k|\frac{1}{\sqrt{\boldsymbol{\nu}_{l,k-1,n-1}}}|\boldsymbol{m}_{l,k-1,n-1}||\partial_l f(\boldsymbol{w}_{k,0})|$$

$$\leq \frac{\beta_1\eta_1}{(1-\beta_1)}\frac{1}{\sqrt{k}\sqrt{k-1}(\sqrt{k}+\sqrt{k-1})}C_1|\partial_l f(\boldsymbol{w}_{k,0})|$$

$$= \frac{\beta_1\eta_k}{(1-\beta_1)}\frac{1}{\sqrt{k-1}(\sqrt{k}+\sqrt{k-1})}C_1|\partial_l f(\boldsymbol{w}_{k,0})|.$$

Summing over $k$ and $\mathbb{L}_{large}^k$ then leads to

$$\sum_{k=1}^{T}\sum_{l\in\mathbb{L}_{large}^k}|\partial_l f(\boldsymbol{w}_{k,0})a_l^3| \leq \frac{\beta_1}{(1-\beta_1)}\sum_{k=1}^{T}\sum_{l\in\mathbb{L}_{large}^k}\frac{\eta_k}{\sqrt{k-1}(\sqrt{k}+\sqrt{k-1})}C_1|\partial_l f(\boldsymbol{w}_{k,0})|$$

$$\leq 2\frac{\beta_1}{(1-\beta_1)\eta_1}\sqrt{d}C_1\sum_{k=1}^{T}\eta_k^2\|\nabla f(\boldsymbol{w}_{k,0})\|.$$

Put ①, ②, and ③ together and applying the notations in Eq. (9), we then have

$$\sum_{k=1}^{T}\sum_{l\in\mathbb{L}_{large}^k}\partial_l f(\boldsymbol{w}_{k,0})(\boldsymbol{u}_{l,k+1}-\boldsymbol{u}_{l,k})$$

$$\leq -\sum_{k=1}^{T}\sum_{l\in\mathbb{L}_{large}^k}\frac{\eta_k}{\sqrt{\boldsymbol{\nu}_{l,k,0}}}\partial_l f(\boldsymbol{w}_{k,0})^2 + \sum_{k=1}^{T}\sum_{l\in\mathbb{L}_{large}^k}\eta_k g(\beta_2)\left(n-1+\frac{1+\beta_1}{1-\beta_1}\right)\frac{|\partial_l f(\boldsymbol{w}_{k,0})|}{\sqrt{\boldsymbol{\nu}_{l,k,0}}}\left(\max_{i\in[n]}|\partial_l f_i(\boldsymbol{w}_{k,0})|\right)$$

$$+ C_8\sum_{k=1}^{T}\eta_k^2\|\nabla f(\boldsymbol{w}_{k,0})\| + C_9\ln T + C_{10}. \qquad (16)$$

We then focus on the first two terms of the RHS of the above inequality. Specifically, we have $\forall k \geq 1$,

$$\sum_{l\in\mathbb{L}_{large}^k}\frac{\eta_k\partial_l f(\boldsymbol{w}_{k,0})^2}{\sqrt{\boldsymbol{\nu}_{l,k,0}}+\xi} - \sum_{l\in\mathbb{L}_{large}^k}\eta_k g(\beta_2)\left(n-1+\frac{1+\beta_1}{1-\beta_1}\right)\frac{|\partial_l f(\boldsymbol{w}_{k,0})|}{\sqrt{\boldsymbol{\nu}_{l,k,0}}+\xi}\left(\max_{i\in[n]}|\partial_l f_i(\boldsymbol{w}_{k,0})|\right)$$

$$\overset{(\star)}{\geq} \sum_{l\in\mathbb{L}_{large}^k}\frac{\eta_k\partial_l f(\boldsymbol{w}_{k,0})^2}{\sqrt{\boldsymbol{\nu}_{l,k,0}}+\xi} - \sum_{l\in\mathbb{L}_{large}^k}\eta_k g(\beta_2)\left(n-1+\frac{1+\beta_1}{1-\beta_1}\right)\frac{|\partial_l f(\boldsymbol{w}_{k,0})|}{\sqrt{\frac{\beta_2^n}{2n}\max_{i\in[n]}|\partial_l f_i(\boldsymbol{w}_{k,0})|}+\xi}\left(\max_{i\in[n]}|\partial_l f_i(\boldsymbol{w}_{k,0})|\right)$$

$$\geq \sum_{l\in\mathbb{L}_{large}^k}\frac{\eta_k\partial_l f(\boldsymbol{w}_{k,0})^2}{2\max_{i\in[n]}|\partial_l f_i(\boldsymbol{w}_{k,0})|+\xi} - \sum_{l\in\mathbb{L}_{large}^k}\eta_k g(\beta_2)\left(n-1+\frac{1+\beta_1}{1-\beta_1}\right)\frac{|\partial_l f(\boldsymbol{w}_{k,0})|}{\sqrt{\frac{\beta_2^n}{2n}\max_{i\in[n]}|\partial_l f_i(\boldsymbol{w}_{k,0})|}+\xi}\left(\max_{i\in[n]}|\partial_l f_i(\boldsymbol{w}_{k,0})|\right)$$

$$\overset{(\circ)}{=} \sum_{l\in[d]}\frac{\eta_k\partial_l f(\boldsymbol{w}_{k,0})^2}{2\max_{i\in[n]}|\partial_l f_i(\boldsymbol{w}_{k,0})|+\xi} - \sum_{l\in\mathbb{L}_{large}^k}\eta_k g(\beta_2)\left(n-1+\frac{1+\beta_1}{1-\beta_1}\right)\frac{|\partial_l f(\boldsymbol{w}_{k,0})|}{\sqrt{\frac{\beta_2^n}{2n}\max_{i\in[n]}|\partial_l f_i(\boldsymbol{w}_{k,0})|}+\xi}\left(\max_{i\in[n]}|\partial_l f_i(\boldsymbol{w}_{k,0})|\right)$$

$$- \frac{nd\eta_k}{2}\left(C_3\eta_k + C_4\sum_{r=1}^{k-1}\sqrt{\beta_2}^{(r-1)n}\eta_{k-r}\sum_{j=0}^{n-1}\|\nabla f(\boldsymbol{w}_{k-r,j})\| + C_4 n\sum_{r=1}^{k-1}\sqrt{\beta_2}^{(r-1)n}\eta_{k-r} + \eta_k C_4\sum_{j=0}^{n-1}\|\nabla f(\boldsymbol{w}_{k,j})\|\right),$$

where Inequality $(\star)$ is due to Corollary D.9 and Equality $(\circ)$ is due to

$$\sum_{l\in\mathbb{L}_{small}^k}\frac{\eta_k\partial_l f(\boldsymbol{w}_{k,0})^2}{2\max_{i\in[n]}|\partial_l f_i(\boldsymbol{w}_{k,0})|+\xi} \leq \sum_{l\in\mathbb{L}_{small}^k}\frac{\eta_k\partial_l f(\boldsymbol{w}_{k,0})^2}{2\max_{i\in[n]}|\partial_l f_i(\boldsymbol{w}_{k,0})|+\xi} \leq \frac{n}{2}\eta_k\sum_{l\in\mathbb{L}_{small}^k}\max_{i\in[n]}|\partial_l f_i(\boldsymbol{w}_{k,0})|$$

$$\leq \frac{nd\eta_k}{2}\left(C_3\eta_k + C_4\sum_{r=1}^{k-1}\sqrt{\beta_2}^{(r-1)n}\eta_{k-r}\sum_{j=0}^{n-1}\|\nabla f(\boldsymbol{w}_{k-r,j})\| + C_4 n\sum_{r=1}^{k-1}\sqrt{\beta_2}^{(r-1)n}\eta_{k-r} + \eta_k C_4\sum_{j=0}^{n-1}\|\nabla f(\boldsymbol{w}_{k,j})\|\right).$$

Summing the both sides of the above inequality then leads to

$$\sum_{k=1}^{T} \sum_{l \in \mathbb{L}_{large}^k} \frac{\eta_k \partial_l f(\boldsymbol{w}_{k,0})^2}{\sqrt{\boldsymbol{\nu}_{l,k,0}} + \xi} - \sum_{k=1}^{T} \sum_{l \in \mathbb{L}_{large}^k} \eta_k g(\beta_2) \left( n - 1 + \frac{1 + \beta_1}{1 - \beta_1} \right) \frac{|\partial_l f(\boldsymbol{w}_{k,0})|}{\sqrt{\boldsymbol{\nu}_{l,k,0}} + \xi} \left( \max_{i \in [n]} |\partial_l f_i(\boldsymbol{w}_{k,0})| \right)$$

$$\geq \sum_{k=1}^{T} \sum_{l \in [d]} \frac{\eta_k \partial_l f(\boldsymbol{w}_{k,0})^2}{2 \max_{i \in [n]} |\partial_l f_i(\boldsymbol{w}_{k,0})| + \xi}$$

$$- \sum_{k=1}^{T} \sum_{l \in \mathbb{L}_{large}^k} \eta_k g(\beta_2) \left( n - 1 + \frac{1 + \beta_1}{1 - \beta_1} \right) \frac{|\partial_l f(\boldsymbol{w}_{k,0})|}{\sqrt{\frac{\beta_2^n}{2n}} \max_{i \in [n]} |\partial_l f_i(\boldsymbol{w}_{k,0})| + \xi} \left( \max_{i \in [n]} |\partial_l f_i(\boldsymbol{w}_{k,0})| \right)$$

$$- \sum_{k=1}^{T} \frac{nd\eta_k}{2} \left( C_3 \eta_k + C_4 \sum_{r=1}^{k-1} \sqrt{\beta_2}^{(r-1)n} \eta_{k-r} \sum_{j=0}^{n-1} \|\nabla f(\boldsymbol{w}_{k-r,j})\| + C_4 n \sum_{r=1}^{k-1} \sqrt{\beta_2}^{(r-1)n} \eta_{k-r} + \eta_k C_4 \sum_{j=0}^{n-1} \|\nabla f(\boldsymbol{w}_{k,j})\| \right)$$

$$\geq \sum_{k=1}^{T} \sum_{l \in [d]} \frac{\eta_k \partial_l f(\boldsymbol{w}_{k,0})^2}{2 \max_{i \in [n]} |\partial_l f_i(\boldsymbol{w}_{k,0})| + \xi}$$

$$- \sum_{k=1}^{T} \sum_{l \in \mathbb{L}_{large}^k} \eta_k g(\beta_2) \left( n - 1 + \frac{1 + \beta_1}{1 - \beta_1} \right) \frac{|\partial_l f(\boldsymbol{w}_{k,0})|}{\sqrt{\frac{\beta_2^n}{2n}} \max_{i \in [n]} |\partial_l f_i(\boldsymbol{w}_{k,0})| + \xi} \left( \max_{i \in [n]} |\partial_l f_i(\boldsymbol{w}_{k,0})| \right)$$

$$- \frac{1}{2} \left( C_5 \sum_{k=1}^{T} \eta_k^2 \|\nabla f(\boldsymbol{w}_{k,0})\| + C_6 \ln T + C_7 \right).$$

Applying the above inequality back to Eq. (16), the proof is completed. $\square$

The following lemma will be useful when translating $\langle \nabla f(\boldsymbol{w}_{k,0}), \frac{1}{\sqrt{\boldsymbol{\nu}_{k,0}}} \odot \nabla f(\boldsymbol{w}_{k,0}) \rangle$ to $\min \left\{ \frac{\|\nabla f(\boldsymbol{w}_{k,0})\|}{\sqrt{D_1}}, \frac{\|\nabla f(\boldsymbol{w}_{k,0})\|^2}{\sqrt{D_0}} \right\}$.

**Lemma D.13.** *Let all conditions in Theorem 4.1 hold. Then, either there exists a iteration $k \in [T]$, such that either*

$$\|\nabla f(\boldsymbol{w}_{k,0})\| \leq 2\sqrt{d}(2\sqrt{2} + 1)\sqrt{D_0} g(\beta_2) \left( n - 1 + \frac{1 + \beta_1}{1 - \beta_1} \right) \sqrt{\frac{2n}{\beta_2^n}},$$

*or for all iteration $k \in [1, T]$, we have that*

$$\sum_{l \in [d]} \frac{\eta_k \partial_l f(\boldsymbol{w}_{k,0})^2}{2 \max_{i \in [n]} |\partial_l f_i(\boldsymbol{w}_{k,0})| + \xi} - \sum_{l \in [d]} \eta_k g(\beta_2) \left( n - 1 + \frac{1 + \beta_1}{1 - \beta_1} \right) \frac{|\partial_l f(\boldsymbol{w}_{k,0})|}{\sqrt{\frac{\beta_2^n}{2n}} \max_{i \in [n]} |\partial_l f_i(\boldsymbol{w}_{k,0})| + \xi} \left( \max_{i \in [n]} |\partial_l f_i(\boldsymbol{w}_{k,0})| \right)$$

$$\geq \eta_k \frac{1}{2(2\sqrt{2} + 1)} \min \left\{ \frac{\|\nabla f(\boldsymbol{w}_{k,0})\|}{\sqrt{D_1}}, \frac{\|\nabla f(\boldsymbol{w}_{k,0})\|^2}{\xi + \sqrt{D_0}} \right\}.$$

*Proof.* To begin with, we have

$$\sum_{l \in [d]} \frac{\eta_k \partial_l f(\boldsymbol{w}_{k,0})^2}{2 \max_{i \in [n]} |\partial_l f_i(\boldsymbol{w}_{k,0})| + \xi} - \sum_{l \in [d]} \eta_k g(\beta_2) \left( n - 1 + \frac{1 + \beta_1}{1 - \beta_1} \right) \frac{|\partial_l f(\boldsymbol{w}_{k,0})|}{\sqrt{\frac{\beta_2^n}{2n}} \max_{i \in [n]} |\partial_l f_i(\boldsymbol{w}_{k,0})| + \xi} \left( \max_{i \in [n]} |\partial_l f_i(\boldsymbol{w}_{k,0})| \right)$$

$$\overset{(\star)}{\geq} \sum_{l \in [d]} \frac{\eta_k \partial_l f(\boldsymbol{w}_{k,0})^2}{2\sqrt{D_1 \|\nabla f(\boldsymbol{w}_{k,0})\|^2 + D_0} + \xi} - \sum_{l \in [d]} \eta_k g(\beta_2) \left( n - 1 + \frac{1 + \beta_1}{1 - \beta_1} \right) \frac{|\partial_l f(\boldsymbol{w}_{k,0})|}{\sqrt{\frac{\beta_2^n}{2n}} \max_{i \in [n]} |\partial_l f_i(\boldsymbol{w}_{k,0})| + \xi} \left( \max_{i \in [n]} |\partial_l f_i(\boldsymbol{w}_{k,0})| \right)$$

$$= \frac{\eta_k \|\nabla f(\boldsymbol{w}_{k,0})\|^2}{2\sqrt{D_1 \|\nabla f(\boldsymbol{w}_{k,0})\|^2 + D_0} + \xi} - \sum_{l \in [d]} \eta_k g(\beta_2) \left( n - 1 + \frac{1 + \beta_1}{1 - \beta_1} \right) \frac{|\partial_l f(\boldsymbol{w}_{k,0})|}{\sqrt{\frac{\beta_2^n}{2n}} \max_{i \in [n]} |\partial_l f_i(\boldsymbol{w}_{k,0})| + \xi} \left( \max_{i \in [n]} |\partial_l f_i(\boldsymbol{w}_{k,0})| \right),$$

where Inequality $(\star)$ is due to that

$$\max_{i \in [n]} |\partial_l f_i(\boldsymbol{w}_{k,0})| = \sqrt{\max_{i \in [n]} |\partial_l f_i(\boldsymbol{w}_{k,0})|^2}$$

$$\leq \sqrt{\sum_{i \in [n]} \sum_{l'=1}^{d} |\partial_{l'} f_i(\boldsymbol{w}_{k,0})|^2} = \sqrt{\sum_{i \in [n]} \|\nabla f_i(\boldsymbol{w}_{k,0})\|^2} \leq \sqrt{D_1 \|\nabla f(\boldsymbol{w}_{k,0})\|^2 + D_0}.$$

We respectively consider the case $\xi \leq \sqrt{D_0}$ and $\xi > \sqrt{D_0}$.

**Case I: $\xi \leq \sqrt{D_0}$.** In this case, we have that

$$
\frac{\eta_k \|\nabla f(\boldsymbol{w}_{k,0})\|^2}{2\sqrt{D_1 \|\nabla f(\boldsymbol{w}_{k,0})\|^2 + D_0} + \xi} - \sum_{l \in [d]} \eta_k g(\beta_2) \left(n - 1 + \frac{1+\beta_1}{1-\beta_1}\right) \frac{|\partial_l f(\boldsymbol{w}_{k,0})|}{\sqrt{\frac{\beta_2^n}{2n}} \max_{i \in [n]} |\partial_l f_i(\boldsymbol{w}_{k,0})| + \xi} \left(\max_{i \in [n]} |\partial_l f_i(\boldsymbol{w}_{k,0})|\right)
$$

$$
\geq \frac{\eta_k \|\nabla f(\boldsymbol{w}_{k,0})\|^2}{2\sqrt{D_1 \|\nabla f(\boldsymbol{w}_{k,0})\|^2 + D_0} + \sqrt{D_0}} - \sum_{l \in [d]} \eta_k g(\beta_2) \left(n - 1 + \frac{1+\beta_1}{1-\beta_1}\right) \frac{|\partial_l f(\boldsymbol{w}_{k,0})|}{\sqrt{\frac{\beta_2^n}{2n}} \max_{i \in [n]} |\partial_l f_i(\boldsymbol{w}_{k,0})|} \left(\max_{i \in [n]} |\partial_l f_i(\boldsymbol{w}_{k,0})|\right)
$$

$$
= \frac{\eta_k \|\nabla f(\boldsymbol{w}_{k,0})\|^2}{2\sqrt{D_1 \|\nabla f(\boldsymbol{w}_{k,0})\|^2 + D_0} + \sqrt{D_0}} - \sum_{l \in [d]} \eta_k g(\beta_2) \left(n - 1 + \frac{1+\beta_1}{1-\beta_1}\right) \sqrt{\frac{2n}{\beta_2^n}} |\partial_l f(\boldsymbol{w}_{k,0})|
$$

$$
\geq \frac{\eta_k \|\nabla f(\boldsymbol{w}_{k,0})\|^2}{2\sqrt{D_1 \|\nabla f(\boldsymbol{w}_{k,0})\|^2 + D_0} + \sqrt{D_0}} - \sqrt{d} \eta_k g(\beta_2) \left(n - 1 + \frac{1+\beta_1}{1-\beta_1}\right) \sqrt{\frac{2n}{\beta_2^n}} \|\nabla f(\boldsymbol{w}_{k,0})\|.
$$

We further discuss the case depending on whether $\|\nabla f(\boldsymbol{w}_{k,0})\|^2 \leq \frac{D_0}{D_1}$ or not.

**Case I.1: $\|\nabla f(\boldsymbol{w}_{k,0})\|^2 \leq \frac{D_0}{D_1}$.** In this case, the last line of the above equations can be further lower bounded by

$$
\frac{\eta_k \|\nabla f(\boldsymbol{w}_{k,0})\|^2}{2\sqrt{D_1 \|\nabla f(\boldsymbol{w}_{k,0})\|^2 + D_0} + \sqrt{D_0}} - \sqrt{d} \eta_k g(\beta_2) \left(n - 1 + \frac{1+\beta_1}{1-\beta_1}\right) \sqrt{\frac{2n}{\beta_2^n}} \|\nabla f(\boldsymbol{w}_{k,0})\|
$$

$$
\geq \frac{\eta_k \|\nabla f(\boldsymbol{w}_{k,0})\|^2}{(2\sqrt{2}+1)\sqrt{D_0}} - \sqrt{d} \eta_k g(\beta_2) \left(n - 1 + \frac{1+\beta_1}{1-\beta_1}\right) \sqrt{\frac{2n}{\beta_2^n}} \|\nabla f(\boldsymbol{w}_{k,0})\|
$$

$$
= \eta_k \left(\frac{\|\nabla f(\boldsymbol{w}_{k,0})\|}{(2\sqrt{2}+1)\sqrt{D_0}} - \sqrt{d} g(\beta_2) \left(n - 1 + \frac{1+\beta_1}{1-\beta_1}\right) \sqrt{\frac{2n}{\beta_2^n}}\right) \|\nabla f(\boldsymbol{w}_{k,0})\|
$$

**Case I.2: $\|\nabla f(\boldsymbol{w}_{k,0})\|^2 > \frac{D_0}{D_1}$.**

$$
\frac{\eta_k \|\nabla f(\boldsymbol{w}_{k,0})\|^2}{2\sqrt{D_1 \|\nabla f(\boldsymbol{w}_{k,0})\|^2 + D_0} + \sqrt{D_0}} - \sqrt{d} \eta_k g(\beta_2) \left(n - 1 + \frac{1+\beta_1}{1-\beta_1}\right) \sqrt{\frac{2n}{\beta_2^n}} \|\nabla f(\boldsymbol{w}_{k,0})\|
$$

$$
\geq \frac{\eta_k \|\nabla f(\boldsymbol{w}_{k,0})\|^2}{(2\sqrt{2}+1)\sqrt{D_1} \|\nabla f(\boldsymbol{w}_{k,0})\|} - \sqrt{d} \eta_k g(\beta_2) \left(n - 1 + \frac{1+\beta_1}{1-\beta_1}\right) \sqrt{\frac{2n}{\beta_2^n}} \|\nabla f(\boldsymbol{w}_{k,0})\|
$$

$$
= \eta_k \left(\frac{1}{(2\sqrt{2}+1)\sqrt{D_1}} - \sqrt{d} g(\beta_2) \left(n - 1 + \frac{1+\beta_1}{1-\beta_1}\right) \sqrt{\frac{2n}{\beta_2^n}}\right) \|\nabla f(\boldsymbol{w}_{k,0})\|
$$

$$
\overset{(*)}{\geq} \eta_k \frac{1}{2(2\sqrt{2}+1)\sqrt{D_1}} \|\nabla f(\boldsymbol{w}_{k,0})\|,
$$

where Inequality $(*)$ is due to the constraint on $\beta_2$.

Therefore, we have either (1). there exists a iteration $k \in [T]$, such that

$$
\|\nabla f(\boldsymbol{w}_{k,0})\| \leq 2\sqrt{d}(2\sqrt{2}+1)\sqrt{D_0} g(\beta_2) \left(n - 1 + \frac{1+\beta_1}{1-\beta_1}\right) \sqrt{\frac{2n}{\beta_2^n}},
$$

or (2).for all $k \in [1, T]$,

$$
\sum_{l \in [d]} \frac{\eta_k \partial_l f(\boldsymbol{w}_{k,0})^2}{2 \max_{i \in [n]} |\partial_l f_i(\boldsymbol{w}_{k,0})| + \xi} - \sum_{l \in [d]} \eta_k g(\beta_2) \left(n - 1 + \frac{1+\beta_1}{1-\beta_1}\right) \frac{|\partial_l f(\boldsymbol{w}_{k,0})|}{\sqrt{\frac{\beta_2^n}{2n}} \max_{i \in [n]} |\partial_l f_i(\boldsymbol{w}_{k,0})| + \xi} \left(\max_{i \in [n]} |\partial_l f_i(\boldsymbol{w}_{k,0})|\right)
$$

$$
\geq \eta_k \frac{1}{2(2\sqrt{2}+1)} \min\left\{\frac{\|\nabla f(\boldsymbol{w}_{k,0})\|}{\sqrt{D_1}}, \frac{\|\nabla f(\boldsymbol{w}_{k,0})\|^2}{\sqrt{D_0}}\right\}.
$$

**Case II:** $\xi > \sqrt{D_0}$**.** In this case, we have that

$$\frac{\eta_k\|\nabla f(\boldsymbol{w}_{k,0})\|^2}{2\sqrt{D_1\|\nabla f(\boldsymbol{w}_{k,0})\|^2 + D_0} + \xi} - \sum_{l\in[d]} \eta_k g(\beta_2)\left(n-1+\frac{1+\beta_1}{1-\beta_1}\right) \frac{|\partial_l f(\boldsymbol{w}_{k,0})|}{\sqrt{\frac{\beta_2^n}{2n}\max_{i\in[n]}|\partial_l f_i(\boldsymbol{w}_{k,0})| + \xi}} \left(\max_{i\in[n]}|\partial_l f_i(\boldsymbol{w}_{k,0})|\right)$$

$$\geq \frac{\eta_k\|\nabla f(\boldsymbol{w}_{k,0})\|^2}{2\sqrt{D_1\|\nabla f(\boldsymbol{w}_{k,0})\|^2 + \xi^2} + \xi} - \sum_{l\in[d]} \eta_k g(\beta_2)\left(n-1+\frac{1+\beta_1}{1-\beta_1}\right) \frac{|\partial_l f(\boldsymbol{w}_{k,0})|}{\sqrt{\frac{\beta_2^n}{2n}\max_{i\in[n]}|\partial_l f_i(\boldsymbol{w}_{k,0})| + \xi}} \left(\max_{i\in[n]}|\partial_l f_i(\boldsymbol{w}_{k,0})|\right).$$

Similar as **Case I**, we further divides the case regarding the value of $\|\nabla f(\boldsymbol{w}_{k,0})\|$.

**Case II.1:** $D_1\|\nabla f(w_{k,0})\|^2 \leq \xi^2$**.** In this case, we have

$$\frac{\eta_k\|\nabla f(\boldsymbol{w}_{k,0})\|^2}{2\sqrt{D_1\|\nabla f(\boldsymbol{w}_{k,0})\|^2 + \xi^2} + \xi} - \sum_{l\in[d]} \eta_k g(\beta_2)\left(n-1+\frac{1+\beta_1}{1-\beta_1}\right) \frac{|\partial_l f(\boldsymbol{w}_{k,0})|}{\sqrt{\frac{\beta_2^n}{2n}\max_{i\in[n]}|\partial_l f_i(\boldsymbol{w}_{k,0})| + \xi}} \left(\max_{i\in[n]}|\partial_l f_i(\boldsymbol{w}_{k,0})|\right)$$

$$\geq \frac{\eta_k\|\nabla f(\boldsymbol{w}_{k,0})\|^2}{(2\sqrt{2}+1)\xi} - \sum_{l\in[d]} \eta_k g(\beta_2)\left(n-1+\frac{1+\beta_1}{1-\beta_1}\right) \frac{|\partial_l f(\boldsymbol{w}_{k,0})|}{\xi} \left(\max_{i\in[n]}|\partial_l f_i(\boldsymbol{w}_{k,0})|\right)$$

$$\geq \frac{\eta_k\|\nabla f(\boldsymbol{w}_{k,0})\|^2}{(2\sqrt{2}+1)\xi} - \eta_k g(\beta_2)\left(n-1+\frac{1+\beta_1}{1-\beta_1}\right) \frac{\|\nabla f(\boldsymbol{w}_{k,0})\|}{\xi} \sqrt{D_1\|\nabla f(\boldsymbol{w}_{k,0})\|^2 + D_0}$$

$$= \frac{\eta_k\|\nabla f(\boldsymbol{w}_{k,0})\|}{\xi}\left(\frac{\|\nabla f(\boldsymbol{w}_{k,0})\|}{2\sqrt{2}+1} - g(\beta_2)\left(n-1+\frac{1+\beta_1}{1-\beta_1}\right)\sqrt{D_1\|\nabla f(\boldsymbol{w}_{k,0})\|^2 + D_0}\right).$$

**Case II.2:** $D_1\|\nabla f(w_{k,0})\|^2 > \xi^2$**.** This case is quite similar to **Case I.2**, and we have

$$\frac{\eta_k\|\nabla f(\boldsymbol{w}_{k,0})\|^2}{2\sqrt{D_1\|\nabla f(\boldsymbol{w}_{k,0})\|^2 + \xi^2} + \xi} - \sum_{l\in[d]} \eta_k g(\beta_2)\left(n-1+\frac{1+\beta_1}{1-\beta_1}\right) \frac{|\partial_l f(\boldsymbol{w}_{k,0})|}{\sqrt{\frac{\beta_2^n}{2n}\max_{i\in[n]}|\partial_l f_i(\boldsymbol{w}_{k,0})| + \xi}} \left(\max_{i\in[n]}|\partial_l f_i(\boldsymbol{w}_{k,0})|\right)$$

$$\geq \frac{\eta_k\|\nabla f(\boldsymbol{w}_{k,0})\|^2}{(2\sqrt{2}+1)\sqrt{D_1}\|\nabla f(\boldsymbol{w}_{k,0})\|} - \sum_{l\in[d]} \eta_k g(\beta_2)\left(n-1+\frac{1+\beta_1}{1-\beta_1}\right) \frac{|\partial_l f(\boldsymbol{w}_{k,0})|}{\sqrt{\frac{\beta_2^n}{2n}\max_{i\in[n]}|\partial_l f_i(\boldsymbol{w}_{k,0})|}} \left(\max_{i\in[n]}|\partial_l f_i(\boldsymbol{w}_{k,0})|\right)$$

$$\geq \frac{\eta_k\|\nabla f(\boldsymbol{w}_{k,0})\|}{(2\sqrt{2}+1)\sqrt{D_1}} - \sqrt{d}\sqrt{\frac{2n}{\beta_2^n}}\eta_k g(\beta_2)\left(n-1+\frac{1+\beta_1}{1-\beta_1}\right)\|\nabla f(\boldsymbol{w}_{k,0})\|$$

$$= \eta_k\left(\frac{1}{(2\sqrt{2}+1)\sqrt{D_1}} - \sqrt{d}g(\beta_2)\left(n-1+\frac{1+\beta_1}{1-\beta_1}\right)\sqrt{\frac{2n}{\beta_2^n}}\right)\|\nabla f(\boldsymbol{w}_{k,0})\|$$

$$\geq \eta_k\frac{1}{2(2\sqrt{2}+1)\sqrt{D_1}}\|\nabla f(\boldsymbol{w}_{k,0})\|.$$

Therefore, we have either (1). there exists a iteration $k \in [T]$, such that

$$\|\nabla f(\boldsymbol{w}_{k,0})\| \leq 2\sqrt{d}(2\sqrt{2}+1)\sqrt{D_0}g(\beta_2)\left(n-1+\frac{1+\beta_1}{1-\beta_1}\right)\sqrt{\frac{2n}{\beta_2^n}},$$

or (2). for all $k \in [1, T]$,

$$\sum_{l\in[d]} \frac{\eta_k \partial_l f(\boldsymbol{w}_{k,0})^2}{2\max_{i\in[n]}|\partial_l f_i(\boldsymbol{w}_{k,0})| + \xi} - \sum_{l\in[d]} \eta_k g(\beta_2)\left(n-1+\frac{1+\beta_1}{1-\beta_1}\right) \frac{|\partial_l f(\boldsymbol{w}_{k,0})|}{\sqrt{\frac{\beta_2^n}{2n}\max_{i\in[n]}|\partial_l f_i(\boldsymbol{w}_{k,0})| + \xi}} \left(\max_{i\in[n]}|\partial_l f_i(\boldsymbol{w}_{k,0})|\right)$$

$$\geq \eta_k \frac{1}{2(2\sqrt{2}+1)} \min\left\{\frac{\|\nabla f(\boldsymbol{w}_{k,0})\|}{\sqrt{D_1}}, \frac{\|\nabla f(\boldsymbol{w}_{k,0})\|^2}{\xi}\right\}.$$

**As a conclusion of Case I and Case II**, we have that either there exists a iteration $k \in [T]$, such that

$$\|\nabla f(\boldsymbol{w}_{k,0})\| \leq 2\sqrt{d}(2\sqrt{2}+1)\sqrt{D_0}g(\beta_2)\left(n-1+\frac{1+\beta_1}{1-\beta_1}\right)\sqrt{\frac{2n}{\beta_2^n}},$$

or for all iteration $k \in [1, T]$, we have that

$$\sum_{l\in[d]} \frac{\eta_k \partial_l f(\boldsymbol{w}_{k,0})^2}{2\max_{i\in[n]}|\partial_l f_i(\boldsymbol{w}_{k,0})| + \xi} - \sum_{l\in[d]} \eta_k g(\beta_2)\left(n-1+\frac{1+\beta_1}{1-\beta_1}\right) \frac{|\partial_l f(\boldsymbol{w}_{k,0})|}{\sqrt{\frac{\beta_2^n}{2n}\max_{i\in[n]}|\partial_l f_i(\boldsymbol{w}_{k,0})| + \xi}} \left(\max_{i\in[n]}|\partial_l f_i(\boldsymbol{w}_{k,0})|\right)$$

$$\geq \eta_k \frac{1}{2(2\sqrt{2}+1)} \min\left\{\frac{\|\nabla f(\boldsymbol{w}_{k,0})\|}{\sqrt{D_1}}, \frac{\|\nabla f(\boldsymbol{w}_{k,0})\|^2}{\xi + \sqrt{D_0}}\right\}.$$

The proof is completed. □

### D.4 PROOF OF THEOREM 4.1

*Proof of Theorem 4.1.* We start by the descent lemma of $f(\boldsymbol{u}_k)$. Specifically, by Lemma D.6, we have

$$
\begin{aligned}
&f(\boldsymbol{u}_{k+1})\\
&\leq f(\boldsymbol{u}_k) + \langle \nabla f(\boldsymbol{w}_{k,0}), \boldsymbol{u}_{k+1} - \boldsymbol{u}_k \rangle + \frac{nL_0 + L_1\sum_{i\in[n]}\|\nabla f_i(\boldsymbol{w}_{k,0})\|}{2}(\|\boldsymbol{w}_{k,0} - \boldsymbol{u}_k\| + \|\boldsymbol{w}_{k,0} - \boldsymbol{u}_{k+1}\|)\|\boldsymbol{u}_{k+1} - \boldsymbol{u}_k\|\\
&\leq f(\boldsymbol{u}_k) + \langle \nabla f(\boldsymbol{w}_{k,0}), \boldsymbol{u}_{k+1} - \boldsymbol{u}_k \rangle + \frac{nL_0 + L_1\sum_{i\in[n]}\|\nabla f_i(\boldsymbol{w}_{k,0})\|}{2}3C_2^2 d\eta_k^2\\
&\leq f(\boldsymbol{u}_k) + \langle \nabla f(\boldsymbol{w}_{k,0}), \boldsymbol{u}_{k+1} - \boldsymbol{u}_k \rangle + \frac{nL_0 + L_1\sqrt{n}\sqrt{\sum_{i\in[n]}\|\nabla f_i(\boldsymbol{w}_{k,0})\|^2}}{2}3C_2^2 d\eta_k^2\\
&\leq f(\boldsymbol{u}_k) + \langle \nabla f(\boldsymbol{w}_{k,0}), \boldsymbol{u}_{k+1} - \boldsymbol{u}_k \rangle + \frac{nL_0 + L_1\sqrt{n}\sqrt{D_0 + D_1\|\nabla f(\boldsymbol{w}_{k,0})\|^2}}{2}3C_2^2 d\eta_k^2\\
&\leq f(\boldsymbol{u}_k) + \langle \nabla f(\boldsymbol{w}_{k,0}), \boldsymbol{u}_{k+1} - \boldsymbol{u}_k \rangle + \frac{nL_0 + L_1\sqrt{n}(\sqrt{D_0} + \sqrt{D_1}\|\nabla f(\boldsymbol{w}_{k,0})\|)}{2}3C_2^2 d\eta_k^2\\
&\overset{(*)}{=} f(\boldsymbol{u}_k) + \sum_{l\in\mathbb{L}_{large}^k}\partial_l f(\boldsymbol{w}_{k,0})(\boldsymbol{u}_{l,k+1} - \boldsymbol{u}_{l,k}) + \sum_{l\in\mathbb{L}_{small}^k}\partial_l f(\boldsymbol{w}_{k,0})(\boldsymbol{u}_{l,k+1} - \boldsymbol{u}_{l,k})\\
&\quad + \frac{nL_0 + L_1\sqrt{n}\sqrt{D_0}}{2}3C_2^2 d\eta_k^2 + \frac{3L_1\sqrt{n}\sqrt{D_1}C_2^2 d\eta_k^2}{2}\|\nabla f(\boldsymbol{w}_{k,0})\|.
\end{aligned}
$$

Summing the above inequality over $k$ from 1 to $T$ then leads to

$$
\begin{aligned}
f(\boldsymbol{u}_{T+1}) \leq{}& f(\boldsymbol{u}_1) + \sum_{k=1}^{T}\sum_{l\in\mathbb{L}_{large}^k}\partial_l f(\boldsymbol{w}_{k,0})(\boldsymbol{u}_{l,k+1} - \boldsymbol{u}_{l,k}) + \sum_{k=1}^{T}\sum_{l\in\mathbb{L}_{small}^k}\partial_l f(\boldsymbol{w}_{k,0})(\boldsymbol{u}_{l,k+1} - \boldsymbol{u}_{l,k})\\
&+ \sum_{k=1}^{T}\frac{nL_0 + L_1\sqrt{n}\sqrt{D_0}}{2}3C_2^2 d\eta_k^2 + \sum_{k=1}^{T}\frac{3L_1\sqrt{n}\sqrt{D_1}C_2^2 d\eta_k^2}{2}\|\nabla f(\boldsymbol{w}_{k,0})\|.
\end{aligned}
$$

Bounding the second term and the third term of the RHS of the above inequality respectively by Lemma D.12 and Lemma D.11, we then arrive at

$$
\begin{aligned}
f(\boldsymbol{u}_{T+1}) \leq{}& f(\boldsymbol{u}_1) - \sum_{k=1}^{T}\sum_{l\in[d]}\frac{\eta_k\partial_l f(\boldsymbol{w}_{k,0})^2}{2\max_{i\in[n]}|\partial_l f_i(\boldsymbol{w}_{k,0})| + \xi}\\
&+ \sum_{k=1}^{T}\sum_{l\in\mathbb{L}_{large}^k}\eta_k g(\beta_2)\left(n - 1 + \frac{1+\beta_1}{1-\beta_1}\right)\frac{|\partial_l f(\boldsymbol{w}_{k,0})|}{\sqrt{\frac{\beta_2^n}{2n}}\max_{i\in[n]}|\partial_l f_i(\boldsymbol{w}_{k,0})| + \xi}\left(\max_{i\in[n]}|\partial_l f_i(\boldsymbol{w}_{k,0})|\right)\\
&+ \left((C_8 + (\tfrac{1}{2} + C_2)C_5)\sum_{k=1}^{T}\eta_k^2\|\nabla f(\boldsymbol{w}_{k,0})\| + (C_9 + (\tfrac{1}{2} + C_2)C_6)\ln T + (C_{10} + (\tfrac{1}{2} + C_2)C_7)\right)\\
&+ \sum_{k=1}^{T}\frac{nL_0 + L_1\sqrt{n}\sqrt{D_0}}{2}3C_2^2 d\eta_k^2 + \sum_{k=1}^{T}\frac{3L_1\sqrt{n}\sqrt{D_1}C_2^2 d\eta_k^2}{2}\|\nabla f(\boldsymbol{w}_{k,0})\|.
\end{aligned}
$$

Suppose now there does not exist an iteration $k \in [T]$, such that

$$
\|\nabla f(\boldsymbol{w}_{k,0})\| \leq 2\sqrt{d}(2\sqrt{2} + 1)\sqrt{D_0}g(\beta_2)\left(n - 1 + \frac{1+\beta_1}{1-\beta_1}\right)\sqrt{\frac{2n}{\beta_2^n}},
$$

since otherwise, the proof has been completed. By Lemma D.13, we then have

$$
\begin{aligned}
&\sum_{l\in\mathbb{L}_{large}^k} \frac{\eta_k \partial_l f(\boldsymbol{w}_{k,0})^2}{\sqrt{\boldsymbol{\nu}_{l,k,0}}+\xi} - \sum_{l\in\mathbb{L}_{large}^k} \eta_k g(\beta_2)\left(n-1+\frac{1+\beta_1}{1-\beta_1}\right)\frac{|\partial_l f(\boldsymbol{w}_{k,0})|}{\sqrt{\boldsymbol{\nu}_{l,k,0}}+\xi}\left(\max_{i\in[n]}|\partial_l f_i(\boldsymbol{w}_{k,0})|\right)\\
&\geq \eta_k \frac{1}{2(2\sqrt{2}+1)} \min\left\{\frac{\|\nabla f(\boldsymbol{w}_{k,0})\|}{\sqrt{D_1}}, \frac{\|\nabla f(\boldsymbol{w}_{k,0})\|^2}{\xi+\sqrt{D_0}}\right\}.
\end{aligned}
$$

Therefore, we have

$$
\begin{aligned}
&f(\boldsymbol{u}_{T+1}) - f(\boldsymbol{u}_1)\\
&\leq -\sum_{k=1}^T \eta_k \frac{1}{2(2\sqrt{2}+1)} \min\left\{\frac{\|\nabla f(\boldsymbol{w}_{k,0})\|}{\sqrt{D_1}}, \frac{\|\nabla f(\boldsymbol{w}_{k,0})\|^2}{\xi+\sqrt{D_0}}\right\} + \left((\frac{1}{2}+C_2)C_5 + C_8\right)\sum_{k=1}^T \eta_k^2\|\nabla f(\boldsymbol{w}_{k,0})\|\\
&\quad + \left((\frac{1}{2}+C_2)C_6 + C_9\right)\ln T + \left((\frac{1}{2}+C_2)C_7 + C_{10}\right) + \sum_{k=1}^T \frac{nL_0 + L_1\sqrt{n}\sqrt{D_0}}{2}3C_2^2 d\eta_k^2 + \sum_{k=1}^T \frac{3L_1\sqrt{n}\sqrt{D_1}C_2^2 d\eta_k^2}{2}\|\nabla f(\boldsymbol{w}_{k,0})\|\\
&\leq -\sum_{k=1}^T \eta_k \frac{1}{2(2\sqrt{2}+1)} \min\left\{\frac{\|\nabla f(\boldsymbol{w}_{k,0})\|}{\sqrt{D_1}}, \frac{\|\nabla f(\boldsymbol{w}_{k,0})\|^2}{\xi+\sqrt{D_0}}\right\} + \left((\frac{1}{2}+C_2)C_5 + C_8 + \frac{3L_1\sqrt{n}\sqrt{D_1}C_2^2 d}{2}\right)\sum_{k=1}^T \eta_k^2\|\nabla f(\boldsymbol{w}_{k,0})\|\\
&\quad + \left((\frac{1}{2}+C_2)C_6 + C_9 + \frac{nL_0 + L_1\sqrt{n}\sqrt{D_0}}{2}3C_2^2 d\eta_1^2\right)\ln T + \left((\frac{1}{2}+C_2)C_7 + C_{10} + \frac{nL_0 + L_1\sqrt{n}\sqrt{D_0}}{2}3C_2^2 d\eta_1^2\right)\\
&\leq \sum_{k=1}^T \eta_k \frac{1}{2(2\sqrt{2}+1)} \min\left\{\frac{\|\nabla f(\boldsymbol{w}_{k,0})\|}{\sqrt{D_1}}, \frac{\|\nabla f(\boldsymbol{w}_{k,0})\|^2}{\xi+\sqrt{D_0}}\right\} + C_{11}\sum_{k=1}^T \eta_k^2\|\nabla f(\boldsymbol{w}_{k,0})\| + C_{12}\ln T + C_{13},
\end{aligned}
$$

where $C_{11}$, $C_{12}$, and $C_{13}$ is defined as

$$
\begin{aligned}
C_{11} &\triangleq (\frac{1}{2}+C_2)C_5 + C_8 + \frac{3L_1\sqrt{n}\sqrt{D_1}C_2^2 d}{2},\\
C_{12} &\triangleq (\frac{1}{2}+C_2)C_6 + C_9 + \frac{nL_0 + L_1\sqrt{n}\sqrt{D_0}}{2}3C_2^2 d\eta_1^2,\\
C_{13} &\triangleq (\frac{1}{2}+C_2)C_7 + C_{10} + \frac{nL_0 + L_1\sqrt{n}\sqrt{D_0}}{2}3C_2^2 d\eta_1^2.
\end{aligned}
$$

On the other hand, as for $\forall k \in [T]$,

$$
\eta_k^2\|\nabla f(\boldsymbol{w}_{k,0})\| \leq \frac{1}{4}\frac{\sqrt{D_0}+\xi}{\sqrt{D_1}}\eta_k^2 + \frac{\sqrt{D_1}}{\sqrt{D_0}+\xi}\eta_k^2\|\nabla f(\boldsymbol{w}_{k,0})\|^2,
$$

we have that

$$
\begin{aligned}
\eta_k^2\|\nabla f(\boldsymbol{w}_{k,0})\| &\leq \frac{1}{4}\frac{\sqrt{D_0}+\xi}{\sqrt{D_1}}\eta_k^2 + \eta_k^2 \min\left\{\|\nabla f(\boldsymbol{w}_{k,0})\|, \frac{\sqrt{D_1}}{\sqrt{D_0}+\xi}\|\nabla f(\boldsymbol{w}_{k,0})\|^2\right\}\\
&= \frac{1}{4}\frac{\sqrt{D_0}+\xi}{\sqrt{D_1}}\eta_k^2 + \sqrt{D_1}\eta_k^2 \min\left\{\frac{\|\nabla f(\boldsymbol{w}_{k,0})\|}{\sqrt{D_1}}, \frac{\|\nabla f(\boldsymbol{w}_{k,0})\|^2}{\sqrt{D_0}+\xi}\right\},
\end{aligned}
$$

and thus,

$$f(\boldsymbol{u}_{T+1}) - f(\boldsymbol{u}_1)$$

$$\leq -\sum_{k=1}^{T} \eta_k \frac{1}{2(2\sqrt{2}+1)} \min\left\{\frac{\|\nabla f(\boldsymbol{w}_{k,0})\|}{\sqrt{D_1}}, \frac{\|\nabla f(\boldsymbol{w}_{k,0})\|^2}{\xi+\sqrt{D_0}}\right\} + C_{11}\sum_{k=1}^{T} \eta_k^2 \|\nabla f(\boldsymbol{w}_{k,0})\| + C_{12}\ln T + C_{13}$$

$$\leq -\sum_{k=1}^{T} \eta_k \frac{1}{2(2\sqrt{2}+1)} \min\left\{\frac{\|\nabla f(\boldsymbol{w}_{k,0})\|}{\sqrt{D_1}}, \frac{\|\nabla f(\boldsymbol{w}_{k,0})\|^2}{\xi+\sqrt{D_0}}\right\} + \frac{\sqrt{D_0}+\xi}{4\sqrt{D_1}}C_{11}\sum_{k=1}^{T}\eta_k^2 + C_{12}\ln T + C_{13}$$

$$+ \sqrt{D_1}C_{11}\sum_{k=1}^{T}\eta_k^2 \min\left\{\frac{\|\nabla f(\boldsymbol{w}_{k,0})\|}{\sqrt{D_1}}, \frac{\|\nabla f(\boldsymbol{w}_{k,0})\|^2}{\xi+\sqrt{D_0}}\right\}$$

$$\leq -\sum_{k=1}^{T} \eta_k \left(\frac{1}{2(2\sqrt{2}+1)} - \sqrt{D_1}C_{11}\eta_k\right) \min\left\{\frac{\|\nabla f(\boldsymbol{w}_{k,0})\|}{\sqrt{D_1}}, \frac{\|\nabla f(\boldsymbol{w}_{k,0})\|^2}{\xi+\sqrt{D_0}}\right\} + \left(C_{12} + \frac{\sqrt{D_0}+\xi}{4\sqrt{D_1}}C_{11}\eta_1^2\right)\ln T$$

$$+ \left(C_{13} + \frac{\sqrt{D_0}+\xi}{4\sqrt{D_1}}C_{11}\eta_1^2\right)$$

$$\leq -\sum_{k=1}^{T} \eta_k \frac{1}{4(2\sqrt{2}+1)} \min\left\{\frac{\|\nabla f(\boldsymbol{w}_{k,0})\|}{\sqrt{D_1}}, \frac{\|\nabla f(\boldsymbol{w}_{k,0})\|^2}{\xi+\sqrt{D_0}}\right\} + \left(C_{12} + \frac{\sqrt{D_0}+\xi}{4\sqrt{D_1}}C_{11}\eta_1^2\right)\ln T$$

$$+ \left(C_{13} + \frac{\sqrt{D_0}+\xi}{4\sqrt{D_1}}C_{11}\eta_1^2\right).$$

Dividing $\sum_{k=1}^{T}\eta_k$ to the both sides of the above inequality, the proof is completed. $\square$

## E EXPERIMENT DETAILS

This section collects experiments and their corresponding settings, and is arranged as follows: to begin with, we show that Adam works well under the different reshuffling order; we then provide the experiment settings of Figure 1.

### E.1 ADAM WORKS WELL UNDER DIFFERENT RESHUFFLING ORDER

We run Adam on ResNet 110 for CIFAR 10 across different random seeds and plot the 10-run mean and variance in Figure 3. One can observe that the performance of Adam is robust with respect to random seed, and support Theorem 4.1 in terms of trajectory-wise convergence. The experiment is based on this repo, where we adopt the default hyperparameters settings.

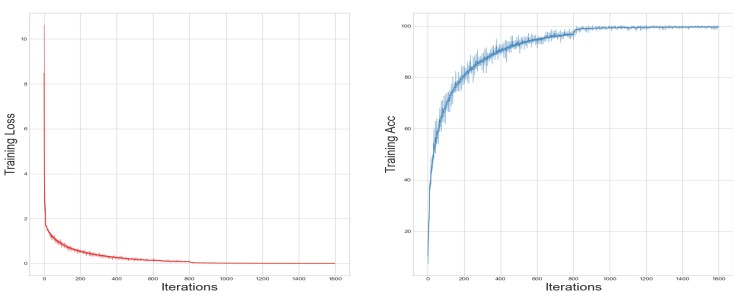

Figure 3: Performance of Adam with different shuffling orders. We respectively plot the training loss and the training accuracy of Adam together with their variances over 10 runs with different random shuffling order. The result indicate the performance of Adam is robust w.r.t. the shuffling order.

### E.2 LOCAL SMOOTHNESS VS. GRADIENT NORM

In this section, we provide the models and hyperparameter settings of Figures 1. We will also illustrate how we evaluate the local smoothness.

**Models and hyper-parameter settings in Figures 1.** In Figure 1, we use exactly the same setting as Vaswani et al. (2017) on WMT 2014 dataset, based on this repo.

**How we evaluate the local smoothness.** We use the same method as Zhang et al. (2019a). Specifically, with a finite-difference step $\alpha$, we calculate the smoothness at $\boldsymbol{w}_k$ as

$$\text{local smoothness} = \max_{\gamma \in \{\alpha, 2\alpha, \cdots, 1\}} \frac{\|\nabla f(\boldsymbol{w}_k + \gamma(\boldsymbol{w}_{k+1} - \boldsymbol{w}_k)) - \nabla f(\boldsymbol{w}_k)\|}{\gamma \|\boldsymbol{w}_{k+1} - \boldsymbol{w}_k\|}.$$

