# OpenReview forum: "Provable Benefit of Adaptivity in Adam"
_ICLR.cc/2024/Conference — ICLR 2024 Conference Withdrawn Submission_

### Official Review · Reviewer_DTjf · 2023-10-31

**Soundness:** 2 fair
**Presentation:** 3 good
**Contribution:** 2 fair
**Rating:** 3
**Confidence:** 4

**Summary:**

This paper provides the convergence analysis of Adam under ($L_0,L_1$)-smooth condition.

**Strengths:**

1. Beyond the $L$-smooth condition, the authors analyze the convergence of Adam under $L_0,L_1$)-smooth condition. Their analysis yields a specific convergence rate (to small error).

2. Moroever, the authors also provide a function on which Adam works but SGD fails.

**Weaknesses:**

1. Several assumptions in this paper differ significantly from standard analysis.
Notably, in Assumption 3.2, the constant $D_1$ should be dependent on the parameter $n$, and $D_1\to\infty$ when $n\to\infty$.
According to the main result in Theorem 4.1, the RHS is on the order of $\sqrt{D_1}$, which falis for large $n$, a critical regime in stocastic optimization.

2. While the authors provide an example that Adam works but SGD fails, there may exist more examples under which Adam fails but SGD works. The results in this paper do not provide insights into the conditions under which Adam is likely to fail, as well as the efficiency of Adam in deep learning.

3. The title suggests a focus on the benefits of "Adaptivity," but this paper lacks a thorough explanation regarding these advantages.

**Questions:**

Please see "Weaknesses".

---

> ### Author Response · Authors · 2023-11-21
> **Response**
>
> We express our appreciation to the reviewer for your constructive feedback. We will thoroughly revise our paper based on these valuable suggestions. However, due to the brevity of the rebuttal period, we are unable to complete this process in time and have decided to withdraw the paper.

---

### Official Review · Reviewer_mpxh · 2023-11-01

**Soundness:** 3 good
**Presentation:** 3 good
**Contribution:** 3 good
**Rating:** 8
**Confidence:** 4

**Summary:**

In this paper, the authors studied the theoretical properties of Adam optimizer. For the existing theoretical analysis, they rely on the bounded smoothness assumption (L-smooth condition), which is not practical in real-world machine learning models. Therefore, the authors proposed the very first convergence result of Adam without assuming the L-smoothness. The L-smoothness is replaced by a (L_0, L_1)-smooth condition, where the norm of Hessian matrix can be upper bounded by a linear combination of constant and gradient norm.

**Strengths:**

1. The writing of this paper is very clear. The theoretical proofs are sound and solid. Although I didn't go through all the proof details in the appendix, I believe all these proofs are theoretically correct.
2. The reference list of this paper is also very complete.
3. The biggest strength is that the authors empirically verified the (L_0, L_1)-smooth condition under the case of LSTM and Transformers, which are both widely used model architecture.

**Weaknesses:**

1. The authors should pay more attention on the details of paper writing. For example, in the last line of Page 2, there is a reference error (Appendix ??).

**Questions:**

For me, there are no additional questions and suggestions. I think this paper worth an "accept" because of its theoretical soundness.

---

> ### Author Response · Authors · 2023-11-21
> **Response**
>
> We express our appreciation to the reviewer for your constructive feedback. We will thoroughly revise our paper based on these valuable suggestions. However, due to the brevity of the rebuttal period, we are unable to complete this process in time and have decided to withdraw the paper.

---

### Official Review · Reviewer_AvTQ · 2023-11-02

**Soundness:** 3 good
**Presentation:** 3 good
**Contribution:** 2 fair
**Rating:** 5
**Confidence:** 4

**Summary:**

This paper studies convergence of the Adam algorithm. Previous work shows that Adam can converge to a bounded region around a stationary point under L-smooth condition. However, L-smoothness does not hold in practical settings (e.g., training of DNNs), so the main focus in this paper is to weaken the assumption to $(L_0, L_1)$-smoothness. The authors show that Adam can converge to a bounded region even under this weaker assumption. Moreover, they also demonstrate the Adam's benefit over SGD by comparing the upper bound of Adam's convergence rate with the lower bound of the SGD's.

**Strengths:**

- They prove the convergence of Adam under the $(L_0, L_1)$-smoothness condition, which is novel.
- Benefits of Adam over SGD is explained via the derived convergence bound.

**Weaknesses:**

- Theoretical contribution is a little minor. Basically, the result is an extension of [1]. Analysis under the $(L_0, L_1)$-smooth assumption is a novel point, but technical novelty of their proof is limited.
- They do not take the bias correction technique of Adam into consideration, which is widely used in practice. I think the result will not be so different even when the bias correction is incorporated, but they should mention it at least.
- They mention that the first term on the right hand side of Eq. (4) is $\Theta ( 1 / (1 - \beta_2)^2 )$. This means that the convergence of Adam deteriorates when $\beta_2$ is chosen to be close to $1$, which seems to contradict to practical observations. They should add discussion on it.

[1] Zhang, Yushun, et al. "Adam can converge without any modification on update rules." Advances in Neural Information Processing Systems 35 (2022): 28386-28399.

**Questions:**

- As I pointed out in the Weakness section, the first term on the right hand side of Eq. (4) is $\Theta ( 1 / (1 - \beta_2)^2 )$, which means that the convergence of Adam deteriorates when $\beta_2$ is chosen to be close to $1$, which seems to contradict to practical observations. How can the gap between theory and practice be explained?

---

> ### Author Response · Authors · 2023-11-21
> **Response**
>
> We express our appreciation to the reviewer for your constructive feedback. We will thoroughly revise our paper based on these valuable suggestions. However, due to the brevity of the rebuttal period, we are unable to complete this process in time and have decided to withdraw the paper.

---

### Official Review · Reviewer_7Yma · 2023-11-06

**Soundness:** 2 fair
**Presentation:** 2 fair
**Contribution:** 2 fair
**Rating:** 3
**Confidence:** 5

**Summary:**

This manuscript provides the convergence analysis for randomly shuffled Adam under the $(L_0,L_1)$-smooth condition and stochastic gradient noise with affine variance. The authors also refine the existing lower bounds of SGD by following existing work [Zhang et al., 2019a].

**Strengths:**

The paper studied an interesting and important problem. Understanding Adam in deep learning is very important.

**Weaknesses:**

1. One of the main theorems (Theorem 4.1) is very weak.

(i) It says that unless $ D_0=0 $, the convergence rate of Adam is worse than SGD. One has to choose very large $\beta_2$ to converge to $\epsilon$-stationary points. As the second term in RHS of formula (4) is of order $1/(1-\beta_2)^2$, and one has to choose $(1-\beta)^2=\epsilon^2$ to make the second term of (4) on the RHS to be small, then $\beta_2$ has to be $1-\epsilon$, and hence $T$ has to be $O(\epsilon^{-8})$ to make the first term of RHS of (4) to be less than $\epsilon^2$. This is much worse than the complexity of SGD. In this setting, the SGD complexity indeed depends on the magnitude of gradient on a sublevel set, but this quantity can be bounded by smoothness constants and independent of $\epsilon$. The $\epsilon$ dependency of SGD is still $\epsilon^{-4}$. Therefore this contradicts the main claim of this paper about the "provable benefit of adaptivity in Adam".

(ii) The Theorem 4.1 requires $O(n\epsilon^{-8})$ gradient calls to find $\epsilon$-stationary points, because the RR-Adam algorithm described in Algorithm 1 denotes $k=1,\ldots,T$ as $T$ as epochs, not $T$ iterations. The hidden $n$ dependency makes this paper extremely weak. For example, there is a simple baseline that significantly dominates RR-Adam analysis in this paper: one can run gradient descent with clipping as in [Zhang et al. 2019a] in a deterministic case (e.g., every epoch), then the number of gradient oracle calls is at most $O(n\epsilon^{-2}$) (Theorem 3 in Zhang et al. 2019a).

(iii) The provable benefit of Adam is proved in [Li et al. 2023] and they prove the $O(\epsilon^{-4})$ gradient complexity of Adam in the setting of stochastic optimization, which I believe is more general than the finite-sum setting as considered in this manuscript and they achieved a much better complexity results.

2. The presentation of the proof of Theorem 4.1 is very difficult to read. There are too many "constants" (e.g., from $C_1$ to $C_{13}$ on Page 18) which depend on each other. I suggest the authors to highlight which term is dominating and what are their orders.

3. The lower bound results are standard extensions of existing lower bounds in [Zhang et al. 2019a, Crawshaw et al. 2022]. As I saw from the proof, the hard instances are the same and the proof idea is also closely following these prior works. The only main difference is concatenating two 1-dimensional instances from previous works to a single 2-dimensional instance. I think this is a very marginal contribution.

Overall, although this paper studied an interesting problem, the upper bound results are too weak and dominated by previous works, and the lower bound results come from straightforward extensions from previous works. Therefore I recommend rejecting this paper.

**Questions:**

Can the authors address my questions? I am happy to increase my score if my concerns can be addressed.

---

> ### Author Response · Authors · 2023-11-21
> **Response**
>
> We express our appreciation to the reviewer for your constructive feedback. We will thoroughly revise our paper based on these valuable suggestions. However, due to the brevity of the rebuttal period, we are unable to complete this process in time and have decided to withdraw the paper. We would like to address a point of contention regarding the comparison of our work with that of [Li et al., 2023], which we believe to be both unfair and inappropriate.
>
> * Firstly, our draft was submitted to flagship conferences nearly a year before [Li et al., 2023] made their submission to arXiv, and our work was even cited by them. We believe that drawing a comparison under such circumstances is not correct.
>
> * Secondly, the analysis presented in our paper holds true when $\lambda = 0$. Here, $\lambda$ represents a term added to $\sqrt{\nu_t}+\lambda$ to improve numerical stability and can be as minimal as $10^{-8}$. Contrarily, Li et al., [2023] operates under the assumption that $\lambda>0$, which creates a vastly different framework. This assumption allows the adaptive learning rate to be instantly upper-bounded, significantly simplifying the analysis. Conversely, in our proof, deriving the upper bound of $\sqrt{\nu_t}$ presents a central and complex challenge. The bounds provided by [Li et al., 2023] show a polynomial dependence on $1/\lambda$, which could potentially result in an extremely loose bound. We acknowledge and appreciate the contribution of [Li et al., 2023] in providing an $O(1/\epsilon4)$ rate in the stochastic setting. However, due to the stark differences in setup, as discussed above, comparing our work with theirs is not even meaningful.